# Curse of Attention:
# A Kernel-Based Perspective for Why Transformers Fail to Generalize on Time Series Forecasting and Beyond

Yekun Ke[1],   Yingyu Liang[2,3],   Zhenmei Shi[3],   Zhao Song[4],   Chiwun Yang[5]

[1]Independent Researcher, [2]The University of Hong Kong, [3]University of Wisconsin-Madison,
[4]The Simons Institute for the Theory of Computing at UC Berkeley, [5]Sun Yat-sen University
keyekun0628@gmail.com, yingyul@hku.hk, yliang@cs.wisc.edu, zhmeishi@cs.wisc.edu,
magic.linuxkde@gmail.com, christiannyang37@gmail.com

The application of transformer-based models on time series forecasting (TSF) tasks has long been popular to study. However, many of these works fail to beat the simple linear residual model, and the theoretical understanding of this issue is still limited. In this work, we propose the first theoretical explanation of the inefficiency of transformers on TSF tasks. We attribute the mechanism behind it to **Asymmetric Learning** in training attention networks. When the sign of the previous step is inconsistent with the sign of the current step in the next-step-prediction time series, attention fails to learn the residual features. This makes it difficult to generalize on out-of-distribution (OOD) data, especially on the sign-inconsistent next-step-prediction data, with the same representation pattern, whereas a linear residual network could easily accomplish it. We hope our theoretical insights provide important necessary conditions for designing the expressive and efficient transformer-based architecture for practitioners.

## 1 Introduction

Attention-based architectures, particularly Transformers, have revolutionized artificial intelligence. Large language models such as Llama [1], Claude-3 [2], GPT-4 [3], and et al. have significantly transformed the AI landscape. Besides, vision models like Vision Transformer(ViT) [4] and Data-efficient Image Transformer(DeiT) [5] have revolutionized the visual domain by directly processing image patches, bypassing the limitations of traditional Convolutional Neural Networks (CNNs). These models demonstrate outstanding performance in fields of natural language processing and computer vision, driving advancements across diverse fields, including content creation [6–8], software development [9–11], multimodal application [12–14], machine translation [15–17] etc.

Time series prediction tasks are crucial for forecasting future trends and have been widely used in making data-driven decisions in various fields, such as finance [18–20], healthcare [21–23] and traffic flow forecasting [24–26]. In addition to their success in NLP, Transformer models have recently gained significant attention in time series prediction tasks. The ability of Transformers to capture involuted patterns and model long-range dependencies has led to their growing adoption in time series prediction tasks, with several recent studies [27–32]. These models utilize self-attention mechanisms to focus on relevant time steps, which makes them particularly well-suited for handling time series data with irregular intervals and high dimensions. Furthermore, some transformer-based methods integrate techniques such as temporal fusion [33], hierarchical attention [34], and patching process [30] etc., allowing them to better capture multi-scale temporal dependencies and adapt to non-stationary patterns in time series.

However, recent studies have challenged the performance of Transformers in time series prediction tasks. Some researchers have found that simple linear layers can outperform more complex Transformers in terms of both accuracy and efficiency [35, 36]. Many works have provided explanations for why Transformer performs worse than simple linear layers on TSF tasks. [35] argue that the

Second Conference on Parsimony and Learning (CPAL 2025).

poor performance of Transformer on TSF tasks stems from its permutation-invariant self-attention mechanism, which results in the loss of temporal information. [37] and [38] attribute the issue to the Transformer's practice of embedding multiple variables into indistinguishable channels, leading to a loss of both variable independence and multivariate correlations. However, there is a lack of theoretical understanding regarding why transformers often perform worse than simple linear models in time series forecasting tasks. To address this gap, we present the first theoretical analysis of this issue, shedding light on the underlying factors contributing to the performance discrepancy.

To demystify the black box, we conducted the following analysis: First, we utilized data generated by the State Space Model (SSM) [39] to model time series data. This approach builds on the work in [40], which demonstrated the SSM's robust modeling capabilities for sequential data. Notably, based on our observation that the linear residual network (N-Linear) [35] performs well in fitting sequential data, we designed a simple task. In this task, the model only needs to apply a straightforward linear mapping to the core features of the time series, which results in relatively small errors.

For the sake of subsequent theoretical analysis, we consider an over-parameterized attention network with a $d = 1$ in our setting where d denotes the input feature dimension, i.e.,

$$f(x, w, a) := \frac{1}{\sqrt{m}} \sum_{r=1}^{m} a_r \cdot \Big\langle \mathsf{softmax}(x_d \cdot w_r \cdot x), x \Big\rangle$$

where $m$ is the hidden neurons number, $a$ and $w$ are the output and hidden layer weights respectively and $x$ is the input data. Our theoretical analysis shows that the training method for next-token prediction induces asymmetric feature updates during gradient descent. Specifically, the parameter $w_r$ will be updated in the direction of the parameter $a_r$. By connecting our setup with vanilla Attention, the above conclusion means that the weights of $W_Q$ and $W_K$ will be updated in the direction of $W_V$. Our results show that in the case of $d = 1$, such asymmetric learning is detrimental to the generalization of sequential data. Then, we introduce inconsistent next-step prediction. Specifically, because $w_r$ updates along the direction of $a_r$, when the model overfits and $a_r = -1$, $w_r$ becomes negative, leading to very small weights for the final timestep feature after applying Softmax. This makes it difficult for the model to learn residual features effectively.

Besides, we further propose a theoretical insight: linear models can exhibit exceptional performance on the task in generalization on SSM sequence data. In contrast, no matter how over-parameterized the attention mechanism is, how large the dataset is, or how long the training time is, it will fail to generalize on SSM sequence data.

Our main contributions can be outlined as follows:

- We demonstrate that asymmetric learning in transformer-based models is the root cause of their underperformance in time series forecasting. Specifically, when the sign of the previous step conflicts with the current step in next-step prediction, the attention mechanism fails to effectively learn residual features, which limits the model's ability to generalize on out-of-distribution (OOD) data.

- We provide a theoretical analysis showing that linear residual models outperform transformers in generalizing to sequential data, as even over-parameterized attention networks fail to match the generalization capability of simple linear models. Moreover, we extend our analysis to $d > 1$ case and discuss several potential solutions for future study.

## 2   Related Work

**Time Series Forecasting.** Time Series Forecasting (TSF) [41–44] is a classical task of predicting future values based on historical data, widely used in finance, weather, traffic, and healthcare. Traditional methods like ARIMA [45] and ETS [46] have been reliable due to their solid theoretical foundations, but they are limited by assumptions such as stability and linearity, affecting real-world accuracy. In recent years, the rapid development of deep learning (DL) has greatly improved

the nonlinear modeling capabilities of time series forecasting (TSF) methods. For example, Liu et al. [47] utilize LSTM [48] for multi-step forecasting in time series tasks and demonstrate that its performance outperforms traditional models. Li et al. [49] present a bidirectional VAE with diffusion, denoise, and disentanglement, improving time series forecasting by augmenting data and enhancing interpretability, outperforming competitive methods in experiments. With Transformer's outstanding performance in NLP and CV, it has quickly been applied to time series forecasting tasks, demonstrating superior performance compared to traditional methods. Notable works include Informer [27], Autoformer [28], FEDformer [29], PatchTST [30], Pyraformer [31], iTransformer [32].

**Neural Tangent Kernel.** The Neural Tangent Kernel (NTK) was initially proposed by Jacot et al. [50] to provide a framework for understanding over-parameterized neural network training behavior. This work showed that, under specific conditions, deep neural network training can be approximated by a linear model, with the NTK governing parameter evolution during gradient descent. Since its introduction, NTK has become a key tool for analyzing training in over-parameterized models. Building on this work, many studies have focused on generalizing the NTK theory to various network architectures at over-parameterization, such as [51–60]. It has been demonstrated that Gradient Descent can effectively train a sufficiently wide neural network and will converge in polynomial time. The NTK technique has gained widespread application in various contexts, including pre-processing analysis [58, 61–64], LoRA adaptation for LLM [65–68], federated learning [69], and estimating scoring functions in diffusion models [70, 71].

**Theory for Understanding Attention Mechanism.** The attention has become a cornerstone in AI, particularly in large language models (LLMs), which excel in NLP tasks such as machine translation, text generation, and sentiment analysis due to their ability to capture complex contextual relationships. However, understanding the attention mechanism from a theoretical perspective remains an ongoing challenge. Several works have explored the theoretical foundations and computational complexities of attention [58, 72–92], focusing on areas such as efficient attention [93–113], optimization [114], and the analysis of emergent abilities [115–125]. Notably, [73] introduced an algorithm with provable guarantees for attention approximation, [126] proved a lower bound for attention computation based on the Strong Exponential Time Hypothesis, and [75] provided both an algorithm and hardness results for static attention computation.

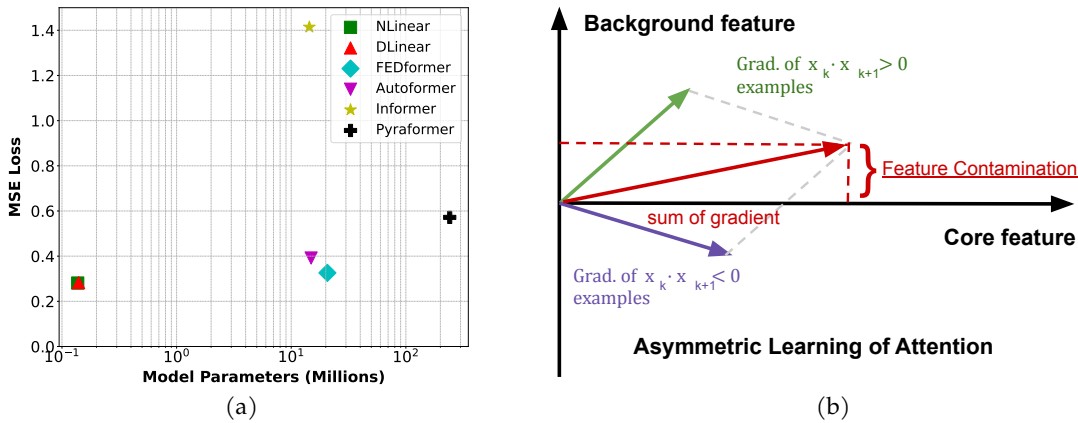

(a)           (b)

Figure 1: (a) We compare the work of previous model [28, 29, 31, 35, 47] on the benchmark dataset ETTh1 and ETTh2. The experimental results show that, even though the simple linear models, NLinear and DLinear, have far fewer parameters than Transformer-based models, they exhibit superior generalization ability on TSF tasks. (b) Theoretical-expected gradient direction of training transformer-based model on TSF tasks. In our setup, we focus on the features at the last time step (also referred to as core features), denoted as $x_{k+1}$ and the features at previous time steps (also referred to as background features), denoted as $x_k$ ($k \in [d]$). Our theoretical findings suggest that the asymmetric feature updates in attention make it difficult for the attention mechanism to learn the recent residual features when the directions of $x_{k+1}$ and $x_k$ are not aligned. In detail, the gradient when training data satisfies $x_k \cdot x_{k+1} < 0$ is contaminated by background features due to the learning disadvantage of attention.

# 3 Background: Transformer Fails to Beat Linear Model in TSF

As a crucial research direction for data science and statistics, time series forecasting (TSF) tasks have played an important role in various domains, including finance analysis, health care, energy management, etc. In recent years, with the outstanding performance of Transformers in the field of Computer Vision (CV) and Natural Language Process (NLP), many studies have applied the Transformer architecture to time series forecasting tasks [27–32, 127]. The primary reason for introducing Transformer-based methods into TSF tasks is their attention mechanism, which effectively models long-range dependencies in the time domain. For instance, Informer [27] introduces the ProbSparse self-attention mechanism and self-attention distilling techniques, enabling Transformer-based methods to handle long sequence time-series forecasting (LSTF) efficiently; FEDformer [29] introduces seasonal-trend decomposition and frequency enhancing techniques, enabling the model to capture global time-series trends; Crossformer [37] introduces the Dimension-Segment-Wise embedding and Two-Stage Attention techniques, enabling Transformer-based models to efficiently capture both cross-time and cross-dimension dependencies for multivariate time series forecasting.

However, it is still debated whether Transformer-based models are more efficient than other deep learning models for time series tasks. The suitability of Transformer-based models for long-term time series forecasting tasks is questioned in [35]. The authors highlight that although these models are effective at capturing semantic correlations, their permutation-invariant self-attention mechanism causes a loss of temporal information. To support this, they introduce a one-layer linear model, LSTF-Linear, which outperforms advanced Transformer-based LTSF models across several TSF Benchmarks. They also suggest revisiting the effectiveness of Transformer-based approaches for TSF tasks. Recently, [128] introduced a novel frequency-domain MLP approach for TSF. By utilizing a global perspective and energy compaction in the frequency domain, this MLP-based method surpasses Transformer-based models, delivering exceptional performance in both short-term and long-term forecasting scenarios. Furthermore, we present the experimental results of existing work [28, 29, 31, 35, 47] on prediction performance on the benchmark datasets ETTh1 and ETTh2, as shown in Figure 1 (a). The data indicates that, despite Transformer-based methods having model parameters 1000 times larger than those of simple linear models, their prediction performance on time series data remains significantly inferior to that of the linear models. This discrepancy raises questions about the effectiveness of such large-scale models in time series forecasting tasks.

Therefore, **why vanilla transformers are not efficient for time series prediction tasks** has become a hotly debated issue recently. A lot of work has shed light on this question: [35] proposed that the permutation-invariant self-attention mechanism may lead to the loss of temporal information. After that, [129] highlights that Transformers in time-series forecasting suffer from overfitting due to their data-dependent attention mechanisms. In contrast, linear models with fixed time-step-dependent weights effectively capture temporal patterns and demonstrate better generalization on datasets with strong temporal dependencies. [32] highlighted the inefficiencies of vanilla Transformer models in time series forecasting, arguing that embedding multiple variables of the same timestamp into a single token results in the loss of crucial multivariate correlations, which hinders the model's ability to capture variable interactions. The token formed at a single time step may fail to capture useful information due to its limited receptive field and the misalignment of events occurring simultaneously. However, the lack of a theoretical explanation behind why the vanilla Transformer model is less efficient than simple linear models in time series tasks remains unexplained. In our paper, we use the NTK framework to analyze and provide a theoretical explanation for the underlying cause of this issue.

# 4 Preliminary: Problem Definition

We present our formal problem definition in this section. In Section 4.1, we introduce the task within our framework, the Residual State Space Model (SSM), and describe how we use the Residual SSM to generate training data. Section 4.2 introduces our two-layer attention model and its training details.

## 4.1 Task and Data

We consider a time series forecasting task with an input space $\mathcal{X} \in \mathbb{R}^d$, a label space $\mathcal{Y} \in \mathbb{R}$, a model class $\mathcal{H} : \mathcal{X} \to \mathbb{R}$, and a loss function $L : \mathcal{Y} \times \mathcal{Y} \to \mathbb{R}$.

For every dataset $\mathcal{D} := \{(x_i, y_i)\}_{i=1}^n$ over $\mathcal{X} \times \mathcal{Y}$ and model $h \in \mathcal{H}$, our training objective of $h$ is given as $L(h) := \frac{1}{2} \sum_{i=1}^n (h(x_i) - y_i)^2$. In our times series forecasting task, there exists a set of distributions $\mathbb{D}$ that consists of all possible distributions to which we would like our model to generalize. In training, we have access to a training distribution set $\mathbb{D}_{train} \subsetneq \mathbb{D}$, where $\mathbb{D}_{train}$ may contain one or multiple training distributions. It's clear that without further assumptions on $\mathbb{D}_{train}$ and $\mathbb{D}$, the time series forecasting task is impossible since no model can generalize to an arbitrary distribution.

In recent years, State Space Models (SSM) [39, 40, 130–135] have been widely applied in various fields, particularly in time series analysis, computer vision, and machine learning. Specifically, [40] offered a simple mathematical explanation for S4's ability to model long-range dependencies and demonstrated the strong performance of S4 and its various variants on benchmark tasks. This also indicates that state space models can represent almost all known time series data. To formalize this, in this work, we assume that our training data and testing data are generated by a residual state space model defined as follows.

**Definition 4.1** (State space model (SSM), informal version of Lemma C.1). *The state space model is defined as follows:*

- *For matrices $\mathcal{A} \in \mathbb{R}^{N \times N}$, $\mathcal{B} \in \mathbb{R}^{N \times 1}$, $\mathcal{C} \in \mathbb{R}^{N \times 1}$*

- *For $k \in [d]$, the state space model is given by:*
$$h_{k+1} := \mathcal{A}h_k + \mathcal{B}u_k \in \mathbb{R}^N$$
$$u_{k+1} := \mathcal{C}^\top h_{k+1} \in \mathbb{R}.$$

- *Denote $\mathcal{K}_k := \mathcal{C}^\top \mathcal{A}^{k-1} \mathcal{B} \in \mathbb{R}$ for $k \in [d]$.*

- *Denote $\mathcal{G}_k := \mathcal{C}^\top \mathcal{A}^{k-1} \in \mathbb{R}^{1 \times N}$ for $k \in [d]$.*

- *We can rewrite $u_k = \sum_{\kappa=1}^{k-1} \mathcal{K}_{k-\kappa} \cdot u_\kappa + \mathcal{G}_k h_1, \forall k \in [d]$.*

Also, we have the following claim about Residual SSM for generating data:

**Claim 4.2** (Residual SSM for generating data, informal version of Claim C.2). *Since we define the state space model in Definition 4.1, we can show that for a initial state $h_1 \in \mathbb{R}^N$, there is:*
$$u_k := \langle \mathcal{P}_k, h_1 \rangle, \ \ \forall k \in [d+1],$$
*where $\mathcal{P}_k := \mathcal{G}_k + \sum_{\kappa=1}^{k-1} \mathcal{K}_{k-\kappa} \cdot \mathcal{P}_\kappa \in \mathbb{R}^N$.*

*Hence, we define residual SSM here. We consider $\{\mathcal{P}_k\}_{k=1}^{d+1} \subset \mathbb{R}^N$ as the features of this SSM. The residual SSM focuses on the last few features to be the core features of the next step prediction. Otherwise, the rest of the features are the background features. We define:*
$$\mathcal{T}_{\text{core}} := \{d - k + 1, \forall k \in [d_0]\}$$
$$\mathcal{T}_{\text{bg}} := [d]/\mathcal{T}_{\text{core}}.$$

*Besides, by choosing some appropriate value for $\mathcal{A}, \mathcal{B}$ and $\mathcal{C}$, we can show that for a certain $\gamma < 1$, we have:*

- *Property 1. The norm for features: $\|\mathcal{P}_k\|_2 = 1, \forall k \in [d+1]$.*

- *Property 2. Similarity of features:*
$$\langle \mathcal{P}_k, \mathcal{P}_{d+1} \rangle = \gamma, \forall k \in \mathcal{T}_{\text{core}}$$
$$\langle \mathcal{P}_{k_1}, \mathcal{P}_{k_2} \rangle = 0, \forall k_1 \in \mathcal{T}_{\text{core}}, k_2 \in \mathcal{T}_{\text{bg}}.$$

- *Property 3. We especially consider $d_0 = 1$.*

With the above definitions, we introduce our data generation model as follows:

**Definition 4.3** (Data Generation, informal version of Definition C.3). *Let the residual state space model be defined as Definition 4.1, then we define the data generator, for $i \in [n]$:*

- *Sample $h_{i,1} \sim \mathcal{N}(0, I_N)$. Generate $u_i = [u_{i,1}, u_{i,2}, \cdots, u_{i,d}, u_{i,d+1}]^\top \in \mathbb{R}^{d+1}$ via Claim 4.2.*

- *Sample $\xi_i \sim \mathcal{N}(0, \sigma \cdot I_{d+1})$ where $\sigma \geq 0$ is a small constant.*

- *$x_i = [x_{i,1}, x_{i,2}, \cdots, x_{i,d}]^\top \in \mathbb{R}^d$ where $x_{i,k} := u_{i,k} + \xi_{i,k}$ for $k \in [d]$.*

- *$y_i := u_{i,d+1} + \xi_{i,d+1} \in \mathbb{R}$.*

*We define the training dataset as $\mathcal{D} := \{(x_i, y_i)\}_{i=1}^n \subset \mathbb{R}^d \times \mathbb{R}$.*

## 4.2 Model and Training.

In this section, we state our model setting and details of its training.

**Model.** In this paper, we consider a two-layer attention model:

$$f(x, w, a) := \frac{1}{\sqrt{m}} \sum_{r=1}^m a_r \cdot \left\langle \mathsf{softmax}(x_d \cdot w_r \cdot x), x \right\rangle$$

with the hidden-layer weights $w(0) := [w_1(0), w_2(0), \cdots, w_m(0)]^\top \in \mathbb{R}^m$ and output-layer weights $a \in \mathbb{R}^m$. To simplify our analysis, we keep output layer weights fixed during training, which is a common assumption in analyzing two-layer neural networks[68, 71, 136]. Such a stylized setting has been widely used for studying the learning behavior of transformer-based models [114, 137, 138], and they gave detailed derivations and guarantees for its connection to attention.

**Assumption: Zero Initialization on Training Data.** For hidden-layer weights, we randomly initialize that $w(0) := [w_1(0), w_2(0), \cdots, w_m(0)]^\top \in \mathbb{R}^m$, where its $r$-th column for $r \in [m]$ is sampled by $w_r(0) \sim \mathcal{N}(0, 1)$. For output layer weights, We randomly initialize $a \in \mathbb{R}^m$ where its $r$-th entry for $r \in [m]$ is sampled by $a_r \sim \mathsf{Uniorm}\{-1, +1\}$. And let training dataset $\mathcal{D} := \{(x_i, y_i)\}_{i=1}^n \subset \mathbb{R}^d \times \mathbb{R}$. Then we assume that $f(x_i, w(0), a) = 0, \forall i \in [n]$ in our setting.

**Training.** Consider a training dataset $\mathcal{D} = \{(x_i, y_i)\}_{i=1}^n$ where the $i$-th data point $(x_i, y_i) \in \mathbb{R}^d \times \mathbb{R}$ which are generated in Definition 4.3. The training loss is measured by the $\ell_2$ norm of the difference between the model prediction and ideal output $y_i$. Formally, the training object is

$$L(w(t)) := \frac{1}{2} \sum_{i=1}^n (f(x_i, w(t), a) - y_i)^2,$$

where $w$ and $a$ denote hidden-layer weights and output-layer weights, respectively. Then, we use gradient descent (GD) to update the trainable weights $w(t)$ with a fixed learning rate $\eta > 0$. Then for $t > 0$, we have

$$w(t + 1) := w(t) - \eta \cdot \nabla_w L(w(t)),$$

where $\eta$ denotes the fixed learning rate in the training process.

# 5 Training Convergence with Asymmetric Learning

In this section, we present the analysis of training convergence with asymmetric learning. In Section 5.1, we will present the key tools we used: the Neural Tangent Kernel (NTK) induced by our model, Kernel Convergence, which is key needed for the NTK analysis, assumptions on NTK, and the associated assumptions. In section 5.2, we present the main result of our paper, which provides a convergence guarantee for asymmetric learning within our framework.

## 5.1 Neural Tangent Kernel

Neural Tangent Kernel (NTK)[50] provides a powerful tool for understanding gradient descent in neural network training, particularly for analyzing the behavior and convergence of deep networks[60, 69, 139–141]. Here, we give the formal definition of NTK in our analysis, which is a kernel function that is driven by hidden-layer weights $w(t) \in \mathbb{R}^{1 \times m}$. To present concisely, we first introduce an operator function in the following. For all $i \in [n]$ and $r \in [m]$, we have

$$\mathsf{u}_{i,r}(t) := \exp(x_{i,r}(t) \cdot w_r(t) \cdot x_i) \in \mathbb{R}^d,$$
$$\alpha_{i,r}(t) := \langle \mathsf{u}_{i,r}(t), \mathbf{1}_d \rangle \in \mathbb{R},$$
$$\mathsf{S}_{i,r}(t) := \alpha_{i,r}(t)^{-1} \cdot \mathsf{u}_{i,r}(t) \in \mathbb{R}^d.$$

Then, we define the kernel matrix $H(t)$ as an $n \times n$ Gram matrix, and the $(i,j)$-th entry of the block is

$$H_{i,j}(t) := \frac{1}{m} x_{i,d} x_{j,d} \sum_{r=1}^{m} \Big( \langle \mathsf{S}_{i,r}(t), x_i^{\circ 2} \rangle - \langle \mathsf{S}_{i,r}(t), x_i \rangle^2 \Big) \cdot \Big( \langle \mathsf{S}_{j,r}(t), x_j^{\circ 2} \rangle - \langle \mathsf{S}_{j,r}(t), x_j \rangle^2 \Big),$$

where we define $x^{\circ 2} := x \circ x$. Here, we introduce the assumption of NTK, which is widely used in literature.

**Assumption on NTK.** In the NTK analysis framework for the convergence of training neural networks, one widely used and mild assumption is that $H^* := H(0)$ is a positive definite (PD) matrix, i.e., its minimum eigenvalue $\lambda := \lambda_{\min}(H^*) > 0$. With this, the theorem of training convergence with Asymmetric Learning is presented as follows.

Next, we introduce the convergence property of the kernel, which is key for the NTK analysis and is formalized below (details in Section F).

**Lemma 5.1** (Kernel Convergence, informal version of Lemma F.4). *For $\delta \in (0, 0.1)$, $B = \max\{1, \sqrt{(1+\sigma^2)\log(nN/\delta)}\}$ and $D = \max\{\sqrt{\log(m/\delta)}, 1\}$. For any $r \in [m]$, we have $|w_r(t) - w_r(0)| \leq R$ and let $R \leq \frac{\lambda}{n \operatorname{poly}(\exp(B^2), \exp(D))}$. Then with probability at least $1 - \delta$, we have $\|H(t) - H(0)\|_F \leq O(nR) \cdot \exp(O(B^2 D))$ and $\lambda_{\min}(H(t)) \geq \lambda/2$.*

*Proof sketch of Lemma 5.1.* For Part 1, we first decompose $|H_{i,j}(t) - H_{i,j}(0)|$ into the sum of four subparts using the triangle inequality. Then, we apply the inequality proven in Lemma K.3 to the upper bound for each part. Then, using the definition of the Frobenius norm, we prove that $\|H(t) - H(0)\|_F \leq O(nR) \cdot \exp(O(B^2 D))$. For Part 2, we can easily get the result by taking the appropriate value of $R$ and Fact B.7. Please see Lemma F.4 for the detailed proof of Lemma 5.1. □

## 5.2 Training Convergence with Asymmetric Learning

Now, we present our first theorem regarding the convergence of training with Asymmetric Learning:

**Theorem 5.2** (Informal version of Theorem I.1). *Given an error $\epsilon > 0$. For $\delta \in (0, 0.1)$, $B = \max\{\sqrt{(1+\sigma^2)\log(nN/\delta)}, 1\}$ and $D = \max\{\sqrt{\log(m/\delta)}, 1\}$. Let $m = \Omega(\operatorname{poly}(\lambda^{-1}, \exp(B^2), \exp(D)), n, d)$ and the learning rate $\eta \leq O(\frac{\lambda\delta}{\operatorname{poly}(\exp(B^2), \exp(D)), n, d})$. Let $T \geq \Omega(\frac{1}{\eta\lambda}\log(nB^2/\epsilon))$, we have: $L(T) \leq \epsilon$.*

*Denote $v_{\min} := \min\{\frac{1}{d}\sum_{k=1}^{d}(x_{i,k} - \overline{x}_i)^2\}_{i=1}^{n}$ where $\overline{x}_i := \frac{1}{d}\sum_{k=1}^{d} x_{i,k}$. The **Asymmetric Learning** of model weights is expressed by $w_r(t)$ updating with $a_r$ as formulated below, for any $t \geq \Omega(\frac{m}{\eta\lambda v_{\min}})$:*

- *Part 1. $\Pr[w_r(t) > 0 | a_r = 1] \geq 1 - \delta$.*

- *Part 2. $\Pr[w_r(t) < 0 | a_r = -1] \geq 1 - \delta$.*

*Proof sketch of Theorem 5.2.* For the upper bound of $L(T)$, we can get the result by combining the result of Part 2 of Lemma H.4, Part 1 of Lemma H.1 and taking the appropriate value of $m, \eta, T$. For the analysis of asymmetric learning, we can get the result by combining the result of Lemma I.3 and taking the appropriate value of $m, \eta, T$. Please see Lemma I.1 for the detailed proof of Lemma 5.2. □

**Residual Feature and Asymmetric Learning.** In our setting, the residual feature represents the feature of the last time step in time series data, which plays a crucial role in the next-step prediction task. For the input data $x \in \mathbb{R}^d$, we take $x_d$ as the residual feature. Our Theorem 5.2 suggests that as training progresses, the direction of the hidden layer weight update $w_r$ tends to align with the sign of the output layer parameters $a_r$. This implies that if $a_r = +1$, $w_r$ is likely to converge to a positive value as the training of the model progresses. Now, we consider the case where the residual feature $x_d$ and the next step label $y$ have the same sign; if $a_r = 1$ and after training progress, the parameter $w_r$ converges to a negative value, the attention score of the residual feature $x_d$ after Softmax function will be extremely small. As a result, the model will struggle to learn the residual feature. A similar analysis can be applied when $x_d$ and $y$ have opposite signs. It depends on $a_r$ taking the value of $-1$ to learn the residual feature effectively.

Based on the analysis above, we have our second main result as follows:

**Theorem 5.3** (Attention fails to learn residual feature, informal version of Theorem I.2). *Let all pre-conditions in Theorem I.1 hold. For any Gaussian vector $x \sim \mathcal{N}(0, \sigma'^2 \cdot I_d)$. For all $r \in [m]$ that satisfies $a_r = -1$, with a probability at least $1 - \delta$, we have:*

$$\mathbb{E}[\mathsf{softmax}_d(x_d \cdot w_r(t) \cdot x)] \leq \mathbb{E}[\mathsf{softmax}_k(x_d \cdot w_r(t) \cdot x)]$$

Please see Lemma I.2 for the proof details of this theorem.

# 6    Attention Fails in Sign-Inconsistent Next-step-prediction

In this section, we define the sign-inconsistent next-step-prediction evaluation task and provide a theoretical analysis of the attention mechanism and residual linear model based on this task. Specifically, we introduce this task in Section 6.1. We present the Residual Linear Network in Section 6.2. In Section 6.3, we give each model a theoretical boundary on this task.

## 6.1    Sign-Inconsistent Next-step-prediction

In this section, we present a new task named Sign-Inconsistent Next-step-prediction. In subsequent sections, we will analyze the theoretical capabilities of the attention mechanism for this task. We define the task formally as follows:

**Definition 6.1.** *Let the residual state space data model be defined as Definition 4.1, then we define the* **sign-inconsistent next-step-prediction** *evaluation task, considering $d = N$:*

1. *Sample $h_{\text{test}} \sim \mathcal{N}(0, I_N)$. Generate $u_{\text{test},i} = [u_{\text{test},i,1}, u_{\text{test},i,2}, \cdots, u_{\text{test},i,d}, u_{\text{test},i,d+1}]^\top \in \mathbb{R}^{d+1}$ via Claim 4.2.*

2. *If $u_{\text{test},i,d} \cdot u_{\text{test},i,d+1} \geq 0$, redo 1.*

3. *Sample $\xi_{\text{test},i} \sim \mathcal{N}(0, \sigma \cdot I_{d+1})$ where $\sigma \geq 0$ is a small constant.*

4. *$x_{\text{test},i} = [x_{\text{test},i,1}, x_{\text{test},i,2}, \cdots, x_{\text{test},i,d}]^\top \in \mathbb{R}^d$ where $x_{\text{test},i,k} := u_{\text{test},i,k} + \xi_{\text{test},i,k}$ for $k \in [d]$.*

5. *$y_{\text{test},i} := u_{\text{test},i,d+1} + \xi_{\text{test},i,d+1} \in \mathbb{R}$.*

*We define the test dataset as $\mathcal{D}_{\text{test}} := \{(x_{\text{test},i}, y_{\text{test},i})\}_{i=1}^{n_{\text{test}}} \subset \mathbb{R}^d \times \mathbb{R}$. Especially, $\{x_i\}_{i=1}^n \cap \{x_{\text{test},i}\}_{i=1}^{n_{\text{test}}} = \emptyset$.*

*For any mapping function $\mathcal{H} : \mathbb{R}^d \to \mathbb{R}$, the OOD risk is given by:*

$$\mathcal{R}(\mathcal{H}) := \mathop{\mathbb{E}}_{h_{\text{test},i} \sim \mathcal{N}(0, I_N)}[(H(x_{\text{test},i}) - y_{\text{test},i})^2]$$

## 6.2    Residual Linear Network

In this section, we present residual linear network[35] mainly to compare it with the attention mechanism on the Sign-Inconsistent next-step-prediction evaluation task. Specifically, the residual

linear network first subtracts the last value of the sequence from the sequence from the input data. This operation removes certain biases or trends in the data, aiming to eliminate unnecessary components that might negatively impact prediction accuracy. Then, the data is passed through a linear layer. This layer can apply more intricate transformations to capture the underlying linear patterns within the data. Finally, the subtracted part will be added back. The purpose of this step is to retain the original characteristics of the data after removing some of the shifts while still benefiting from the transformations applied. The formal definition of the residual linear network is as follows:

**Definition 6.2.** *Given an input vector $x \in \mathbb{R}^d$. Denote $w_{\mathrm{lin}} \in \mathbb{R}^d$ as the model weight. The residual linear network is defined by:*

$$f_{\mathrm{lin}}(x) := \langle w_{\mathrm{lin}}, x - x_d \cdot \mathbf{1}_d \rangle + x_d.$$

## 6.3 Generalizations

This section provides a proposition demonstrating the bound on the OOD risk of the residual linear network and attention mechanisms for the Sign-Inconsistent next-step-prediction evaluation task.

**Proposition 6.3** (Informal version of Proposition J.3). *We have:*

- *Part 1. Let all pre-conditions in Theorem 5.2 hold, there is no $w(t) \in \mathbb{R}^m$ that satisfies $\mathcal{R}(f) \leq \widetilde{O}(\sigma^2)$.*

- *Part 2. There exists and exists only one $w_{\mathrm{lin}}^*$ that satisfies $\sum_{k=1}^{d-1} w_{\mathrm{lin},k}^* \cdot \mathcal{P}_k = \mathcal{P}_{d+1} - \mathcal{P}_d$. Hence, we have $\mathcal{R}(f_{\mathrm{lin}}) \leq \widetilde{O}(\sigma^2)$.*

Please see Proposition J.3 for the detailed proof of this proposition.

**Remark.** Part 1 of Proposition 6.3 shows that even if the width of the hidden layers is sufficient, and the model is trained for a long enough time, the attention mechanism fails to reduce the OOD risk to a sufficiently low level in this task. In contrast, Part 2 shows that a set of parameters exists for the residual linear model that can reduce the OOD risk to the same bound in this task. In conclusion, we theoretically prove that the attention mechanism performs worse than a simple residual linear model on OOD generalization tasks. This proof provides insight into why Transformers underperform on TSF tasks compared to simple linear models.

# 7 Discussion

Based on our theoretical results above, we provide a discussion about **Asymmetric Learning** in the case of the multi-dimensional transformer in Section 7.1 and a discussion about some potential solutions to **Asymmetric Learning** in Section 7.2.

## 7.1 Asymmetric Learning in Multi-Dimension Case

We consider the real-world case of training a transformer-based model. For each layer of attention, we define:

$$\mathrm{Attn}(X, W) := SXW_V,$$

where $X \in \mathbb{R}^{L \times d}$ is the output of previous layer, $S := \mathrm{softmax}(XWX^\top) \in \mathbb{R}^{n \times n}$ is the attention matrix, $L$ is sequence length and $d$ is dimension. Moreover, $W := W_Q W_K^\top / \sqrt{d} \in \mathbb{R}^{d \times d}$ is the combination of query and key projections, $W_V \in \mathbb{R}^{d \times d}$ is the value projection. Therefore, by simple calculation, we have the gradient of $W$ in the back-propagation process. Given the gradient of the next layer $G \in \mathbb{R}^{L \times d}$, we have:

$$\begin{aligned}
\frac{\mathrm{d}L}{\mathrm{d}W} &= \sum_{i=1}^{L} \sum_{j=1}^{d} \frac{\mathrm{d}}{\mathrm{d}W} \mathrm{Attn}_{i,j}(X) \cdot G_{i,j} \\
&= X^\top (S \odot (GW_V^\top X^\top - (\mathrm{Attn}(X, W) \odot G)\mathbf{1}_{d \times n}))X,
\end{aligned}$$

where the $L$ denotes the training objective of the whole model. Thus, we update $W$ using the learning rate $\eta > 0$:

$$X(W - \eta \frac{\mathrm{d}L}{\mathrm{d}W})X^\top = XWX^\top - \eta XX^\top (S \odot (GW_V^\top X^\top - (\mathsf{Attn}(X, W) \odot G)\mathbf{1}_{d \times n}))XX^\top.$$

Since $XX^\top$ is a positive definite matrix, attention matrix $S$ is an all-positive matrix and every column of matrix $(\mathsf{Attn}(X, W) \odot G)\mathbf{1}_{d \times n}$ is provably to equal, affecting little to attention matrix, we suggest the updated attention matrix will greatly depend on the term $GW_V^\top X^\top$.

We now focus on diagonal entries, so-called local entries, for $i \in [n]$, we have the following two cases:

- **Case 1.** When $\langle G_i, W_V^\top X_i \rangle > 0$, attention fails to allocate larger values on local entries but attends to other entries.

- **Case 2.** When $\langle G_i, W_V^\top X_i \rangle < 0$, attention successfully allocates larger values to the local feature.

## 7.2 Potential Solutions

We provide several potential solutions as follows:

- **Differential Transformer.** [142] introduces Differential Transformer that implements $\mathsf{DiffAttn}(X, W) := (S_1 - \lambda S_2)XW_V$ where $\lambda \in (0, 1)$ is a trainable parameter, $S_1 := \mathsf{softmax}(XW_1X)$ and $S_2 := \mathsf{softmax}(XW_2X)$. It amplifies attention to the relevant context while canceling noise, which might be a potential approach to relieving the asymmetric learning in attention.

- **Patching.** Due to the sensitivity of the TSF tasks, patching is a constructive trick that enhances the capability of the attention mechanism to catch precise features [4]. Usually, a patching layer is a reshape transformation, denoted $P(\cdot)$. For a time series $x \in \mathbb{R}^T$ for $T$ steps, $P(x) \in \mathbb{R}^{L \times d}$ not only reshapes the data but also combines padding and reusing it.

- **Rotary Position Embedding (RoPE).** Since the property of long-term decay of RoPE [143], this solution will force the attention to allocate larger values to the local feature. On the other hand, our theoretical results also emphasize the importance of time-varying inductive bias in attention.

- **Gradient Correction, Regularization and Weight Decay.** We believe utilizing the correction or regularization term could help relieve the situation in **Case 2**. For instance, Adam optimizer performs better than SGD in training a transformer. Few prior works have discussed the importance of these terms on fast training and generalization [144–147].

# 8 Conclusion

In this work, we give the first theoretical explanation of the learning mechanism behind the transformer-based models' inefficient performance on TSF tasks. We focus on the attention network to predict the next-step in the time series, whereas we find that the value of output-layer $a_r$ (value projection in attention network) will lead the asymmetric learning to the hidden-weights $w_r$, and it further leads the softmax scores on some important features unavoidably being low value. That is, attention fails to learn the most common behavior in TSF tasks, residual feature (a.k.a differential feature). Our theoretical confirmation could provide more constructive insights for practitioners to design and improve more efficient transformer-based architecture for the field of time series.

# Acknowledgement

We thank all anonymous reviewers for their constructive feedback and helpful discussion.

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

# Appendix

## Contents

# A  Notations

We denote the Gaussian distribution with mean $\mu$ and covariance $\Sigma$ as $\mathcal{N}(\mu, \Sigma)$. For any positive integer $n$, we denote the set $\{1, 2, ..., n\}$ as $[n]$.

In our paper, we use $\mathbb{E}[]$ to denote expectation. We use $\Pr[]$ to denote probability. Given a vector $z \in \mathbb{R}^n$, we represent the $\ell_2$ norm of $z$ as $\|z\|_2 := (\sum_{i=1}^n z_i^2)^{1/2}$. We denote the $\ell_1$ norm of $z$ as $\|z\|_1 := \sum_{i=1}^n |z_i|$ and $\|z\|_0$ as the number of non-zero entries in $z$, $\|z\|_\infty$ as $\max_{i \in [n]} |z_i|$. We use $z^\top$ to denote the transpose of a $z$. We use $\langle \cdot, \cdot \rangle$ to denote the inner product. Given a matrix $A \in \mathbb{R}^{n \times d}$, we use $\mathrm{vec}(A)$ to represent a length $nd$ vector. We use $\|A\|_F := (\sum_{i \in [n], j \in [d]} A_{i,j}^2)^{1/2}$ to represent the Frobenius norm of $A$. For a function $f(x)$, we say $f$ is $L$-Lipschitz if $\|f(x) - f(y)\|_2 \leq L \cdot \|x - y\|_2$. Let $\mathcal{D}$ denote a distribution. We use $x \sim \mathcal{D}$ to denote that we sample a random variable $x$ from distribution $\mathcal{D}$. The p.s.d is denoted the positive-semidefinite matrix.

As we have multiple indexes, to avoid confusion, we usually use $i, j \in [n]$ to index the training data, $\ell \in [d]$ to index the output dimension, $r \in [m]$ to index neuron number.

Given a matrix $X \in \mathbb{R}^{N \times N}$, we define $X^0 = I_N$, $X^1 = X$, $X^2 = X \cdot X$, etc. In this paper, $d$ represents the number of time steps in the time series.

# B  Probability Tools and Facts

Firstly, we present Hoeffding bound lemma as in [148].

**Lemma B.1** (Hoeffding bound, [148]). *Let $Z_1, \ldots, Z_n$ be $n$ independent variables bounded in $[a_i, b_i]$ for $a_i, b_i \in \mathbb{R}$. Define $Z := \sum_{i=1}^n Z_i$, then we will have:*

$$\Pr[|Z - \mathbb{E}[Z]| \geq t] \leq 2 \exp(-\frac{2t^2}{\sum_{i=1}^n (b_i - a_i)^2})$$

Then, we present some useful facts which will be used in our paper.

**Fact B.2.** *For a Gaussian variable $x \sim \mathcal{N}(0, \sigma^2 \cdot I_d)$ where $\sigma \in \mathbb{R}$, then for any $t > 0$, we have:*

$$\Pr[x \leq t] \leq \frac{2t}{\sqrt{2\pi}\sigma}$$

**Fact B.3.** *For $n$ variables $X_i \sim \mathcal{N}(0, \sigma_i^2)$ where $\sigma_i \in \mathbb{R}$ for $i \in [n]$, we have:*

$$\sum_{i=1}^n X_i \sim \mathcal{N}(0, \sum_{i=1}^n \sigma_i^2)$$

**Fact B.4.** *Given a Gaussian vector $x \sim \mathcal{N}(0, \sigma^2 I_d)$ with $\sigma \in \mathbb{R}$, for any fixed $u \in \mathbb{R}^d$, we can show that:*

$$\langle x, u \rangle \sim \mathcal{N}(0, \sigma^2 \|u\|_2^2 \cdot I_d)$$

**Fact B.5.** *For a variable $X \sim \mathcal{N}(0, \sigma^2)$, with probability at least $1 - \delta$, we have:*

$$|X| \leq C\sigma\sqrt{\log(1/\delta)}$$

**Fact B.6.** *For $x \in (-0.01, 0.01)$, the following approximation holds*

$$\exp(x) = 1 + x + \Theta(1)x^2.$$

**Fact B.7.** *For two matrices $H, \widetilde{H} \in \mathbb{R}^{n \times n}$, we have:*

$$\lambda_{\min}(\widetilde{H}) \geq \lambda_{\min}(H) - \|H - \widetilde{H}\|_F$$

**Fact B.8.** *Given a softmax vector $s \in \mathbb{R}^d$ where $\langle s, \mathbf{1}_d \rangle = 1$ and $s_k \geq 0, \forall k \in [d]$ and a vector $x \in \mathbb{R}^d$. We define $\overline{x} := \frac{1}{d}\sum_{k=1}^d x_k$, $v_x := \frac{1}{d}\sum_{k=1}^d (x - \overline{x})^2$. There exists a small constant $c > 0$ such that:*

$$\frac{1}{d}\langle \mathbf{1}_d, x - \mathbf{1}_d \cdot \langle s, x \rangle \rangle \geq c \cdot v_x$$

**Fact B.9.** *For $x \in (0, 1)$, integer $t \geq 0$, we have:*

$$\sum_{\tau=1}^t (1 - x)^\tau \leq -\frac{1}{\log(1 - x)} \leq \frac{2}{x}$$

# C Data

## C.1 Residual State Space Model

**Definition C.1** (State space model (SSM)). *The state space model is defined as follows:*

- *For matrices $\mathcal{A} \in \mathbb{R}^{N \times N}, \mathcal{B} \in \mathbb{R}^{N \times 1}, \mathcal{C} \in \mathbb{R}^{N \times 1}$*

- *For $k \in [d]$, the state space model is given by:*

$$h_{k+1} := \mathcal{A}h_k + \mathcal{B}u_k \in \mathbb{R}^N$$
$$u_{k+1} := \mathcal{C}^\top h_{k+1} \in \mathbb{R}$$

- *Denote $\mathcal{K}_k := \mathcal{C}^\top \mathcal{A}^{k-1}\mathcal{B} \in \mathbb{R}$ for $k \in [d]$.*

- *Denote $\mathcal{G}_k := \mathcal{C}^\top \mathcal{A}^{k-1} \in \mathbb{R}^{1 \times N}$ for $k \in [d]$.*

- *We can rewrite $u_k$ by:*

$$u_k = \sum_{\kappa=1}^{k-1} \mathcal{K}_{k-\kappa} \cdot u_\kappa + \mathcal{G}_k h_1, \forall k \in [d]$$

**Claim C.2** (Residual SSM for generating data). *Since we define the state space model in Definition C.1, we can show that for a initial state $h_1 \in \mathbb{R}^N$, there is:*

$$u_k := \langle \mathcal{P}_k, h_1 \rangle, \forall k \in [d+1],$$

*where $\mathcal{P}_k := \mathcal{G}_k + \sum_{\kappa=1}^{k-1} \mathcal{K}_{k-\kappa} \cdot \mathcal{P}_\kappa \in \mathbb{R}^N$.*

*Hence, we define residual SSM here. We consider $\{\mathcal{P}_k\}_{k=1}^{d+1} \subset \mathbb{R}^N$ as the features of this SSM. The residual SSM focuses on the last few features to be the core features of the next step prediction. Otherwise, the rest of the features are the background features. We define:*

$$\mathcal{T}_{\text{core}} := \{d - k + 1, \forall k \in [d_0]\}$$
$$\mathcal{T}_{\text{bg}} := [d]/\mathcal{T}_{\text{core}}$$

*Besides, by choosing some appropriate value for $\mathcal{A}, \mathcal{B}$ and $\mathcal{C}$, we can show that for a certain $\gamma < 1$, we have:*

- *Property 1. The norm for features:*

$$\|\mathcal{P}_k\|_2 = 1, \forall k \in [d+1]$$

- *Property 2. Similarity of features:*

$$\langle \mathcal{P}_k, \mathcal{P}_{d+1} \rangle = \gamma, \forall k \in \mathcal{T}_{\text{core}}$$
$$\langle \mathcal{P}_{k_1}, \mathcal{P}_{k_2} \rangle = 0, \forall k_1 \in \mathcal{T}_{\text{core}}, k_2 \in \mathcal{T}_{\text{bg}}$$

- *Property 3. Residual features:*

$$\mathcal{P}_{d+1} = \frac{d-1}{d} \sum_{k \in \mathcal{T}_{\text{core}}} \frac{1}{d_0} \cdot \mathcal{P}_k + \frac{1}{d} \sum_{k \in \mathcal{T}_{\text{bg}}} \frac{1}{d - d_0} \cdot \mathcal{P}_k$$

- *Property 4. We especially consider $d_0 = 1$.*

*Proof.* We have:

$$u_k = \sum_{\kappa=1}^{k-1} \mathcal{K}_{k-\kappa} \cdot u_\kappa + \mathcal{G}_k h_1, \forall k \in [d]$$
$$= \langle \mathcal{P}_k, h_1 \rangle$$

Above, the first equation is trivially from Definition C.1. The 2nd equation is based on simple algebra and the definition of $\mathcal{P}_k, \forall k \in [d]$. □

## C.2 ID Data Generator

**Definition C.3.** *We define the residual state space model as specified in Definition C.1. Then, we are able to define the data generator for $i \in [n]$:*

- *Sample $h_{i,1} \sim \mathcal{N}(0, I_N)$.*

- *Generate $u_i = [u_{i,1}, u_{i,2}, \cdots, u_{i,d}, u_{i,d+1}]^\top \in \mathbb{R}^{d+1}$ via Claim C.2.*

- *Sample $\xi_i \sim \mathcal{N}(0, \sigma \cdot I_{d+1})$ where $\sigma \geq 0$ is a small constant.*

- *$x_i = [x_{i,1}, x_{i,2}, \cdots, x_{i,d}]^\top \in \mathbb{R}^d$ where $x_{i,k} := u_{i,k} + \xi_{i,k}$ for $k \in [d]$.*

- *$y_i := u_{i,d+1} + \xi_{i,d+1} \in \mathbb{R}$.*

*We define the test dataset as $\mathcal{D} := \{(x_i, y_i)\}_{i=1}^n \subset \mathbb{R}^d \times \mathbb{R}$.*

## C.3 OOD Sign-Inconsistent Next-step-prediction Task

**Definition C.4.** *Let the residual state space data model be defined in Definition C.1. Then we can define the **sign-inconsistent next-step-prediction** evaluation task:*

1. *Sample $h_{\text{test}} \sim \mathcal{N}(0, I_n)$. Generate $u_{\text{test},i} = [u_{\text{test},i,1}, u_{\text{test},i,2}, \cdots, u_{\text{test},i,d}, u_{\text{test},i,d+1}]^\top \in \mathbb{R}^{d+1}$ via Claim C.2.*

2. *If $u_{\text{test},i,d} \cdot u_{\text{test},i,d+1} \geq 0$, redo 1.*

3. *Sample $\xi_{\text{test},i} \sim \mathcal{N}(0, \sigma \cdot I_{d+1})$ where $\sigma \geq 0$ is a small constant.*

4. *$x_{\text{test},i} = [x_{\text{test},i,1}, x_{\text{test},i,2}, \cdots, x_{\text{test},i,d}]^\top \in \mathbb{R}^d$ where $x_{\text{test},i,k} := u_{\text{test},i,k} + \xi_{\text{test},i,k}$ for $k \in [d]$.*

5. *$y_{\text{test},i} := u_{\text{test},i,d+1} + \xi_{\text{test},i,d+1} \in \mathbb{R}$.*

*We define the test dataset as $\mathcal{D}_{\text{test}} := \{(x_{\text{test},i}, y_{\text{test},i})\}_{i=1}^{n_{\text{test}}} \subset \mathbb{R}^d \times \mathbb{R}$. Especially, $\{x_i\}_{i=1}^n \cap \{x_{\text{test},i}\}_{i=1}^{n_{\text{test}}} = \emptyset$.*
*The OOD risk is given by:*

$$\mathcal{R}(H) := \lim_{n_{\text{text}} \to +\infty} \frac{1}{n_{\text{text}}} \sum_{i=1}^{n_{\text{test}}} (H(x_{\text{test},i}) - y_{\text{test},i})^2$$

# D Problem Setup

## D.1 Weights and Initialization

**Definition D.1.** *We give the following definitions:*

- **Hidden-layer weights** $w \in \mathbb{R}^m$**.** *We define the hidden-layer weights $w := [w_1, w_2, \cdots, w_m]^\top \in \mathbb{R}^m$ where $w_r \in \mathbb{R}, \forall r \in [m]$.*

- **Output-layer weights** $a \in \mathbb{R}^m$**.** *We define the output-layer weights $a := [a_1, a_2, \cdots, a_m]^\top \in \mathbb{R}^m$, especially, vector $a$ is fixed during the training.*

**Definition D.2.** *We give the following initialization:*

- **Initialization of hidden-layer weights** $w \in \mathbb{R}^m$**.** *We randomly initialize that $w(0) := [w_1(0), w_2(0), \cdots, w_m(0)]^\top \in \mathbb{R}^m$, where its $r$-th column for $r \in [m]$ is sampled by $w_r(0) \sim \mathcal{N}(0, 1)$.*

- **Initialization of output-layer weights** $a \in \mathbb{R}^m$**.** *We randomly initialize $a \in \mathbb{R}^m$ where its $r$-th entry for $r \in [m]$ is sampled by $a_r \sim \text{Uniorm}\{-1, +1\}$.*

## D.2   Model

**Definition D.3.** *Suppose we have the following:*

- *For a input vector $x \in \mathbb{R}^d$.*

- *For a hidden-layer weights $w \in \mathbb{R}^m$ as Definition D.1.*

- *For a output-layer weights $a \in \mathbb{R}^m$ as Definition D.1.*

*We define:*

$$f(x, w, a) := \frac{1}{\sqrt{m}} \sum_{r=1}^{m} a_r \cdot \Big\langle \mathsf{softmax}(x_d \cdot w_r \cdot x), x \Big\rangle$$

## D.3   Training

**Definition D.4.** *Suppose we have the following:*

- *Initialize $w(0) \in \mathbb{R}^m$ as specified in Definition D.2.*

- *Initialize $a \in \mathbb{R}^m$ as specified in Definition D.2*

- *Define $f : \mathbb{R}^d \times \mathbb{R}^m \times \mathbb{R}^m \to \mathbb{R}$ as specified in Definition D.3.*

- *For any $t \geq 0$.*

- *Let training dataset $\mathcal{D} := \{(x_i, y_i)\}_{i=1}^{n} \subset \mathbb{R}^d \times \mathbb{R}$ be given in Definition C.3.*

*Then we can define:*

$$L(t) := \frac{1}{2} \sum_{i=1}^{n} (f(x_i, w(t), a) - y_i)^2$$

**Definition D.5.** *Assuming the following conditions are satisfied:*

- *Initialize $w(0) \in \mathbb{R}^m$ as specified in Definition D.2.*

- *Initialize $a \in \mathbb{R}^m$ as specified in Definition D.2*

- *Define $f : \mathbb{R}^d \times \mathbb{R}^m \times \mathbb{R}^m \to \mathbb{R}$ as specified in Definition D.3.*

- *Define $L(t)$ be specified in Definition D.4.*

- *Denote the learning rate $\eta > 0$.*

*We define:*

$$\begin{aligned} w(t+1) &:= w(t) - \eta \cdot \Delta w(t) \\ &= w(t) - \eta \nabla_{w(t)} L(t) \end{aligned}$$

## D.4   Evaluation

**Definition D.6.** *Assuming we have the following conditions:*

- *Initialize $w(0) \in \mathbb{R}^m$ as specified in Definition D.2.*

- *Initialize $a \in \mathbb{R}^m$ as specified in Definition D.2*

- *Define $f : \mathbb{R}^d \times \mathbb{R}^m \times \mathbb{R}^m \to \mathbb{R}$ as specified in Definition D.3.*

- *Let test dataset $\mathcal{D}_{\text{test}} := \{(x_{\text{test},i}, y_{\text{test},i})\}_{i=1}^{n_{\text{test}}} \subset \mathbb{R}^d \times \mathbb{R}$ be defined as Definition C.3.*

*We define:*

$$L_{\text{test}}(t) := \frac{1}{2n_{\text{test}}} \cdot \sum_{i=1}^{n} (f(x_{\text{test},i}, w(t), a) - y_{\text{test},i})^2$$

## D.5 Assumption 1: Zero Initialization on Training Data

**Assumption D.7.** *Assuming the following conditions are satisfied:*

- *Initialize $w(0) \in \mathbb{R}^m$ as specified in Definition D.2.*
- *Initialize $a \in \mathbb{R}^m$ as specified in Definition D.2*
- *Let $f : \mathbb{R}^d \times \mathbb{R}^m \times \mathbb{R}^m \to \mathbb{R}$ be given in Definition D.3.*
- *Let training dataset $\mathcal{D} := \{(x_i, y_i)\}_{i=1}^{n} \subset \mathbb{R}^d \times \mathbb{R}$ be defined in Definition D.4.*

*We assume that:*

$$f(x_i, w(0), a) = 0, \forall i \in [n]$$

# E Gradient Descent

## E.1 Simplifications

**Definition E.1.** *Suppose we have the following:*

- *Initialize $w(0) \in \mathbb{R}^m$ as specified in Definition D.2.*
- *Initialize $a \in \mathbb{R}^m$ as specified in Definition D.2.*
- *Let $f : \mathbb{R}^d \times \mathbb{R}^m \times \mathbb{R}^m \to \mathbb{R}$ be given in Definition D.3.*
- *Let training dataset $\mathcal{D} := \{(x_i, y_i)\}_{i=1}^{n} \subset \mathbb{R}^d \times \mathbb{R}$ be specified in Definition D.4.*
- *For $i \in [n]$, $r \in [m]$.*
- *For any $t \geq 0$.*

*Now, We define $\mathsf{u}_{i,r}(t)$ as the following:*

$$\mathsf{u}_{i,r}(t) := \exp\left(x_{i,d} \cdot w_r(t) \cdot x_i\right) \in \mathbb{R}^d$$

**Definition E.2.** *Suppose we have the following:*

- *Initialize $w(0) \in \mathbb{R}^m$ as specified in Definition D.2.*
- *Initialize $a \in \mathbb{R}^m$ as specified in Definition D.2.*
- *For any $t \geq 0$.*
- *Let training dataset $\mathcal{D} := \{(x_i, y_i)\}_{i=1}^{n} \subset \mathbb{R}^d \times \mathbb{R}$ be specified as Definition D.4.*
- *For $i \in [n]$, $r \in [m]$.*
- *Define $\mathsf{u}_{i,r}(t) \in \mathbb{R}^d$ as specified in Definition E.1.*

*We define:*

$$\alpha_{i,r}(t) = \langle \mathsf{u}_{i,r}(t), \mathbf{1}_d \rangle \in \mathbb{R}$$

**Definition E.3.** *Suppose we have the following:*

- *Initialize $w(0) \in \mathbb{R}^m$ as specified in Definition D.2.*
- *Initialize $a \in \mathbb{R}^m$ as specified in Definition D.2.*
- *For any $t \geq 0$.*
- *Let training dataset $\mathcal{D} := \{(x_i, y_i)\}_{i=1}^n \subset \mathbb{R}^d \times \mathbb{R}$ be specified as Definition D.4.*
- *For $i \in [n]$, $r \in [m]$.*
- *Define $\mathsf{u}_{i,r}(t) \in \mathbb{R}^d$ as specified in Definition E.1.*
- *Define $\alpha_{i,r}(t) \in \mathbb{R}$ as specified in Definition E.1.*

*We define:*

$$\mathsf{S}_{i,r}(t) = \alpha_{i,r}(t)^{-1} \cdot \mathsf{u}_{i,r}(t) \in \mathbb{R}^d$$

**Definition E.4.** *Suppose we have the following:*

- *Initialize $w(0) \in \mathbb{R}^m$ as specified in Definition D.2.*
- *Initialize $a \in \mathbb{R}^m$ as specified in Definition D.2.*
- *For any $t \geq 0$.*
- *Let training dataset $\mathcal{D} := \{(x_i, y_i)\}_{i=1}^n \subset \mathbb{R}^d \times \mathbb{R}$ be specified as Definition D.4.*
- *For $i \in [n]$, $r \in [m]$.*
- *Define $\mathsf{u}_{i,r}(t) \in \mathbb{R}^d$ as specified in Definition E.1.*
- *Define $\alpha_{i,r}(t) \in \mathbb{R}$ as specified in Definition E.1.*
- *Define $\mathsf{S}_{i,r}(t) \in \mathbb{R}$ as specified in Definition E.3.*

*We define:*

$$\mathsf{F}_i(t) := \frac{1}{\sqrt{m}} \sum_{r=1}^m a_r \cdot \left\langle \mathsf{S}_{i,r}(t), x_i \right\rangle \in \mathbb{R}$$

## E.2 Gradient Computations

**Lemma E.5.** *Suppose we have the following:*

- *Initialize $w(0) \in \mathbb{R}^m$ as specified in Definition D.2.*
- *Initialize $a \in \mathbb{R}^m$ as specified in Definition D.2.*
- *For any $t \geq 0$.*
- *Let training dataset $\mathcal{D} := \{(x_i, y_i)\}_{i=1}^n \subset \mathbb{R}^d \times \mathbb{R}$ be specified as Definition C.3.*
- *For $i \in [n]$, $r \in [m]$, $k \in [d]$.*
  *item Define $\mathsf{u}_{i,r}(t) \in \mathbb{R}^d$ as specified in Definition E.1.*
- *Define $\alpha_{i,r}(t) \in \mathbb{R}$ as specified in Definition E.1.*
- *Define $\mathsf{S}_{i,r}(t) \in \mathbb{R}$ as specified in Definition E.3.*
- *Define $\mathsf{F}_i(t) \in \mathbb{R}$ as specified in Definition E.4.*
- *Define $L(t) \in \mathbb{R}$ as specified in Definition D.4.*

*Then we have*

- *Part 1.*

$$\frac{d}{dw_r(t)}\mathsf{u}_{i,r}(t) = x_{i,d} \cdot \mathsf{u}_{i,r}(t) \circ x_i \in \mathbb{R}^d$$

- *Part 2.*

$$\frac{d}{dw_r(t)}\alpha_{i,r}(t) = x_{i,d}\langle \mathsf{u}_{i,r}(t) \circ x_i, \mathbf{1}_d\rangle \in \mathbb{R}$$

- *Part 3.*

$$\frac{d}{dw_r(t)}\alpha_{i,r}(t)^{-1} = -x_{i,d} \cdot \alpha_{i,r}(t)^{-1}\langle \mathsf{S}_{i,r}(t) \circ x_i, \mathbf{1}_d\rangle \in \mathbb{R}$$

- *Part 4.*

$$\frac{d}{dw_r(t)}\mathsf{S}_{i,r}(t) = x_{i,d}\Big(x_i - \langle \mathsf{S}_{i,r}(t), x_i\rangle \cdot \mathbf{1}_d\Big) \circ \mathsf{S}_{i,r}(t) \in \mathbb{R}^d$$

- *Part 5.*

$$\frac{d}{dw_r(t)}\mathsf{F}_i(t) = \frac{1}{\sqrt{m}} \cdot a_r x_{i,d} \cdot \Big(\langle \mathsf{S}_{i,r}(t), x_i^{\circ 2}\rangle - \langle \mathsf{S}_{i,r}(t), x_i\rangle^2\Big) \in \mathbb{R}$$

- *Part 6.*

$$\frac{d}{dw_r(t)}L(t) = \frac{1}{\sqrt{m}} a_r \sum_{i=1}^{n}(\mathsf{F}_i(t) - y_i) \cdot x_{i,d} \cdot \Big(\langle \mathsf{S}_{i,r}(t), x_i^{\circ 2}\rangle - \langle \mathsf{S}_{i,r}(t), x_i\rangle^2\Big) \in \mathbb{R}$$

*Proof.* **Proof of Part 1.** Consider the following reasoning:

$$\frac{d}{dw_r(t)}\mathsf{u}_{i,r}(t) = \exp(x_{i,d} \cdot w_r(t) \cdot x_i) \cdot x_{i,d} \circ x_i$$
$$= x_{i,d} \cdot \mathsf{u}_{i,r}(t) \circ x_i$$

where the first equation is due to simple differential rules, and the second step is trivially from Definition E.1.

**Proof of Part 2.** We have

$$\frac{d}{dw_r(t)}\alpha_{i,r}(t) = \langle\frac{d}{dw_r(t)}\mathsf{u}_{i,r}, \mathbf{1}_d\rangle$$
$$= x_{i,d} \cdot \langle \mathsf{u}_{i,r}(t) \circ x_i, \mathbf{1}_d\rangle$$

where the first equation is due to Definition E.2, and the second step is because of part 1 of this lemma.

**Proof of Part 3.** We have

$$\frac{d}{dw_r(t)}\alpha_{i,r}(t)^{-1} = -\alpha_{i,r}(t)^{-2} \cdot \frac{d}{dw_r(t)}\alpha_{i,r}(t)$$
$$= -x_{i,d} \cdot \alpha_{i,r}(t)^{-2} \cdot \langle \mathsf{u}_{i,r}(t) \circ x_i, \mathbf{1}_d\rangle$$
$$= -x_{i,d} \cdot \alpha_{i,r}(t)^{-1} \cdot \langle (\mathsf{u}_{i,r}(t) \cdot \alpha_{i,r}(t)^{-1}) \circ x_i, \mathbf{1}_d\rangle$$
$$= -x_{i,d} \cdot \alpha_{i,r}(t)^{-1} \cdot \langle \mathsf{S}_{i,r}(t) \circ x_i, \mathbf{1}_d\rangle$$

where is a result of applying the chain rule, the second step is derived from Part 2 of this Lemma, the third step is a consequence of basic algebraic manipulation, and the last step follows from Definition E.3.

**Proof of Part 4.** We have

$$\frac{d}{dw_r(t)}\mathsf{S}_{i,r}(t) = \frac{d}{dw_r(t)}(\alpha_{i,r}(t)^{-1} \cdot \mathsf{u}_{i,r}(t))$$

$$= (\frac{\mathrm{d}}{\mathrm{d}w_r(t)}\alpha_{i,r}(t)^{-1}) \cdot \mathsf{u}_{i,r}(t) + (\frac{\mathrm{d}}{\mathrm{d}w_r(t)}\mathsf{u}_{i,r}(t)) \cdot \alpha_{i,r}(t)^{-1}$$

$$= -x_{i,d} \cdot \alpha_{i,r}(t)^{-1} \cdot \langle \mathsf{S}_{i,r}(t) \circ x_i, \mathbf{1}_d \rangle \cdot \mathsf{u}_{i,r}(t) + x_{i,d} \cdot \alpha_{i,r}(t)^{-1} \cdot \mathsf{u}_{i,r}(t) \circ x_i$$

$$= -x_{i,d} \cdot \langle S_{i,r}(t), x_i \rangle \cdot \mathbf{1}_d \circ S_{i,r}(t) + x_{i,d} \cdot S_{i,r}(t) \circ x_i$$

$$= x_{i,d} \cdot (x_i - \langle S_{i,r}(t), x_i \rangle \cdot \mathbf{1}_d) \circ S_{i,r}(t)$$

where the first step is based on Definition E.3, the second step is derived using basic differentiation rules, the third step is based on Part 1 and 3, and the last two result from straightforward algebraic manipulation.

**Proof of Part 5.** We have

$$\frac{\mathrm{d}}{\mathrm{d}w_r(t)}\mathsf{F}_i(t) := \frac{1}{\sqrt{m}}\sum_{j=1}^{m} a_j \cdot \langle \frac{\mathrm{d}}{\mathrm{d}w_r(t)}S_{i,j}(t), x_i \rangle$$

$$= \frac{1}{\sqrt{m}}a_r \cdot \langle x_{i,d} \cdot (x_i - \langle S_{i,r}(t), x_i \rangle \cdot \mathbf{1}_d) \circ S_{i,r}(t), x_i \rangle$$

$$= \frac{1}{\sqrt{m}}a_r x_{i,d}(\langle S_{i,r}(t), x_i^{\circ 2} \rangle - \langle (\langle S_{i,r}(t), x_i \rangle \cdot \mathbf{1}_d) \circ S_{i,r}(t), x_i \rangle)$$

$$= \frac{1}{\sqrt{m}}a_r x_{i,d}(\langle S_{i,r}(t), x_i^{\circ 2} \rangle - \langle S_{i,r}(t), x_i \rangle^2)$$

where the first step is a consequence of Definition E.4, the second step is derived from Part 4 of this lemma, and the third and last steps result from basic algebraic operations.

**Proof of Part 6.** We have

$$\frac{\mathrm{d}}{\mathrm{d}w_r(t)}L(t) = \sum_{i=1}^{n}(F_i(t) - y_i)\frac{\mathrm{d}}{\mathrm{d}w_r(t)}F_i(t)$$

$$= \frac{1}{\sqrt{m}}a_r \sum_{i=1}^{n}(F_i(t) - y_i) \cdot x_{i,d} \cdot (\langle S_{i,r}(t), x_i^{\circ 2} \rangle - \langle S_{i,r}(t), x_i \rangle^2)$$

where the first step is based on Definition D.4, while the second step is derived from Part 5 of this Lemma. $\square$

# F   Neural Tangent Kernel

## F.1   Kernel Function

**Definition F.1.** *Assuming the following conditions are satisfied:*

- *Initialize $w(0) \in \mathbb{R}^m$ as specified in Definition D.2.*

- *Initialize $a \in \mathbb{R}^m$ as specified in Definition D.2.*

- *For any integer $t \geq 0$.*

- *Define training dataset $\mathcal{D} := \{(x_i, y_i)\}_{i=1}^{n} \subset \mathbb{R}^d \times \mathbb{R}$ as specified in Definition D.4.*

- *Let $\mathsf{S}_{i,r}(t) \in \mathbb{R}$ be defined according to Definition E.3.*

- *For $(i, j)$ in $[n] \times [n]$.*

*We define the kernel function as $H(t) \in \mathbb{R}^{n \times n}$, where its $(i, j)$-th entry is given by:*

$$H_{i,j}(t) := \frac{1}{m}x_{i,d}x_{j,d}\sum_{r=1}^{m}\left( \langle \mathsf{S}_{i,r}(t), x_i^{\circ 2} \rangle - \langle \mathsf{S}_{i,r}(t), x_i \rangle^2 \right) \cdot \left( \langle \mathsf{S}_{j,r}(t), x_j^{\circ 2} \rangle - \langle \mathsf{S}_{j,r}(t), x_j \rangle^2 \right)$$

## F.2 Assumption 2: NTK is PD

**Definition F.2** (Neural Tangent Kernel (NTK)). *Assuming the following conditions are satisfied:*

- *Initialize $w(0) \in \mathbb{R}^m$ as specified in Definition D.2.*

- *Initialize $a \in \mathbb{R}^m$ as specified in Definition D.2.*

- *For any integer $t \geq 0$.*

- *Define training dataset $\mathcal{D} := \{(x_i, y_i)\}_{i=1}^n \subset \mathbb{R}^d \times \mathbb{R}$ as specified in Definition D.4.*

- *Let $\mathsf{S}_{i,r}(t) \in \mathbb{R}$ be defined according to Definition E.3.*

- *For $(i, j)$ in $[n] \times [n]$.*

- *Let $H(t) \in \mathbb{R}^{n \times n}$ be defined in Definition F.1.*

*We define the kernel function as $H^* \in \mathbb{R}^{n \times n}$, where its $(i, j)$-th entry is given by:*

$$H_{i,j}^* := H_{i,j}(0)$$

$$= \frac{1}{m} x_{i,d} x_{j,d} \sum_{r=1}^m \left( \langle \mathsf{S}_{i,r}(0), x_i^{\circ 2} \rangle - \langle \mathsf{S}_{i,r}(0), x_i \rangle^2 \right) \cdot \left( \langle \mathsf{S}_{j,r}(0), x_j^{\circ 2} \rangle - \langle \mathsf{S}_{j,r}(0), x_j \rangle^2 \right)$$

**Assumption F.3.** *We assume that $H^*$ (defined in Definition F.2) is positive definite, with its smallest eigenvalue, denoted as $\lambda := \lambda_{\min}(H^)$, being greater than 0.*

## F.3 Kernel Convergence and PD Property during Training

**Lemma F.4** (Kernel Convergence, formal version of Lemma 5.1). *Assuming the following conditions are satisfied:*

- *Initialize $w(0) \in \mathbb{R}^m$ as specified in Definition D.2.*

- *Initialize $a \in \mathbb{R}^m$ as specified in Definition D.2.*

- *For any integer $t \geq 0$.*

- *Define training dataset $\mathcal{D} := \{(x_i, y_i)\}_{i=1}^n \subset \mathbb{R}^d \times \mathbb{R}$ as specified in Definition C.3.*

- *Define $\mathsf{S}_{i,r}(t) \in \mathbb{R}$ as specified in according to Definition E.3.*

- *For $(i, j)$ in $[n] \times [n]$.*

- *Define $H(t) \in \mathbb{R}^{n \times n}$ as specified in Definition F.1.*

- *Define $B$ specified in Definition K.1.*

- *Define $D$ specified in Definition K.2.*

- *We define $R := \max_{t \geq 0} \max_{r \in [m]} |w_r(t) - w_r(0)|$.*

- *Let $R \leq \frac{\lambda}{n \operatorname{poly}(\exp(B^2), \exp(D))}$.*

- *Let $\delta \in (0, 0.1)$.*

*Thus, with probability at least $1 - \delta$, the following holds:*

- *Part 1.*

$$\|H(t) - H^*\|_F \leq O(nR) \cdot \exp(O(B^2 D))$$

- *Part 2.*

$$\lambda_{\min}(H(t)) \geq \lambda/2$$

*Proof.* **Proof of Part 1.** Firstly, we have

$$\left| H_{i,j}(t) - H_{i,j}(0) \right|$$

$$= \left| \frac{1}{m} x_{i,d} x_{j,d} \sum_{r=1}^{m} \left( \langle \mathsf{S}_{i,r}(t), x_i^{\circ 2} \rangle - \langle \mathsf{S}_{i,r}(t), x_i \rangle^2 \right) \cdot \left( \langle \mathsf{S}_{j,r}(t), x_j^{\circ 2} \rangle - \langle \mathsf{S}_{j,r}(t), x_j \rangle^2 \right) \right.$$

$$\left. - \frac{1}{m} x_{i,d} x_{j,d} \sum_{r=1}^{m} \left( \langle \mathsf{S}_{i,r}(0), x_i^{\circ 2} \rangle - \langle \mathsf{S}_{i,r}(0), x_i \rangle^2 \right) \cdot \left( \langle \mathsf{S}_{j,r}(0), x_j^{\circ 2} \rangle - \langle \mathsf{S}_{j,r}(0), x_j \rangle^2 \right) \right|$$

$$\leq |x_{i,d} x_{j,d}| \max_{r \in [m]} \left( U_{1,i,j,r} + U_{2,i,j,r} + U_{3,i,j,r} + U_{4,i,j,r} \right) \tag{1}$$

Above the first equation is a consequence of Definition F.1 and Definition F.2, and the 2nd step can be obtained by applying the triangle inequality.

We define:

$$U_{1,i,j,r} := \left| \left( \langle \mathsf{S}_{i,r}(t), x_i^{\circ 2} \rangle - \langle \mathsf{S}_{i,r}(t), x_i \rangle^2 \right) \cdot \left( \langle \mathsf{S}_{j,r}(t), x_j^{\circ 2} \rangle - \langle \mathsf{S}_{j,r}(t), x_j \rangle^2 \right) \right.$$

$$\left. - \left( \langle \mathsf{S}_{i,r}(0), x_i^{\circ 2} \rangle - \langle \mathsf{S}_{i,r}(t), x_i \rangle^2 \right) \cdot \left( \langle \mathsf{S}_{j,r}(t), x_j^{\circ 2} \rangle - \langle \mathsf{S}_{j,r}(t), x_j \rangle^2 \right) \right|$$

$$U_{2,i,j,r} := \left| \left( \langle \mathsf{S}_{i,r}(0), x_i^{\circ 2} \rangle - \langle \mathsf{S}_{i,r}(t), x_i \rangle^2 \right) \cdot \left( \langle \mathsf{S}_{j,r}(t), x_j^{\circ 2} \rangle - \langle \mathsf{S}_{j,r}(t), x_j \rangle^2 \right) \right.$$

$$\left. - \left( \langle \mathsf{S}_{i,r}(0), x_i^{\circ 2} \rangle - \langle \mathsf{S}_{i,r}(0), x_i \rangle^2 \right) \cdot \left( \langle \mathsf{S}_{j,r}(t), x_j^{\circ 2} \rangle - \langle \mathsf{S}_{j,r}(t), x_j \rangle^2 \right) \right|$$

$$U_{3,i,j,r} := \left| \left( \langle \mathsf{S}_{i,r}(0), x_i^{\circ 2} \rangle - \langle \mathsf{S}_{i,r}(0), x_i \rangle^2 \right) \cdot \left( \langle \mathsf{S}_{j,r}(t), x_j^{\circ 2} \rangle - \langle \mathsf{S}_{j,r}(t), x_j \rangle^2 \right) \right.$$

$$\left. - \left( \langle \mathsf{S}_{i,r}(0), x_i^{\circ 2} \rangle - \langle \mathsf{S}_{i,r}(0), x_i \rangle^2 \right) \cdot \left( \langle \mathsf{S}_{j,r}(0), x_j^{\circ 2} \rangle - \langle \mathsf{S}_{j,r}(t), x_j \rangle^2 \right) \right|$$

$$U_{4,i,j,r} := \left| \left( \langle \mathsf{S}_{i,r}(0), x_i^{\circ 2} \rangle - \langle \mathsf{S}_{i,r}(0), x_i \rangle^2 \right) \cdot \left( \langle \mathsf{S}_{j,r}(0), x_j^{\circ 2} \rangle - \langle \mathsf{S}_{j,r}(t), x_j \rangle^2 \right) \right.$$

$$\left. - \left( \langle \mathsf{S}_{i,r}(0), x_i^{\circ 2} \rangle - \langle \mathsf{S}_{i,r}(0), x_i \rangle^2 \right) \cdot \left( \langle \mathsf{S}_{j,r}(0), x_j^{\circ 2} \rangle - \langle \mathsf{S}_{j,r}(0), x_j \rangle^2 \right) \right|$$

Before we bound all terms, we first provide some tools:

$$\|x_i\|_2 \leq \sqrt{d} \cdot O(B) \tag{2}$$

where this step uses $\ell_2$ norm and can be trivially obtain from Part 1 of Lemma K.3.

We can show

$$\|S_{i,r}(t) - S_{i,r}(0)\|_2 \leq \exp(O(B^2 D)) \cdot O(RB^2)/\sqrt{d} \tag{3}$$

where this step uses the definition of $\ell_2$ norm and is a consequence of Part 1,3 of Lemma K.3.

Next, we can get

$$\|x_i^{\circ 2}\|_2 = \sqrt{\sum_{k \in [d]} x_{i,k}^4}$$

$$\leq \sqrt{d} \cdot O(B^2) \tag{4}$$

where the first is based on $\ell_2$ norm, while the 2nd step is based on Lemma K.3Part 1.

We proceed to show

$$|\langle S_{i,r}(t), x_i^{\circ 2} \rangle| \leq \|S_{i,r}(t)\|_2 \cdot \|x_i^{\circ 2}\|_2$$

$$\leq \sqrt{d} \cdot \frac{\exp(O(B^2(D+R)))}{d} \cdot \sqrt{d} \cdot O(B^2)$$

$$= \exp(O(B^2(D+R))) \cdot O(B^2) \tag{5}$$

Above, the first equation uses Cacuchy-Schwarz inequality. The second step combines the result of Eq. (4), Part 9 of Lemma K.3 and definition of $\ell_2$ norm. The final step applies basic algebra.

In the same way, we have

$$
\begin{aligned}
|\langle S_{i,r}(0), x_i^{\circ 2}\rangle| &\le \|S_{i,r}(0)\|_2 \cdot \|x_i^{\circ 2}\|_2 \\
&\le \sqrt{d} \cdot \frac{\exp(O(B^2 D))}{d} \cdot \sqrt{d} \cdot O(B^2) \\
&= \exp(O(B^2 D)) \cdot O(B^2)
\end{aligned}
\tag{6}
$$

where the first step is a result of Cauchy-Schwarz inequality, the second step is derived from Eq. (4), Part 8 of Lemma K.3 and definition of $\ell_2$ norm, and the last equation applies basic algebraic manipulation.

Furthermore, we can have

$$
\begin{aligned}
|\langle S_{i,r}(t), x_i\rangle| &\le \|S_{i,j}(t)\|_2 \cdot \|x_i\|_2 \\
&\le \sqrt{d} \cdot \frac{\exp(O(B^2(D+R)))}{d} \cdot \sqrt{d} \cdot O(B) \\
&\le \exp(O(B^2(D+R))) \cdot O(B)
\end{aligned}
\tag{7}
$$

where the first step is a result of Cauchy inequality, the second step is derived from Eq. (2), Part 9 of Lemma K.3 and definition of $\ell_2$ norm, and the final equation comes from basic algebraic manipulation.

In the same way, we can have

$$
\begin{aligned}
|\langle S_{i,r}(0), x_i\rangle| &\le \|S_{i,j}(0)\|_2 \cdot \|x_i\|_2 \\
&\le \sqrt{d} \cdot \frac{\exp(O(B^2 D))}{d} \cdot \sqrt{d} \cdot O(B) \\
&\le \exp(O(B^2 D)) \cdot O(B)
\end{aligned}
\tag{8}
$$

Above the first equation is a result of Cauchy inequality, the second step is based on Eq. (2), Part 8 of Lemma K.3 and definition of $\ell_2$ norm, and the last step comes from basic algebraic manipulation.

Then we are able to bound $U_{1,i,j,r}, U_{2,i,j,r}, U_{3,i,j,r}$ and $U_{4,i,j,r}$.

To bound $U_{1,i,j,r}$, we have

$$
\begin{aligned}
U_{1,i,j,r} &= \left| \langle S_{i,r}(t) - S_{i,r}(0), x_i^{\circ 2}\rangle \cdot \left( \langle S_{j,r}(t), x_j^{\circ 2}\rangle - \langle S_{j,r}(t), x_j\rangle^2 \right) \right| \\
&\le |\langle S_{i,r}(t) - S_{i,r}(0), x_i^{\circ 2}\rangle| \cdot |\langle S_{j,r}(t), x_j^{\circ 2}\rangle - \langle S_{j,r}(t), x_j\rangle^2| \\
&\le \|S_{i,r}(t) - S_{i,r}(0)\|_2 \cdot \|x_i^{\circ 2}\|_2 \cdot |\langle S_{j,r}(t), x_j^{\circ 2}\rangle - \langle S_{j,r}(t), x_j\rangle^2| \\
&\le \|S_{i,r}(t) - S_{i,r}(0)\|_2 \cdot \|x_i^{\circ 2}\|_2 \cdot (|\langle S_{j,r}(t), x_j^{\circ 2}\rangle| + |\langle S_{j,r}(t), x_j\rangle^2|) \\
&\le \frac{1}{\sqrt{d}} \cdot \exp(O(B^2 D)) \cdot O(RB^2) \cdot \sqrt{d} \cdot O(B^2) \cdot \exp(O(B^2(D+R))) \cdot O(B^2) \\
&= \exp(O(B^2(D+R))) \cdot O(RB^6)
\end{aligned}
\tag{9}
$$

where the first and second steps are based on basic algebraic manipulations, the third step is a consequence of the Cauchy inequality, the fourth step can be trivially obtained by applying the triangle inequality, the 5th step is a consequence of Eq. (3), (4), (5) and (7), and the final step results from basic algebraic manipulation.

To bound $U_{2,i,j,r}$, we have

$$
\begin{aligned}
U_{2,i,j,r} &= |\langle S_{i,r}(0), x_i\rangle^2 - \langle S_{i,r}(t), x_i\rangle^2| \cdot |\langle S_{j,r}(t), x_j^{\circ 2}\rangle - \langle S_{j,r}(t), x_j\rangle^2| \\
&= |\langle S_{i,r}(0) - S_{i,r}(t), x_i\rangle| \cdot |\langle S_{i,r}(0) + S_{i,r}(t), x_i\rangle| \cdot |\langle S_{j,r}(t), x_j^{\circ 2}\rangle - \langle S_{j,r}(t), x_j\rangle^2| \\
&\le \|S_{i,r}(0) - S_{i,r}(t)\|_2 \cdot \|x_i\|_2 \cdot (|\langle S_{i,r}(0), x_i\rangle| + |\langle S_{i,r}(t), x_i\rangle|) \\
&\quad \cdot |\langle S_{j,r}(t), x_j^{\circ 2}\rangle - \langle S_{j,r}(t), x_j\rangle^2|
\end{aligned}
$$

$$
\begin{aligned}
&\leq \|S_{i,r}(0) - S_{i,r}(t)\|_2 \cdot \|x_i\|_2 \cdot (|\langle S_{i,r}(0), x_i\rangle| + |\langle S_{i,r}(t), x_i\rangle|) \\
&\quad \cdot (|\langle S_{j,r}(t), x_j^{\circ 2}\rangle| + |\langle S_{j,r}(t), x_j\rangle^2|) \\
&\leq \frac{1}{\sqrt{d}} \cdot \exp(O(B^2 D)) \cdot O(RB^2) \cdot \sqrt{d} \cdot O(B) \cdot \frac{\exp(O(B^2(D+R)))}{\sqrt{d}} \\
&\quad \cdot \sqrt{d} \cdot O(B) \cdot \exp(O(B^2(D+R))) \cdot O(B^2) \\
&= \exp(O(B^2(D+R))) \cdot O(RB^6)
\end{aligned}
\tag{10}
$$

where the first and second step are based on basic algebraic manipulations, the 3rd step is a consequence of the Cauchy inequality and triangle inequality, the 4th step is due to triangle inequality, the fifth step follows from Eq. (2), (3), (5), (7) and (8), and the last step results from basic algebraic manipulations.

To bound $U_{3,i,j,r}$, we have

$$
\begin{aligned}
U_{3,i,j,r} &= |\langle S_{i,r}(0), x_i^{\circ 2}\rangle - \langle S_{i,r}(0), x_i\rangle^2| \cdot |\langle S_{j,r}(t) - S_{j,r}(0), x_j^{\circ 2}\rangle| \\
&\leq (|\langle S_{i,r}(0), x_i^{\circ 2}\rangle| + |\langle S_{i,r}(0), x_i\rangle^2|) \cdot |\langle S_{j,r}(t) - S_{j,r}(0), x_j^{\circ 2}\rangle| \\
&\leq (|\langle S_{i,r}(0), x_i^{\circ 2}\rangle| + |\langle S_{i,r}(0), x_i\rangle^2|) \cdot \|S_{j,r}(t) - S_{j,r}(0)\|_2 \cdot \|x_j^{\circ 2}\|_2 \\
&\leq \exp(O(B^2 D)) \cdot O(B^2) \cdot \exp(O(B^2 D)) \cdot O(RB^2) \cdot \frac{1}{\sqrt{d}} \cdot \sqrt{d} \cdot O(B^2) \\
&= \exp(O(B^2 D)) \cdot O(RB^6)
\end{aligned}
\tag{11}
$$

where the first two steps are based on basic algebraic manipulations, the third step is a consequence of triangle inequality, and the 4th step can be obtained by applying Cauchy inequality, the 5th step is a consequence of Eq. (3), (4), (6) and (8), and the final step results from basic algebraic manipulation.

To bound $U_{4,i,j,r}$, we have

$$
\begin{aligned}
U_{4,i,j,r} &= |\langle S_{i,r}(0), x_i^{\circ 2}\rangle - \langle S_{i,r}(0), x_i\rangle^2| \cdot |\langle S_{j,r}(0), x_j\rangle^2 - \langle S_{j,r}(t), x_j\rangle^2| \\
&= |\langle S_{i,r}(0), x_i^{\circ 2}\rangle - \langle S_{i,r}(0), x_i\rangle^2| \cdot |\langle S_{j,r}(0) - S_{j,r}(t), x_j\rangle| \cdot |\langle S_{j,r}(0) + S_{j,r}(t), x_j\rangle| \\
&\leq (|\langle S_{i,r}(0), x_i^{\circ 2}\rangle| + |\langle S_{i,r}(0), x_i\rangle^2|) \cdot |\langle S_{j,r}(0) - S_{j,r}(t), x_j\rangle| \cdot |\langle S_{j,r}(0) + S_{j,r}(t), x_j\rangle| \\
&\leq (|\langle S_{i,r}(0), x_i^{\circ 2}\rangle| + |\langle S_{i,r}(0), x_i\rangle^2|) \cdot \|S_{j,r}(0) - S_{j,r}(t)\|_2 \cdot \|x_j\|_2 \\
&\quad \cdot (|\langle S_{j,r}(0), x_j\rangle| + |\langle S_{j,r}(t), x_j\rangle|) \\
&\leq \exp(O(B^2 D)) \cdot O(B^2) \cdot \frac{1}{\sqrt{d}} \cdot \exp(O(B^2 D)) \cdot O(RB^2) \\
&\quad \cdot \sqrt{d} \cdot O(B) \cdot exp(O(B^2(D+R))) \cdot O(B) \\
&= \exp(O(B^2(D+R))) \cdot O(RB^6)
\end{aligned}
\tag{12}
$$

where the first and second steps are the result of basic algebraic manipulations, the third step follows from triangle inequality, the fourth step is derived from Cauchy-Schwarz inequality nad triangle inequality, the fifth step is due to Eq. (2), (3), (6), (8) and (7) and the last step follows from basic algebraic manipulationsic manipulations.

Then, we can have

$$
\begin{aligned}
|H_{i,j}(t) - H_{i,j}(0)| &\leq |x_{i,d} x_{j,d}| \max_{r \in [m]} (U_{1,i,j,r} + U_{2,i,j,r} + U_{2,i,j,r} + U_{3,i,j,r}) \\
&\leq O(B^2) \cdot \exp(O(B^2(D+R))) \cdot O(RB^6) \\
&= O(RB^8) \cdot \exp(O(B^2(D+R))) \\
&\leq O(R) \cdot \exp(O(B^2 D))
\end{aligned}
\tag{13}
$$

where the 1st step is derived from Eq. (1), the 2nd step combines the result of Eq. (9), (10), (11) and (12), the third step is based on basic algebraic manipulations, the final step can be obtained from $R \in (0, 0.01)$, $B \geq 1$ and then $O(\text{poly}(B)) \leq \exp(O(B))$.

Finally, with probability $1 - \delta$,

$$
\|H(t) - H(0)\|_F \leq O(nR) \cdot \exp(O(B^2 D))
$$

this step results from Eq. (13) and the definition of Frobenius norm.

**Proof of Part 2.** We have

$$\|H(t) - H(0)\|_F \leq O(nR) \cdot \exp(O(B^2 D))$$
$$\leq \lambda/2 \tag{14}$$

Above, the first inequality can be derived from Part 1 of this lemma, and the second inequality is a consequence of the choice value of $R$.

Then, we can have

$$\lambda_{\min}(H(t)) \geq \lambda_{\min}(H^*) - \|H(t) - H^*\|_F$$
$$\geq \lambda_{\min}(H^*) - \lambda/2$$
$$= \lambda/2$$

Above the first inequality is a consequence of Fact B.7. The second inequality can be trivially obtained from Eq. (14) and the final equation is based on $\lambda_{\min}(H^*) = \lambda$. □

# G Training Dynamic

## G.1 Decomposing Loss

**Lemma G.1.** *Assuming the following conditions are satisfied:*

- *Let $i \in [n]$, $r \in [m]$ and $k \in [d]$.*

- *Let integer $t > 0$.*

- *Let training dataset $\mathcal{D} := \{(x_i, y_i)\}_{i=1}^n \subset \mathbb{R}^d \times \mathbb{R}$ be specified as Definition C.3.*

- *Initialize $w(0) \in \mathbb{R}^m$ as specified in Definition D.2.*

- *Initialize $a \in \mathbb{R}^m$ as specified in Definition D.2.*

- *Define $L(t) \in \mathbb{R}$ as specified in Definition D.4.*

- *Define $\eta > 0$ as specified in Definition D.5.*

- *Define $\Delta w_r(t) \in \mathbb{R}$ as specified in Definition D.5.*

- *Define $\mathsf{u}_{i,r}(t) \in \mathbb{R}^d$ as specified in Definition E.1.*

- *Define $\alpha_{i,r}(t) \in \mathbb{R}$ as specified in Definition E.2.*

- *Define $\mathsf{S}_{i,r}(t) \in \mathbb{R}^d$ as specified in Definition E.3.*

- *Let $\mathsf{F}_i(t) \in \mathbb{R}$ be defined as Definition E.4.*

- *Define*

$$C_1 := -\eta \frac{1}{\sqrt{m}} \sum_{i=1}^n (\mathsf{F}_i(t) - y_i) \cdot \sum_{r=1}^m a_r \cdot \left( \langle \mathsf{S}_{i,r}(t), (x_{i,d}\Delta w_r(t)) \cdot x_i^{\circ 2} \rangle \right.$$
$$\left. + \langle \mathsf{S}_{i,r}(t), x_i \rangle^2 \cdot (x_{i,d}\Delta w_r(t)) \right)$$

- *Define*

$$C_2 := -\eta^2 \Theta(1) \frac{1}{\sqrt{m}} \sum_{i=1}^n (\mathsf{F}_i(t) - y_i) \cdot \sum_{r=1}^m a_r \cdot \langle \mathsf{S}_{i,r}(t), (x_{i,d}\Delta w_r(t))^2 \cdot x_i^{\circ 3} \rangle$$

- *Define*

$$C_3 := -\eta^2 \Theta(1) \frac{1}{\sqrt{m}} \sum_{i=1}^{n} (\mathsf{F}_i(t) - y_i) \cdot \sum_{r=1}^{m} a_r \cdot \langle \mathsf{S}_{i,r}(t), (x_{i,d} \Delta w_r(t))^2 \cdot x_i^{\circ 2} \rangle \cdot \langle \mathsf{S}_{i,r}(t), x_i \rangle$$

- *Define*

$$C_4 := -\frac{1}{\sqrt{m}} \sum_{i=1}^{n} (\mathsf{F}_i(t) - y_i) \cdot \sum_{r=1}^{m} a_r \cdot \langle \mathsf{S}_{i,r}(t), \beta_{i,r}(t) \rangle \cdot \langle \mathsf{S}_{i,r}(t+1) - \mathsf{S}_{i,r}(t), x_i \rangle$$

- *Define*

$$C_5 := \frac{1}{2} \| \mathsf{F}(t) - \mathsf{F}(t+1) \|_2^2$$

*Then, we can obtain the following:*

$$L(t+1) = L(t) + C_1 + C_2 + C_3 + C_4 + C_5$$

*Proof.* First, we denote that:

$$\beta_{i,r}(t) := x_{i,d} \cdot \eta \Delta w_r(t) \cdot x_i + \Theta(1) \cdot (x_{i,d} \cdot \eta \Delta w_r(t) \cdot x_i)^{\circ 2}$$

We have:

$$
\begin{aligned}
\mathsf{u}_{i,r}(t) - \mathsf{u}_{i,r}(t+1) &= \mathsf{u}_{i,r}(t) - \exp\left( x_{i,d} \cdot w_r(t+1) \cdot x_i \right) \\
&= \mathsf{u}_{i,r}(t) - \exp\left( x_{i,d} \cdot (w_r(t) - \eta \Delta w_r(t)) \cdot x_i \right) \\
&= \mathsf{u}_{i,r}(t) - \exp\left( x_{i,d} \cdot w_r(t) \cdot x_i \right) \circ \exp\left( - x_{i,d} \cdot \eta \Delta w_r(t) \cdot x_i \right) \\
&= \mathsf{u}_{i,r}(t) - \mathsf{u}_{i,r}(t) \circ \exp\left( - x_{i,d} \cdot \eta \Delta w_r(t) \cdot x_i \right) \\
&= \mathsf{u}_{i,r}(t) \circ \left( x_{i,d} \cdot \eta \Delta w_r(t) \cdot x_i + \Theta(1) \cdot (x_{i,d} \cdot \eta \Delta w_r(t) \cdot x_i)^{\circ 2} \right) \\
&= \mathsf{u}_{i,r}(t) \circ \beta_{i,r}(t)
\end{aligned}
\tag{15}
$$

Above, the first equation is derived from Definition E.1. Then the second equation follows from Definition D.5 and the third equation is a result of basic algebraic manipulations, the fourth step is because of Definition E.1, and the final step is based on the definition of $\beta_{i,r}(t)$.

Next, we have:

$$
\begin{aligned}
\alpha_{i,r}(t) - \alpha_{i,r}(t+1) &= \langle \mathsf{u}_{i,r}(t), \mathbf{1}_d \rangle - \langle \mathsf{u}_{i,r}(t+1), \mathbf{1}_d \rangle \\
&= \langle \mathsf{u}_{i,r}(t) - \mathsf{u}_{i,r}(t+1), \mathbf{1}_d \rangle \\
&= \left\langle \mathsf{u}_{i,r}(t) \circ \beta_{i,r}(t), \mathbf{1}_d \right\rangle \\
&= \left\langle \mathsf{u}_{i,r}(t), \beta_{i,r}(t) \right\rangle
\end{aligned}
\tag{16}
$$

Above, the first equation is derived from Definition E.2. And the second step follows from basic algebraic manipulations, the third step is a consequence of Eq. (15), the last step is due to basic algebraic manipulations.

We obtain:

$$
\begin{aligned}
\mathsf{S}_{i,r}(t) - \mathsf{S}_{i,r}(t+1) &= \alpha_{i,r}(t)^{-1} \mathsf{u}_{i,r}(t) - \alpha_{i,r}(t+1)^{-1} \mathsf{u}_{i,r}(t+1) \\
&= \alpha_{i,r}(t)^{-1} (\mathsf{u}_{i,r}(t) - \mathsf{u}_{i,r}(t+1)) + (\alpha_{i,r}(t)^{-1} - \alpha_{i,r}(t+1)^{-1}) \mathsf{u}_{i,r}(t+1) \\
&= \alpha_{i,r}(t)^{-1} (\mathsf{u}_{i,r}(t) - \mathsf{u}_{i,r}(t+1)) + \alpha_{i,r}(t)^{-1} (\alpha_{i,r}(t) - \alpha_{i,r}(t+1)) \mathsf{S}_{i,r}(t+1) \\
&= \mathsf{S}_{i,r}(t) \circ \beta_{i,r}(t) + \alpha_{i,r}(t)^{-1} (\alpha_{i,r}(t) - \alpha_{i,r}(t+1)) \mathsf{S}_{i,r}(t+1)
\end{aligned}
$$

$$= \mathsf{S}_{i,r}(t) \circ \beta_{i,r}(t) + \left\langle \mathsf{S}_{i,r}(t), \beta_{i,r}(t) \right\rangle \cdot \mathsf{S}_{i,r}(t+1) \tag{17}$$

where the first step is based on Definition E.3, the second step follows from basic algebraic manipulations, the third step comes from Definition E.3, the fourth step is derived from Eq. (15), the fifth step follows from Eq. (16).

Hence, we get:

$$\mathsf{F}_i(t) - \mathsf{F}_i(t+1) = \frac{1}{\sqrt{m}} \sum_{r=1}^{m} a_r \cdot \langle \mathsf{S}_{i,r}(t) - \mathsf{S}_{i,r}(t+1), x \rangle$$

$$= \frac{1}{\sqrt{m}} \sum_{r=1}^{m} a_r \cdot \langle \mathsf{S}_{i,r}(t) \circ \beta_{i,r}(t) + \langle \mathsf{S}_{i,r}(t), \beta_{i,r}(t) \rangle \cdot \mathsf{S}_{i,r}(t+1), x_i \rangle$$

$$= \frac{1}{\sqrt{m}} \sum_{r=1}^{m} a_r \cdot \left( \langle \mathsf{S}_{i,r}(t), \beta_{i,r}(t) \circ x_i \rangle + \langle \mathsf{S}_{i,r}(t), \beta_{i,r}(t) \rangle \cdot \langle \mathsf{S}_{i,r}(t+1), x_i \rangle \right)$$

$$= \frac{1}{\sqrt{m}} \sum_{r=1}^{m} a_r \cdot \left( \langle \mathsf{S}_{i,r}(t), \beta_{i,r}(t) \circ x_i \rangle + \langle \mathsf{S}_{i,r}(t), \beta_{i,r}(t) \rangle \cdot \langle \mathsf{S}_{i,r}(t), x_i \rangle \right.$$

$$\left. + \langle \mathsf{S}_{i,r}(t), \beta_{i,r}(t) \rangle \cdot \langle \mathsf{S}_{i,r}(t+1) - \mathsf{S}_{i,r}(t), x_i \rangle \right)$$

$$= v_{1,i} + v_{2,i} + v_{3,i} + v_{4,i}$$

where the first step is derived from Definition E.4, the second step is a consequence of Eq. (17), the third and fourth step follow from basic algebraic manipulations, and the last step follows from defining:

$$v_{1,i} := \eta \frac{1}{\sqrt{m}} \sum_{r=1}^{m} a_r \cdot \left( \langle \mathsf{S}_{i,r}(t), (x_{i,d} \Delta w_r(t)) \cdot x_i^{\circ 2} \rangle + \langle \mathsf{S}_{i,r}(t), x_i \rangle^2 \cdot (x_{i,d} \Delta w_r(t)) \right)$$

$$v_{2,i} := \eta^2 \Theta(1) \frac{1}{\sqrt{m}} \sum_{r=1}^{m} a_r \cdot \langle \mathsf{S}_{i,r}(t), (x_{i,d} \Delta w_r(t))^2 \cdot x_i^{\circ 3} \rangle$$

$$v_{3,i} := \eta^2 \Theta(1) \frac{1}{\sqrt{m}} \sum_{r=1}^{m} a_r \cdot \langle \mathsf{S}_{i,r}(t), (x_{i,d} \Delta w_r(t))^2 \cdot x_i^{\circ 2} \rangle \cdot \langle \mathsf{S}_{i,r}(t), x_i \rangle$$

$$v_{4,i} := \frac{1}{\sqrt{m}} \sum_{r=1}^{m} a_r \cdot \langle \mathsf{S}_{i,r}(t), \beta_{i,r}(t) \rangle \cdot \langle \mathsf{S}_{i,r}(t+1) - \mathsf{S}_{i,r}(t), x_i \rangle$$

Finally, we can show that:

$$L(t+1) = \frac{1}{2} \sum_{i=1}^{n} (\mathsf{F}_i(t+1) - y_i)^2$$

$$= \frac{1}{2} \|\mathsf{F}(t+1) - y\|_2^2$$

$$= \frac{1}{2} \|\mathsf{F}(t+1) - \mathsf{F}(t) + \mathsf{F}(t) - y\|_2^2$$

$$= \frac{1}{2} \|\mathsf{F}(t) - y\|_2^2 - \langle \mathsf{F}(t) - \mathsf{F}(t+1), \mathsf{F}(t) - y \rangle + \frac{1}{2} \|\mathsf{F}(t) - \mathsf{F}(t+1)\|_2^2$$

$$= L(t) + C_1 + C_2 + C_3 + C_4 + C_5$$

Above, the first equation is based on Definition D.4, the second, third, and fourth steps are the result of basic algebraic manipulations, and the last step is due to the statement of the lemma and defining:

$$C_1 := \langle v_1, y - \mathsf{F}(t) \rangle$$
$$C_2 := \langle v_2, y - \mathsf{F}(t) \rangle$$
$$C_3 := \langle v_3, y - \mathsf{F}(t) \rangle$$

$$C_4 := \langle v_4, y - \mathsf{F}(t) \rangle$$

$$C_5 := \frac{1}{2} \|\mathsf{F}(t) - \mathsf{F}(t+1)\|_2^2$$

$\square$

## G.2 Bounding $C_1$

**Lemma G.2.** *Assuming the following conditions are satisfied:*

- *Let $i \in [n]$, $r \in [m]$ and $k \in [d]$.*
- *Let integer $t > 0$.*
- *Let training dataset $\mathcal{D} := \{(x_i, y_i)\}_{i=1}^n \subset \mathbb{R}^d \times \mathbb{R}$ be specified as Definition C.3.*
- *Initialize $w(0) \in \mathbb{R}^m$ as specified in Definition D.2.*
- *Initialize $a \in \mathbb{R}^m$ as specified in Definition D.2.*
- *Define $L(t) \in \mathbb{R}$ as specified in Definition D.4.*
- *Define $\eta > 0$ as specified in Definition D.5.*
- *Define $\Delta w_r(t) \in \mathbb{R}$ as specified in Definition D.5.*
- *Define $\mathsf{u}_{i,r}(t) \in \mathbb{R}^d$ as specified in Definition E.1.*
- *Define $\alpha_{i,r}(t) \in \mathbb{R}$ as specified in Definition E.2.*
- *Define $\mathsf{S}_{i,r}(t) \in \mathbb{R}^d$ as specified in Definition E.3*
- *Define $\mathsf{F}_i(t) \in \mathbb{R}$ as specified in Definition E.4.*
- *Following Lemma G.1 to define*

$$C_1 := -\eta \frac{1}{\sqrt{m}} \sum_{i=1}^n (\mathsf{F}_i(t) - y_i) \cdot \sum_{r=1}^m a_r \cdot \Big( \langle \mathsf{S}_{i,r}(t), (x_{i,d} \Delta w_r(t)) \cdot x_i^{\circ 2} \rangle$$
$$+ \langle \mathsf{S}_{i,r}(t), x_i \rangle^2 \cdot (x_{i,d} \Delta w_r(t)) \Big)$$

*Then we have:*

$$C_1 \leq -\eta \lambda \cdot L(t)$$

*Proof.* We have:

$$C_1 = -\eta \frac{1}{\sqrt{m}} \sum_{i=1}^n (\mathsf{F}_i(t) - y_i) \cdot \sum_{r=1}^m a_r \cdot \Big( \langle \mathsf{S}_{i,r}(t), (x_{i,d} \Delta w_r(t)) \cdot x_i^{\circ 2} \rangle + \langle \mathsf{S}_{i,r}(t), x_i \rangle^2 \cdot (x_{i,d} \Delta w_r(t)) \Big)$$

Then by plugging:

$$\Delta w_r(t) = \frac{1}{\sqrt{m}} a_r \sum_{i=1}^n (\mathsf{F}_i(t) - y_i) \cdot x_{i,d} \cdot (\langle \mathsf{S}_{i,r}(t), x_i^{\circ 2} \rangle - \langle \mathsf{S}_{i,r}(t), x_i \rangle^2)$$

We can show that:

$$C_1 = -\eta \frac{1}{m} \sum_{i=1}^n (\mathsf{F}_i(t) - y_i) \cdot \sum_{j=1}^n (\mathsf{F}_j(t) - y_j) \cdot$$

$$x_{i,d}x_{j,d}\sum_{r=1}^{m}\left(\langle \mathsf{S}_{i,r}(t),x_i^{\circ 2}\rangle + \langle \mathsf{S}_{i,r}(t),x_i\rangle^2\right)\cdot\left(\langle \mathsf{S}_{j,r}(t),x_j^{\circ 2}\rangle + \langle \mathsf{S}_{j,r}(t),x_j\rangle^2\right)$$

$$= -\eta(\mathsf{F}(t)-y)^\top H(t)(\mathsf{F}(t)-y)$$
$$\leq -\eta\lambda/2\cdot\|\mathsf{F}(t)-y\|_2^2$$
$$= -\eta\lambda\cdot L(t)$$

where the first step is the definition of $C_1$, and the second step is derived from Definition F.1, the third step can be obtained from Part 2 of Lemma F.4, the last step is due to Definition D.4. □

## G.3 Bounding $C_2$

**Lemma G.3.** *Assuming the following conditions are satisfied:*

- *Let $i \in [n]$, $r \in [m]$ and $k \in [d]$.*
- *Let integer $t > 0$.*
- *Define training dataset $\mathcal{D} := \{(x_i, y_i)\}_{i=1}^n \subset \mathbb{R}^d \times \mathbb{R}$ as specified in Definition C.3.*
- *Initialize $w(0) \in \mathbb{R}^m$ as specified in Definition D.2.*
- *Initialize $a \in \mathbb{R}^m$ as specified in Definition D.2.*
- *Define $L(t) \in \mathbb{R}$ as specified in Definition D.4.*
- *Define $\Delta w_r(t) \in \mathbb{R}$ as specified in Definition D.5.*
- *Define $\mathsf{S}_{i,r}(t)$ as specified in Definition E.3.*
- *Define $\mathsf{F}_i(t)$ as specified in Definition E.4.*
- *Let learning rate $\eta < 1$.*
- *Define $B > 1$ as specified in Definition K.1.*
- *Define $D > 1$ as specified in Definition K.2.*
- *Let $R \in (0, 0.01/B^2)$.*
- *Let $\delta \in (0, 0.1)$.*
- *Let $m \geq \Omega(\lambda^{-2}n^7 d \cdot \exp(O(B^2 D)))$.*
- *Following Lemma G.1 to define*

$$C_2 := -\eta^2\Theta(1)\frac{1}{\sqrt{m}}\sum_{i=1}^{n}(\mathsf{F}_i(t)-y_i)\cdot\sum_{r=1}^{m}a_r\cdot\langle \mathsf{S}_{i,r}(t),(x_{i,d}\Delta w_r(t))^2\cdot x_i^{\circ 3}\rangle$$

*Consequently, with probability at least $1 - \delta$:*

$$C_2 \leq \frac{1}{8}\eta\lambda\cdot L(t)$$

*Proof.* Firstly, we have

$$|\langle \mathsf{S}_{i,r}(t),(x_{i,d}\Delta w_r(t))^2\cdot x_i^{\circ 3}\rangle| \leq \|S_{i,r}(t)\|_2\cdot\|(x_{i,d}\Delta w_r(t))^2\cdot x_i^{\circ 3}\|_2$$
$$\leq \sqrt{d}\cdot\frac{\exp(O(B^2 D))}{d}\cdot O(B^2)\cdot\sqrt{d}\cdot O(B^3)$$
$$\cdot\frac{n^3}{m}\cdot\exp(O(B^2 D))\cdot\|\mathsf{F}(t)-y\|_2^2$$

$$= \frac{n^3}{m} \exp(O(B^2 D)) \cdot \|\mathsf{F}(t) - y\|_2^2 \tag{18}$$

where the 1st step is a consequence of Cauchy inequality, the 2nd step is based on Part 1, 9 of Lemma K.3, Lemma H.3 and the definition of $\ell_2$ norm, and the final step is derived from basic algebraic manipulations and $O(B) \leq \exp(O(B^2))$.

And we proceed to bound $\|\mathsf{F}(t) - y\|_2$, we have

$$\|\mathsf{F}(t) - y\|_2 = \sqrt{2L(t)}$$
$$\leq \sqrt{\exp\left(O(B^2 D)\right) \cdot O(nR^2) + O(nB^2)}$$
$$\leq \sqrt{\exp\left(O(B^2 D)\right) \cdot O(nR^2)} + \sqrt{O(nB^2)}$$
$$= \exp(O(B^2 D)) \cdot O(\sqrt{n}R) + O(\sqrt{n}B) \tag{19}$$

where the first step is based on Definition D.4, the second step follows from Lemma H.2, and the third step and fourth step result from basic algebraic manipulations.

Now, we can show that

$$\Big| \sum_{i=1}^n v_{2,i} \cdot (\mathsf{F}_i(t) - y_i) \Big| = \Big| \sum_{i=1}^n \eta^2 \Theta(1) \frac{1}{\sqrt{m}} \sum_{r=1}^m a_r \cdot \langle \mathsf{S}_{i,r}(t), (x_{i,d} \Delta w_r(t))^2 \cdot x_i^{\circ 3} \rangle \cdot (\mathsf{F}_i(t) - y_i) \Big|$$

$$= \frac{\eta^2}{\sqrt{m}} \cdot \Big| \sum_{i=1}^n \sum_{r=1}^m a_r \cdot \langle \mathsf{S}_{i,r}(t), (x_{i,d} \Delta w_r(t))^2 \cdot x_i^{\circ 3} \rangle \cdot (\mathsf{F}_i(t) - y_i) \Big|$$

$$\leq \frac{\eta^2}{\sqrt{m}} \cdot \Big| \sum_{r=1}^m a_r \max_{i \in [n]} \langle \mathsf{S}_{i,r}(t), (x_{i,d} \Delta w_r(t))^2 \cdot x_i^{\circ 3} \rangle \Big| \cdot \|\mathsf{F}(t) - y\|_1$$

$$\leq \frac{\eta^2 \sqrt{d}}{\sqrt{m}} \cdot \Big| \sum_{r=1}^m a_r \max_{i \in [n]} \langle \mathsf{S}_{i,r}(t), (x_{i,d} \Delta w_r(t))^2 \cdot x_i^{\circ 3} \rangle \Big| \cdot \|\mathsf{F}(t) - y\|_2$$

$$\leq \frac{\eta^2 \sqrt{d}}{\sqrt{m}} \cdot \Big( \exp(O(B^2 D)) \cdot O(\sqrt{n}R) + O(\sqrt{n}B) \Big)$$

$$\cdot \Big| \sum_{r=1}^m a_r \max_{i \in [n]} \langle \mathsf{S}_{i,r}(t), (x_{i,d} \Delta w_r(t))^2 \cdot x_i^{\circ 3} \rangle \Big| \tag{20}$$

where the first step derived from definition of $v_{2,i}$ in Lemma G.1, the second step results from basic algebraic manipulations, the third step comes from definition of $\ell_1$ norm along with basic algebraic manipulations, the fourth step leverages the inequality $\|x\|_1 \leq \sqrt{d}\|x\|_2$, and the final step is due to Eq. (19).

We can then use Hoeffding's inequality (Lemma B.1) for the random variable

$$a_r \max_{i \in [n]} \langle \mathsf{S}_{i,r}(t), (x_{i,d} \Delta w_r(t))^2 \cdot x_i^{\circ 3} \rangle$$

for $r \in [m]$, and by $\mathbb{E}[a_r \max_{i \in [n]} \langle \mathsf{S}_{i,r}(t), (x_{i,d} \Delta w_r(t))^2 \cdot x_i^{\circ 3} \rangle] = 0$, we have with probability $1 - \delta$,

$$\Big| \sum_{r=1}^m a_r \max_{i \in [n]} \langle \mathsf{S}_{i,r}(t), (x_{i,d} \Delta w_r(t))^2 \cdot x_i^{\circ 3} \rangle \Big| \leq O(\frac{n^3}{m}) \exp(O(B^2 D)) \cdot \|\mathsf{F}(t) - y\|_2^2 \cdot \sqrt{m \log(m/\delta)}$$

$$\leq O(\frac{n^3}{\sqrt{m}}) \exp(O(B^2 D)) \cdot \|\mathsf{F}(t) - y\|_2^2 \cdot \sqrt{\log(m/\delta)} \tag{21}$$

Above, the first inequality is derived from Hoeffding Inequality (Lemma B.1) and Eq. (18). The second inequality follows from basic algebraic manipulations.

Then we can have

$$\Big| \sum_{i=1}^n v_{2,i} \cdot (\mathsf{F}_i(t) - y_i) \Big|$$

$$\leq \frac{\eta^2 \sqrt{d}}{\sqrt{m}} \cdot \|\mathsf{F}(t) - y\|_2 \cdot O(\frac{n^3}{\sqrt{m}}) \exp(O(B^2 D)) \cdot \|\mathsf{F}(t) - y\|_2^2 \cdot \sqrt{\log(m/\delta)}$$

$$\leq O(\frac{\eta^2 n^3 \sqrt{d \log(m/\delta)}}{m}) \exp(O(B^2 D)) \Big( \exp(O(B^2 D)) \cdot O(\sqrt{n} R) + O(\sqrt{n} B) \Big) \cdot \|\mathsf{F}(t) - y\|_2^2$$

$$\leq O(\frac{\eta^2 n^3 \sqrt{d \log(m/\delta)}}{m}) \exp(O(B^2 D)) \cdot O(\sqrt{n}(R + B)) \cdot \|\mathsf{F}(t) - y\|_2^2$$

$$\leq O(\frac{\eta^2 n^{3.5} \sqrt{d} \cdot \sqrt{\log(m/\delta)}}{m}) \cdot \exp(O(B^2 D)) \cdot O(2B) \cdot \|\mathsf{F}(t) - y\|_2^2$$

$$\leq O(\eta^2 \frac{n^{3.5} \cdot d^{0.5}}{\sqrt{m}}) \cdot \exp(O(B^2 D)) \cdot \|\mathsf{F}(t) - y\|_2^2$$

$$\leq O(\eta \frac{n^{3.5} \cdot d^{0.5}}{\sqrt{m}}) \cdot \exp(O(B^2 D)) \cdot \|\mathsf{F}(t) - y\|_2^2$$

Above, the first inequality combines the result of Eq. (20) and Eq. (21). The second step can be obtained from basic algebraic manipulations; the third step is due to $R \leq B$ and basic algebraic manipulations, and the fourth step leverages the inequality $\sqrt{\log(m/\delta)} \leq \sqrt{m}$ and $O(B) \leq \exp(O(B^2 D))$,

Finally, by the lemma condition, we have

$$O(\eta \frac{n^{3.5} \cdot d^{0.5}}{\sqrt{m}}) \cdot \exp(O(B^2 D)) \cdot \|\mathsf{F}(t) - y\|_2^2 \leq \frac{1}{8} \eta \lambda \cdot L(t)$$

Then, we complete the proof. $\qquad\square$

### G.4 Bounding $C_3$

**Lemma G.4.** *Assuming the following conditions are satisfied:*

- *Let $i \in [n]$, $r \in [m]$ and $k \in [d]$.*

- *Let integer $t > 0$.*

- *Define training dataset $\mathcal{D} := \{(x_i, y_i)\}_{i=1}^n \subset \mathbb{R}^d \times \mathbb{R}$ as specified in Definition C.3.*

- *Initialize $w(0) \in \mathbb{R}^m$ as specified in Definition D.2.*

- *Initialize $a \in \mathbb{R}^m$ as specified in Definition D.2.*

- *Define $L(t) \in \mathbb{R}$ as specified in Definition D.4.*

- *Define $\Delta w_r(t) \in \mathbb{R}$ as specified in Definition D.5.*

- *Define $\mathsf{S}_{i,r}(t)$ as specified in Definition E.3.*

- *Define $\mathsf{F}_i(t)$ as specified in Definition E.4.*

- *Let learning rate $\eta < 1$.*

- *Define $B > 1$ as specified in Definition K.1.*

- *Define $D > 1$ as specified in Definition K.2.*

- *Let $R \in (0, 0.01/B^2)$.*

- *Let $\delta \in (0, 0.1)$.*

- *Let $m \geq \Omega(\lambda^{-2} n^7 d \cdot \exp(O(B^2 D)))$.*

- *Followsing Lemma G.1 to define*

$$C_3 := -\eta^2 \Theta(1) \frac{1}{\sqrt{m}} \sum_{i=1}^n (\mathsf{F}_i(t) - y_i) \cdot \sum_{r=1}^m a_r \cdot \langle S_{i,r}(t), (x_{i,d} \Delta w_r(t))^2 \cdot x_i^{\circ 2} \rangle \cdot \langle S_{i,r}(t), x_i \rangle$$

*Consequently, with probability at least $1 - \delta$:*

$$C_3 \leq \frac{1}{8}\eta\lambda \cdot L(t)$$

*Proof.* Firstly, we go to bound $|\langle S_{i,r}(t), (x_{i,d}\Delta w_r(t))^2 \cdot x_i^{\circ 2}\rangle|$, we have

$$
\begin{aligned}
|\langle S_{i,r}(t), (x_{i,d}\Delta w_r(t))^2 \cdot x_i^{\circ 2}\rangle| &\leq \|S_{i,r}(t)\|_2 \cdot \|(x_{i,d}\Delta w_r(t))^2 \cdot x_i^{\circ 2}\|_2 \\
&\leq \sqrt{d} \cdot \frac{\exp(O(B^2(D+R)))}{d} \cdot O(B^2) \cdot \sqrt{d} \cdot O(B^2) \\
&\quad \cdot \frac{n^3}{m} \cdot \exp(O(B^2 D)) \cdot \|\mathsf{F}(t) - y\|_2^2 \\
&\leq \exp(O(B^2 D)) \cdot \frac{n^3}{m} \cdot \|\mathsf{F}(t) - y\|_2^2
\end{aligned}
\tag{22}
$$

where the first step is a consequence of Cauchy inequality, the second step is based on Lemma K.3 Part 1, 9, definition of $\ell_2$ norm and Lemma H.3, and the last step is because of $O(B^4) \leq \exp(O(B^2))$ and basic algebraic manipulations.

Then, we can show that

$$
\begin{aligned}
|C_3| &= \Big| - \eta^2\Theta(1)\frac{1}{\sqrt{m}}\sum_{i=1}^{n}(\mathsf{F}_i(t) - y_i) \cdot \sum_{r=1}^{m} a_r \cdot \langle S_{i,r}(t), (x_{i,d}\Delta w_r(t))^2 \cdot x_i^{\circ 2}\rangle \cdot \langle S_{i,r}(t), x_i\rangle \Big| \\
&\leq \frac{\eta^2}{\sqrt{m}}\Big| \sum_{i=1}^{n}\sum_{r=1}^{m} a_r \cdot \langle S_{i,r}(t), (x_{i,d}\Delta w_r(t))^2 \cdot x_i^{\circ 2}\rangle \cdot \langle S_{i,r}(t), x_i\rangle \cdot (\mathsf{F}_i(t) - y_i)\Big| \\
&\leq \frac{\eta^2}{\sqrt{m}}\Big| \sum_{r=1}^{m} a_r \max_{i\in[n]} \cdot \langle S_{i,r}(t), (x_{i,d}\Delta w_r(t))^2 \cdot x_i^{\circ 2}\rangle \cdot \langle S_{i,r}(t), x_i\rangle \Big| \cdot \|\mathsf{F}(t) - y\|_1 \\
&\leq \frac{\eta^2\sqrt{d}}{\sqrt{m}}\Big| \sum_{r=1}^{m} a_r \max_{i\in[n]} \cdot \langle S_{i,r}(t), (x_{i,d}\Delta w_r(t))^2 \cdot x_i^{\circ 2}\rangle \cdot \langle S_{i,r}(t), x_i\rangle \Big| \cdot \|\mathsf{F}(t) - y\|_2 \\
&\leq \frac{\eta^2\sqrt{d}}{\sqrt{m}} \cdot \Big( \exp(O(B^2 D)) \cdot O(\sqrt{n}R) + O(\sqrt{n}B)\Big) \\
&\quad \cdot \Big| \sum_{r=1}^{m} a_r \max_{i\in[n]} \cdot \langle S_{i,r}(t), (x_{i,d}\Delta w_r(t))^2 \cdot x_i^{\circ 2}\rangle \cdot \langle S_{i,r}(t), x_i\rangle \Big|
\end{aligned}
\tag{23}
$$

where the first step is from the condition given in this Lemma, the second step is derived through basic algebra manipulations, and the third step comes from the definition of $\ell_1$ norm and basic algebraic manipulations, the fourth step utilizes the inequality $\|x\|_1 \leq \sqrt{d}\|x\|_2$, and the final step is because of Eq. (19).

We can then use Hoeffding's inequality (Lemma B.1) for the random variable

$$a_r \cdot \max_{i\in[n]}\langle S_{i,r}(t), (x_{i,d}\Delta w_r(t))^2 \cdot x_i^{\circ 2}\rangle \cdot \langle S_{i,r}(t), x_i\rangle$$

for $r \in [m]$, and by $\mathbb{E}[a_r \cdot \max_{i\in[n]}\langle S_{i,r}(t), (x_{i,d}\Delta w_r(t))^2 \cdot x_i^{\circ 2}\rangle \cdot \langle S_{i,r}(t), x_i\rangle] = 0$, we have with probability $1 - \delta$,

$$
\begin{aligned}
&\Big| a_r \cdot \max_{i\in[n]}\langle S_{i,r}(t), (x_{i,d}\Delta w_r(t))^2 \cdot x_i^{\circ 2}\rangle \cdot \langle S_{i,r}(t), x_i\rangle \Big| \\
&\leq O(\frac{n^3}{m}) \cdot \exp(O(B^2 D)) \cdot \exp(O(B^2 D)) \cdot \|\mathsf{F}(t) - y\|_2^2 \cdot \sqrt{m\log(m/\delta)} \\
&\leq O(\frac{n^3}{\sqrt{m}}) \cdot \exp(O(B^2 D)) \cdot \|\mathsf{F}(t) - y\|_2^2 \cdot \sqrt{\log(m/\delta)}
\end{aligned}
\tag{24}
$$

where the first step is derived from Eq. (22) and Lemma B.1, the second step is due to basic algebraic manipulations.

Now, we are able to bound

$$|C_3| \leq \frac{\eta^2 \sqrt{d}}{\sqrt{m}} \cdot (\exp(O(B^2 D)) \cdot O(\sqrt{n}R) + O(\sqrt{n}B)) \cdot O(\frac{n^3}{\sqrt{m}})$$
$$\cdot \exp(O(B^2 D)) \cdot \|\mathsf{F}(t) - y\|_2^2 \cdot \sqrt{\log(m/\delta)}$$
$$\leq O(\frac{\eta^2 n^3 d^{0.5} \sqrt{\log(m/\delta)}}{m}) \cdot \exp(O(B^2 D)) \cdot \Big( \exp(O(B^2 D)) \cdot O(\sqrt{n}R) + O(\sqrt{n}B) \Big) \cdot \|\mathsf{F}(t) - y\|_2^2$$
$$\leq O(\frac{\eta^2 n^3 d^{0.5} \sqrt{\log(m/\delta)}}{m}) \exp(O(B^2 D)) \cdot O(\sqrt{n}(R + B)) \cdot \|\mathsf{F}(t) - y\|_2^2$$
$$\leq O(\eta \frac{n^{3.5} d^{0.5}}{\sqrt{m}}) \cdot \exp(O(B^2 D)) \cdot O(B) \cdot \|\mathsf{F}(t) - y\|_2^2$$
$$\leq O(\eta \frac{n^{3.5} d^{0.5}}{\sqrt{m}}) \cdot \exp(O(B^2 D)) \cdot \|\mathsf{F}(t) - y\|_2^2$$

Above the first inequality is a combination result of Eq. (23) and Eq. (24). The second and third inequalities follow from basic algebraic manipulations. The fourth step is a consequence of $\eta < 1$, $R \ll B$ and $\sqrt{\log(m/\delta)} \leq \sqrt{m}$, and the last step is based on the fact that $O(B) \leq \exp(O(B^2 D))$.

Finally, by the lemma condition, we have

$$O(\eta \frac{n^{3.5} \cdot d^{0.5}}{\sqrt{m}}) \cdot \exp(O(B^2 D)) \cdot \|\mathsf{F}(t) - y\|_2^2 \leq \frac{1}{8} \eta \lambda \cdot L(t)$$

Then, we complete the proof. $\qquad\square$

## G.5  Bounding $C_4$

**Lemma G.5.** *Assuming the following conditions are satisfied:*

- *Let $i \in [n]$, $r \in [m]$ and $k \in [d]$.*

- *Let integer $t > 0$.*

- *Define training dataset $\mathcal{D} := \{(x_i, y_i)\}_{i=1}^n \subset \mathbb{R}^d \times \mathbb{R}$ as specified in Definition C.3.*

- *Initialize $w(0) \in \mathbb{R}^m$ as specified in Definition D.2.*

- *Initialize $a \in \mathbb{R}^m$ as specified in Definition D.2.*

- *Define $L(t) \in \mathbb{R}$ as specified in Definition D.4.*

- *Define $\Delta w_r(t) \in \mathbb{R}$ as specified in Definition D.5.*

- *Define $\mathsf{S}_{i,r}(t)$ as specified in Definition E.3.*

- *Define $\mathsf{F}_i(t)$ as specified in Definition E.4.*

- *Let learning rate $\eta < 1$.*

- *Define $B > 1$ as specified in Definition K.1.*

- *Define $D > 1$ as specified in Definition K.2.*

- *Let $R \in (0, 0.01/B^2)$.*

- *Let $\delta \in (0, 0.1)$.*

- *Let $m \geq \Omega(\lambda^{-3} n^5 d^2 \cdot \exp(O(B^2 D)))$.*

- *Following Lemma G.1 to define*
$$\beta_{i,r}(t) := x_{i,d} \cdot \eta \Delta w_r(t) \cdot x_i + \Theta(1) \cdot (x_{i,d} \cdot \eta \Delta w_r(t) \cdot x_i)^{\circ 2}$$

- *Following Lemma G.1 to define*

$$C_4 := -\frac{1}{\sqrt{m}} \sum_{i=1}^{n} (\mathsf{F}_i(t) - y_i) \cdot \sum_{r=1}^{m} a_r \cdot \langle \mathsf{S}_{i,r}(t), \beta_{i,r}(t) \rangle \cdot \langle \mathsf{S}_{i,r}(t+1) - \mathsf{S}_{i,r}(t), x_i \rangle$$

*Consequently, with probability at least $1 - \delta$:*

$$C_4 \leq \frac{1}{8} \eta \lambda \cdot L(t)$$

*Proof.* Firstly, we begin to bound $|\langle \mathsf{S}_{i,r}(t+1) - \mathsf{S}_{i,r}(t), x_i \rangle|$, and we have

$$
\begin{aligned}
|\langle \mathsf{S}_{i,r}(t+1) - \mathsf{S}_{i,r}(t), x_i \rangle| &\leq \|\mathsf{S}_{i,r}(t+1) - \mathsf{S}_{i,r}(t)\|_2 \|x_i\|_2 \\
&\leq \sqrt{d} \cdot \frac{\exp(O(B^2 D)) \cdot O(RB^2)}{d} \cdot \sqrt{d} \cdot O(B) \\
&\leq \exp(O(B^2 D)) \cdot O(RB^3) \\
&\leq \exp(O(B^2 D)) \cdot O(R)
\end{aligned}
\tag{25}
$$

Above, the first inequality is a result of using Cauchy inequality, and the second inequality combines the result of Part 1, 13 of Lemma K.3, the 3rd step is derived from basic algebraic manipulations and the last step is based on the fact that $O(B^3) \leq \exp(O(B^2 D))$.

Then we proceed to bound $\|x_{i,d} \cdot \eta \Delta w_r(t) \cdot x_i\|_2$.

We have

$$
\begin{aligned}
\|x_{i,d} \cdot \eta \Delta w_r(t) \cdot x_i\|_2 &\leq \eta \cdot O(B) \cdot \sqrt{d} \cdot O(B) \cdot \frac{n^{3/2}}{\sqrt{m}} \cdot \exp(O(B^2 D)) \cdot \|\mathsf{F}(t) - y\|_2 \\
&\leq \eta \frac{n^{1.5} d^{0.5}}{\sqrt{m}} \cdot \exp(O(B^2 D)) \cdot \|\mathsf{F}(t) - y\|_2
\end{aligned}
\tag{26}
$$

Above the 1st step is based on Part 1 of Lemma K.3, Lemma H.3 and definition of $\ell_2$ norm, the second step comes from basic algebraic manipulations and the fact that $O(B^2) \leq \exp(O(B^2))$

Thus, we can get that

$$
\begin{aligned}
\|\Theta(1) \cdot (x_{i,d} \cdot \eta \Delta w_r(t) \cdot x_i)^{\circ 2}\|_2 &\leq \sqrt{d} \cdot \eta^2 \cdot \frac{n^3}{m} \cdot \exp(O(B^2 D)) \cdot \|\mathsf{F}(t) - y\|_2^2 \\
&\leq \eta^2 \frac{n^3 d^{0.5}}{m} \cdot \exp(O(B^2 D)) \cdot \|\mathsf{F}(t) - y\|_2^2 \\
&\leq \eta^2 \frac{n^3 d^{0.5}}{m} \cdot \exp(O(B^2 D)) \cdot \Big( \exp(O(B^2 D)) \cdot O(\sqrt{n} R) \\
&\quad + O(\sqrt{n} B) \Big) \cdot \|\mathsf{F}(t) - y\|_2 \\
&\leq \eta^2 \frac{n^{3.5} d^{0.5}}{m} \cdot \exp(O(B^2 D)) \cdot \|\mathsf{F}(t) - y\|_2
\end{aligned}
\tag{27}
$$

Above the first step is a consequence of Eq. (26) and definition of $\ell_2$ norm, and the second step is derived from basic algebraic manipulation.

Now we are able to bound to bound $|\langle \mathsf{S}_{i,r}(t), \beta_{i,r}(t) \rangle|$. We have

$$
\begin{aligned}
|\langle \mathsf{S}_{i,r}(t), \beta_{i,r}(t) \rangle| &\leq \|\mathsf{S}_{i,r}(t)\|_2 \cdot \|\beta_{i,r}(t)\|_2 \\
&\leq \sqrt{d} \cdot \frac{\exp(O(B^2(D + R)))}{d} \cdot \|\beta_{i,r}(t)\|_2 \\
&\leq \frac{\exp(O(B^2 D))}{\sqrt{d}} \cdot (\|x_{i,d} \cdot \eta \Delta w_r(t) \cdot x_i\|_2 + \|\Theta(1) \cdot (x_{i,d} \cdot \eta \Delta w_r(t) \cdot x_i)^{\circ 2}\|_2) \\
&\leq \frac{\exp(O(B^2 D))}{\sqrt{d}} \cdot (\eta \frac{d^{0.5} n^{1.5}}{\sqrt{m}} + \eta^2 \frac{n^{3.5} d^{0.5}}{m}) \cdot \exp(O(B^2 D)) \cdot \|\mathsf{F}(t) - y\|_2
\end{aligned}
$$

$$\leq (\eta \frac{n^{1.5}}{\sqrt{m}} + \eta^2 \frac{n^{3.5}}{m}) \cdot \exp(O(B^2 D)) \cdot \|\mathsf{F}(t) - y\|_2$$

$$\leq 2\eta \frac{n^{1.5}}{\sqrt{m}} \cdot \exp(O(B^2 D)) \cdot \|\mathsf{F}(t) - y\|_2$$

$$\leq \eta \frac{n^{1.5}}{\sqrt{m}} \cdot \exp(O(B^2 D)) \cdot \|\mathsf{F}(t) - y\|_2 \tag{28}$$

where the 1st step follows from Cauchy inequality, the second step is a consequence of Part 9 of Lemma K.3 and definition of $\ell_2$ norm, the 3rd step is because of basic algebraic manipulations and triangle inequality, the fourth step is obtained using Eq. (26) and Eq. (27), and the fifth step follows from basic algebraic manipulations, the sixth step is derived from $\|x_{i,d} \cdot \eta \Delta w_r(t) \cdot x_i\|_2 \geq \|\Theta(1) \cdot (x_{i,d} \cdot \eta \Delta w_r(t) \cdot x_i)^{\circ 2}\|_2$, and last step follows from $O(1) \leq \exp(O(B^2 D))$.

Then, we can show that

$$|C_4| = \left| -\frac{1}{\sqrt{m}} \sum_{i=1}^{n} \sum_{r=1}^{m} a_r \cdot \langle \mathsf{S}_{i,r}(t), \beta_{i,r}(t) \rangle \cdot \langle \mathsf{S}_{i,r}(t+1) - \mathsf{S}_{i,r}(t), x_i \rangle \cdot (\mathsf{F}_i(t) - y_i) \right|$$

$$\leq \frac{1}{\sqrt{m}} \cdot \left| \sum_{r=1}^{m} a_r \cdot \max_{i \in [n]} \langle \mathsf{S}_{i,r}(t), \beta_{i,r}(t) \rangle \cdot \langle \mathsf{S}_{i,r}(t+1) - \mathsf{S}_{i,r}(t), x_i \rangle \right| \cdot \|\mathsf{F}(t) - y\|_1$$

$$\leq \frac{\sqrt{d}}{\sqrt{m}} \cdot \left| \sum_{r=1}^{m} a_r \cdot \max_{i \in [n]} \langle \mathsf{S}_{i,r}(t), \beta_{i,r}(t) \rangle \cdot \langle \mathsf{S}_{i,r}(t+1) - \mathsf{S}_{i,r}(t), x_i \rangle \right| \cdot \|\mathsf{F}(t) - y\|_2$$

$$\tag{29}$$

Above, the first step is derived from the condition stated in this lemma, the second step results from basic algebra and the definition of the $\ell_1$ norm, the final step is based on the inequality $|x|_1 \leq \sqrt{d}|x|_2$.

Next, We use Hoeffding's Inequality (Lemma B.1) on the random variable

$$a_r \cdot \max_{i \in [n]} \langle \mathsf{S}_{i,r}(t), \beta_{i,r}(t) \rangle \cdot \langle \mathsf{S}_{i,r}(t+1) - \mathsf{S}_{i,r}(t), x_i \rangle$$

for $r \in [m]$, and by $\mathbb{E}[a_r \cdot \max_{i \in [n]} \langle \mathsf{S}_{i,r}(t), \beta_{i,r}(t) \rangle \cdot \langle \mathsf{S}_{i,r}(t+1) - \mathsf{S}_{i,r}(t), x_i \rangle] = 0$, we have probability $1 - \delta$,

$$\left| a_r \cdot \max_{i \in [n]} \langle \mathsf{S}_{i,r}(t), \beta_{i,r}(t) \rangle \cdot \langle \mathsf{S}_{i,r}(t+1) - \mathsf{S}_{i,r}(t), x_i \rangle \right|$$

$$\leq O(\eta \frac{n^{1.5}}{\sqrt{m}}) \cdot \exp(O(B^2 D)) \cdot \|\mathsf{F}(t) - y\|_2 \cdot \exp(O(B^2 D)) \cdot O(R) \cdot \sqrt{m \log(m/\delta)}$$

$$\leq O(\eta \cdot n^{1.5}) \cdot \sqrt{\log(m/\delta)} \cdot \exp(O(B^2 D)) \cdot O(R) \cdot \|\mathsf{F}(t) - y\|_2 \tag{30}$$

where the first step combines the result of Eq. (25), Eq. (28) and Lemma B.1, the second step is obtained through basic algebraic manipulations.

Now, we are able to bound

$$|C_4| \leq \frac{\sqrt{d}}{\sqrt{m}} \cdot O(\eta \cdot n^{1.5}) \cdot \sqrt{\log(m/\delta)} \cdot \exp(O(B^2 D)) \cdot O(R) \cdot \|\mathsf{F}(t) - y\|_2^2$$

$$\leq O(\eta \frac{n^{1.5} d^{0.5} \sqrt{\log(m/\delta)}}{\sqrt{m}}) \cdot \exp(O(B^2 D)) \cdot O(R) \cdot \|\mathsf{F}(t) - y\|_2^2$$

$$\leq O(\eta \frac{n^{1.5} d^{0.5} m^{\frac{1}{6}}}{\sqrt{m}}) \cdot \exp(O(B^2 D)) \cdot O(R) \cdot \|\mathsf{F}(t) - y\|_2^2$$

$$\leq O(\eta \frac{n^{1.5} d^{0.5}}{m^{\frac{1}{3}}}) \cdot \exp(O(B^2 D)) \cdot O(B) \cdot \|\mathsf{F}(t) - y\|_2^2$$

$$\leq O(\eta \frac{n^{1.5} d^{0.5}}{m^{\frac{1}{3}}}) \cdot \exp(O(B^2 D)) \cdot \|\mathsf{F}(t) - y\|_2^2$$

Above, the 1st inequality combines the result of Eq. (29) and Eq. (30), and the second inequality is derived through basic algebraic manipulations. The third step uses the inequality $\sqrt{\log(m/\delta)} \leq m^{\frac{1}{6}}$, the fourth step is based on $R \leq B$ and basic algebraic manipulations, and the final step relies on the fact that $O(B) \leq \exp(O(B^2 D))$.

Finally, based on the lemma condition, we will get

$$O(\eta \frac{n^{1.5} d^{0.5}}{m^{\frac{1}{3}}}) \cdot \exp(O(B^2 D)) \cdot \|F(t) - y\|_2^2 \leq \frac{1}{8} \eta \lambda L(t)$$

Then, we complete the proof. $\qquad \square$

## G.6 Bounding $C_5$

**Lemma G.6.** *Assuming the following conditions are satisfied:*

- *Let $i \in [n]$, $r \in [m]$ and $k \in [d]$.*

- *Let integer $t > 0$.*

- *Define training dataset $\mathcal{D} := \{(x_i, y_i)\}_{i=1}^n \subset \mathbb{R}^d \times \mathbb{R}$ as specified in Definition C.3.*

- *Initialize $w(0) \in \mathbb{R}^m$ as specified in Definition D.2.*

- *Initialize $a \in \mathbb{R}^m$ as specified in Definition D.2.*

- *Define $L(t) \in \mathbb{R}$ as specified in Definition D.4.*

- *Define $\Delta w_r(t) \in \mathbb{R}$ as specified in Definition D.5.*

- *Define $\mathsf{S}_{i,r}(t)$ as specified in Definition E.3.*

- *Define $\mathsf{F}_i(t)$ as specified in Definition E.4.*

- *Let learning rate $\eta < 1$.*

- *Define $B > 1$ as specified in Definition K.1.*

- *Define $D > 1$ as specified in Definition K.2.*

- *Let $R \in (0, 0.01/B^2)$.*

- *Let $\delta \in (0, 0.1)$.*

- *Let $\eta \leq O(\lambda n^{-4} d^{-1} \exp(O(B^2 D))^{-1})$.*

- *Following Lemma G.1 to define*

$$\beta_{i,r}(t) := x_{i,d} \cdot \eta \Delta w_r(t) \cdot x_i + \Theta(1) \cdot (x_{i,d} \cdot \eta \Delta w_r(t) \cdot x_i)^{\circ 2}$$

- *Following Lemma G.1*

$$C_5 := \frac{1}{2} \|F(t) - F(t+1)\|_2^2$$

*Then, with a probability at least $1 - \delta$, we have,*

$$C_5 \leq \frac{1}{8} \eta \lambda \cdot L(t)$$

*Proof.* We have

$$\frac{1}{2} \|F(t+1) - F(t)\|_2^2 = \frac{1}{2} \sum_{i=1}^n (F_i(t+1) - F_i(t))^2$$

$$
= \frac{1}{2} \sum_{i=1}^{n} (\sum_{r=1}^{m} a_r \cdot \langle \mathsf{S}_{i,r}(t+1), x_i \rangle - \sum_{r=1}^{m} a_r \cdot \langle \mathsf{S}_{i,r}(t), x_i \rangle)^2
$$

$$
= \frac{1}{2} \sum_{i=1}^{n} (\sum_{r=1}^{m} a_r \langle \mathsf{S}_{i,r}(t+1) - \mathsf{S}_{i,r}(t), x_i \rangle)^2
$$

$$
= \frac{1}{2} \sum_{i=1}^{n} (\sum_{r=1}^{m} a_r \langle \alpha_{i,r}(t+1)^{-1} \cdot \mathsf{u}_{i,r}(t+1) - \alpha_{i,r}(t)^{-1} \cdot \mathsf{u}_{i,r}(t), x_i \rangle)^2
$$

$$
= \frac{1}{2} \sum_{i=1}^{n} (\sum_{r=1}^{m} a_r \langle (\alpha_{i,r}(t+1)^{-1} - \alpha_{i,r}(t)^{-1}) \cdot \mathsf{u}_{i,r}(t+1), x_i \rangle
$$

$$
+ \sum_{r=1}^{m} a_r \langle (\mathsf{u}_{i,r}(t+1) - \mathsf{u}_{i,r}(t)) \cdot \alpha_{i,r}(t)^{-1}, x_i \rangle)^2
$$

$$
= \frac{1}{2} \sum_{i=1}^{n} (Q_{i,1} + Q_{i,2})^2
$$

Above, the first equation is because of the definition of the $\ell_2$ norm. The 2nd equation comes from Definition E.4. The 3rd equation results from basic algebraic manipulations. The fourth equation is derived from Definition E.3. The fifth equation follows from basic algebraic manipulations. The last equation is based on the following definition:

$$
Q_{i,1} := \sum_{r=1}^{m} a_r \langle (\alpha_{i,r}(t+1)^{-1} - \alpha_{i,r}(t)^{-1}) \cdot \mathsf{u}_{i,r}(t+1), x_i \rangle
$$

$$
Q_{i,2} := \sum_{r=1}^{m} a_r \langle (\mathsf{u}_{i,r}(t+1) - \mathsf{u}_{i,r}(t)) \cdot \alpha_{i,r}(t)^{-1}, x_i \rangle
$$

**To bound $Q_{i,1}$.** For the first term, we first bound

$$
|\langle (\alpha_{i,r}(t+1)^{-1} - \alpha_{i,r}(t)^{-1}) \cdot \mathsf{u}_{i,r}(t+1), x_i \rangle|
$$

$$
\leq |\alpha_{i,r}(t+1)^{-1} - \alpha_{i,r}(t)^{-1}| \cdot \|\mathsf{u}_{i,r}(t+1)\|_2 \cdot \|x_i\|_2
$$

$$
\leq \eta \frac{n^{1.5}}{d\sqrt{m}} \cdot \exp(O(B^2 D)) \cdot \|\mathsf{F}(t) - y\|_2 \cdot \sqrt{d} \cdot \exp(O(B^2 D)) \cdot \sqrt{d} \cdot O(B)
$$

$$
\leq \eta \frac{n^{1.5}}{\sqrt{m}} \cdot \exp(O(B^2 D)) \cdot \|\mathsf{F}(t) - y\|_2 \tag{31}
$$

where the first step is based on the Cauchy-Schwarz inequality and basic algebra, the next step combines Part 1,5 of Lemma K.3, definition of $\ell_2$ norm and Lemma G.7, and the final step follows from $o(B) \leq \exp(O(B^2 D))$ and basic algebraic manipulations.

Then we can apply Hoeffding bound to random variable $a_r \langle (\alpha_{i,r}(t+1)^{-1} - \alpha_{i,r}(t)^{-1}) \cdot \mathsf{u}_{i,r}(t+1), x_i \rangle$ for $r \in [m]$ and $\mathbb{E}[\sum_{r=1}^{m} a_r \langle (\alpha_{i,r}(t+1)^{-1} - \alpha_{i,r}(t)^{-1}) \cdot \mathsf{u}_{i,r}(t+1), x_i \rangle]$, we have

$$
|Q_{i,1}| \leq |\sum_{r=1}^{m} a_r \langle (\alpha_{i,r}(t+1)^{-1} - \alpha_{i,r}(t)^{-1}) \cdot \mathsf{u}_{i,r}(t+1), x_i \rangle|
$$

$$
\leq O(\eta \frac{n^{1.5}}{\sqrt{m}}) \cdot \exp(O(B^2 D)) \cdot \|\mathsf{F}(t) - y\|_2 \cdot \sqrt{m \log(m/\delta)}
$$

$$
= O(\eta n^{1.5}) \cdot \exp(O(B^2 D)) \cdot \|\mathsf{F}(t) - y\|_2 \cdot \sqrt{\log(m/\delta)}
$$

where the first step follows from definition of $Q_{i,1}$, the second step follows from Eq. (31) and Lemma B.1, the last step follows from basic algebraic manipulations.

**To bound $Q_{i,2}$.** For the second term, we first bound

$$
|\langle (\mathsf{u}_{i,r}(t+1) - \mathsf{u}_{i,r}(t)) \cdot \alpha_{i,r}(t)^{-1}, x_i \rangle|
$$

$$\leq |\alpha_{i,r}(t)^{-1}| \cdot \|\mathsf{u}_{i,r}(t+1) - \mathsf{u}_{i,r}(t)\|_2 \cdot \|x_i\|_2$$

$$\leq \frac{\exp(O(B^2 D))}{d} \cdot \sqrt{d} \cdot O(B) \cdot \|\mathsf{u}_{i,r}(t) \circ \beta_{i,r}(t)\|_2$$

$$\leq \frac{\exp(O(B^2 D))}{\sqrt{d}} \cdot O(B) \cdot \|\mathsf{u}_{i,r}(t)\|_2 \cdot \|\beta_{i,r}(t)\|_2$$

$$\leq \frac{\exp(O(B^2 D))}{\sqrt{d}} \cdot O(B) \cdot \sqrt{d} \cdot \exp(O(B^2 D))$$

$$\cdot (\eta \frac{n^{1.5} d^{0.5}}{\sqrt{m}} + \eta^2 \frac{n^{3.5} d^{0.5}}{m}) \cdot \exp(O(B^2 D)) \cdot \|\mathsf{F}(t) - y\|_2$$

$$\leq \eta \frac{n^{1.5} d^{0.5}}{\sqrt{m}} \exp(O(B^2 D)) \cdot \|\mathsf{F}(t) - y\|_2 \tag{32}$$

where the 1st step is derived from basic algebraic manipulations and Cauchy inequality, the 2nd step utilizes Lemma K.3 Part 1 and 7 along with the definition of the $\ell_2$ norm, the third step is a result of applying the Cauchy inequality, the fourth step combines Part 5 of Lemma K.3, the definition of $\ell_2$ norm, Eq. (26) and Eq. (27), the last step leverages the inequality $\|x_{i,d} \cdot (-\eta \Delta w_r(t) \cdot x_i)\|_2 \geq \|\Theta(1) \cdot (x_{i,d} \cdot (-\eta \Delta w_r(t) \cdot x_i)^{\circ 2}\|_2$.

We then can use Hoeffding Inequality (Lemma B.1) to random variable $a_r \langle (\mathsf{u}_{i,r}(t+1) - \mathsf{u}_{i,r}(t)) \cdot \alpha_{i,r}(t)^{-1}, x_i \rangle$, for $r \in [m]$ and $\mathbb{E}[\sum_{r=1}^{m} a_r \langle (\mathsf{u}_{i,r}(t+1) - \mathsf{u}_{i,r}(t)) \cdot \alpha_{i,r}(t)^{-1}, x_i \rangle] = 0$.

Then we have

$$|Q_{i,2}| \leq |\sum_{r=1}^{m} a_r \langle (\mathsf{u}_{i,r}(t+1) - \mathsf{u}_{i,r}(t)) \cdot \alpha_{i,r}(t)^{-1}, x_i \rangle|$$

$$\leq O(\eta \frac{n^{1.5} d^{0.5}}{\sqrt{m}}) \cdot \exp(O(B^2 D)) \cdot \|\mathsf{F}(t) - y\|_2 \cdot \sqrt{m \log(m/\delta)}$$

$$\leq O(\eta n^{1.5} \cdot d^{0.5}) \cdot \exp(O(B^2 D)) \cdot \|\mathsf{F}(t) - y\|_2 \cdot \sqrt{\log(m/\delta)} \tag{33}$$

where the first step is a consequence of the definition of $Q_{i,2}$, the second step is derived from Eq. (32) and Lemma B.1 and the final step is a result of basic algebraic manipulation.

Hence we have

$$\frac{1}{2} \sum_{i=1}^{n} (Q_{i,1} + Q_{i,2})^2$$

$$\leq \frac{1}{2} n \cdot \max_{i \in [n]} (2Q_{i,2})^2$$

$$\leq 2n \cdot O(\eta^2 n^3 d) \cdot \exp(O(B^2 D)) \cdot \log(m/\delta) \cdot \|\mathsf{F}(t) - y\|_2^2$$

$$\leq O(\eta^2 n^4 d) \cdot \exp(O(B^2 D)) \cdot D^2 \cdot \|\mathsf{F}(t) - y\|_2^2$$

$$\leq O(\eta^2 n^4 d) \cdot \exp(O(B^2 D)) \cdot \|\mathsf{F}(t) - y\|_2^2$$

where the first step is based on $Q_{i,1} \leq Q_{i,2}$, the second step is a consequence of Eq. (33) and basic algebraic manipulations, the 3rd step is based on Definition K.2 and basic algebraic manipulations, and the final step uses the inequality $O(D^2) \leq \exp(O(B^2 D))$. $\qquad\square$

## G.7 Helpful Lemma

**Lemma G.7.** *Assuming the following conditions are satisfied:*

- *Let $i \in [n]$, $r \in [m]$ and $k \in [d]$.*

- *Let integer $t > 0$.*

- *Define training dataset $\mathcal{D} := \{(x_i, y_i)\}_{i=1}^{n} \subset \mathbb{R}^d \times \mathbb{R}$ as specified in Definition C.3.*

- *Define $a \in \mathbb{R}^m$ as specified in Definition D.2.*

- *Define $\Delta w_r(t) \in \mathbb{R}$ as specified in Definition D.5.*

- *Define $\mathsf{u}_{i,r}(t) \in \mathbb{R}^d$ as specified in Definition E.1*

- *Define $\alpha_{i,r}(t) \in \mathbb{R}$ as specified in Definition E.2.*

- *Define $\mathsf{F}_{i,r}(t) \in \mathbb{R}$ as specified in Definition E.4.*

- *Define $\beta_{i,r}(t) \in \mathbb{R}^d$ as specified in Lemma G.1.*

- *Define $B > 1$ as specified in Definition K.1.*

- *Define $D > 1$ as specified in Definition K.2.*

- *Let $R \in (0, 0.01/B^2)$.*

- *Let $\delta \in (0, 0.1)$.*

*With probability at least $1 - \delta$, we obtain:*

- **Part 1.**

$$|\alpha_{i,r}(t+1) - \alpha_{i,r}(t)| \leq \eta \frac{n^{1.5}d}{\sqrt{m}} \exp(O(B^2 D)) \cdot \|\mathsf{F}(t) - y\|_2$$

- **Part 2.**

$$|\alpha_{i,r}(t+1)^{-1} - \alpha_{i,r}(t)^{-1}| \leq \eta \frac{n^{1.5}}{d\sqrt{m}} \cdot \exp(O(B^2 D)) \cdot \|\mathsf{F}(t) - y\|_2$$

*Proof.* **Proof of Part 1.** Firstly, we will have

$$
\begin{aligned}
&|\alpha_{i,r}(t+1) - \alpha_{i,r}(t)| \\
=& |\langle \mathsf{u}_{i,r}(t), \beta_{i,r}(t)\rangle| \\
\leq& \|\mathsf{u}_{i,r}(t)\|_2 \cdot \|\beta_{i,r}(t)\|_2 \\
\leq& \sqrt{d} \cdot \exp(O(B^2 D)) \cdot (\|x_{i,d} \cdot (-\eta \Delta w_r(t) \cdot x_i)\|_2 + \|\Theta(1) \cdot (x_{i,d} \cdot (-\eta \Delta w_r(t) \cdot x_i)^{\circ 2}\|_2) \\
\leq& \sqrt{d} \cdot \exp(O(B^2 D)) \cdot (\eta \frac{n^{1.5}d^{0.5}}{\sqrt{m}} + \eta^2 \frac{n^{3.5}d^{0.5}}{m}) \cdot \exp(O(B^2 D)) \cdot \|\mathsf{F}(t) - y\|_2 \\
\leq& \eta \frac{n^{1.5}d}{\sqrt{m}} \exp(O(B^2 D)) \cdot \|\mathsf{F}(t) - y\|_2
\end{aligned}
$$

where the first step is derived from Eq. (16), the second step is obtained by using Cauchy-Schwarz inequality, the third step combines Part 5 of Lemma K.3 and triangle inequality, the fourth step can be obtained by Eq. (26) and Eq. (27), the last step is a consequence of basic algebraic manipulations and $\|x_{i,d} \cdot (-\eta \Delta w_r(t) \cdot x_i)\|_2 \geq \|\Theta(1) \cdot (x_{i,d} \cdot (-\eta \Delta w_r(t) \cdot x_i)^{\circ 2}\|_2$.

**Proof of Part 2.** We have

$$
\begin{aligned}
&|\alpha_{i,r}(t+1)^{-1} - \alpha_{i,r}(t)^{-1}| \\
=& \alpha_{i,r}(t+1)^{-1} \cdot \alpha_{i,r}(t)^{-1} |\alpha_{i,r}(t+1) - \alpha_{i,r}(t)| \\
\leq& \frac{\exp(O(B^2 D))}{d} \cdot \frac{\exp(O(B^2 D))}{d} \cdot \eta \frac{n^{1.5}d}{\sqrt{m}} \cdot \exp(O(B^2 D)) \cdot \|\mathsf{F}(t) - y\|_2 \\
\leq& \eta \frac{n^{1.5}}{d\sqrt{m}} \cdot \exp(O(B^2 D)) \cdot \|\mathsf{F}(t) - y\|_2
\end{aligned}
$$

where the 1st step involves basic algebra, the 2nd step applies Lemma K.3 Part 7 and the definition of the $\ell_2$ norm, and the final step results from basic algebraic manipulations. $\square$

# H Inductions

## H.1 Induction for Loss

**Lemma H.1.** *Assuming the following conditions are satisfied:*

- *Let $i \in [n]$ and $r \in [m]$.*

- *Let $C > 0$ be a sufficiently large constant.*

- *Let $\sigma > 0$ be a small constant.*

- *Define $u \in \mathbb{R}$ as specified in Claim C.2.*

- *Define $\mathcal{P} \in \mathbb{R}^N$ as specified in Claim C.2.*

- *Define $t > 0$ be an integer.*

- *Define $h \in \mathbb{R}^N$ as specified in Definition C.3.*

- *Define $\xi \in \mathbb{R}$ as specified in Definition C.3.*

- *Define training dataset $\mathcal{D} := \{(x_i, y_i)\}_{i=1}^n \subset \mathbb{R}^d \times \mathbb{R}$ as specified in Definition C.3.*

- *Define $L(t) \in \mathbb{R}$ as specified in Definition D.4.*

- *Define $\mathsf{F}_i(t) \in \mathbb{R}$ as specified in Definition E.4.*

- *Define $B \in \mathbb{R}$ as specified in Definition K.1.*

- *Define $D \in \mathbb{R}$ as specified in Definition K.2.*

- *Define*

$$C_1 := -\eta \frac{1}{\sqrt{m}} \sum_{i=1}^n (\mathsf{F}_i(t) - y_i) \cdot \sum_{r=1}^m a_r \cdot \Big( \langle \mathsf{S}_{i,r}(t), (x_{i,d} \Delta w_r(t)) \cdot x_i^{\circ 2} \rangle$$
$$+ \langle \mathsf{S}_{i,r}(t), x_i \rangle^2 \cdot (x_{i,d} \Delta w_r(t)) \Big)$$

- *Define*

$$C_2 := -\eta^2 \Theta(1) \frac{1}{\sqrt{m}} \sum_{i=1}^n (\mathsf{F}_i(t) - y_i) \cdot \sum_{r=1}^m a_r \cdot \langle \mathsf{S}_{i,r}(t), (x_{i,d} \Delta w_r(t))^2 \cdot x_i^{\circ 3} \rangle$$

- *Define*

$$C_3 := -\eta^2 \Theta(1) \frac{1}{\sqrt{m}} \sum_{i=1}^n (\mathsf{F}_i(t) - y_i) \cdot \sum_{r=1}^m a_r \cdot \langle \mathsf{S}_{i,r}(t), (x_{i,d} \Delta w_r(t))^2 \cdot x_i^{\circ 2} \rangle \cdot \langle \mathsf{S}_{i,r}(t), x_i \rangle$$

- *Define*

$$C_4 := -\frac{1}{\sqrt{m}} \sum_{i=1}^n (\mathsf{F}_i(t) - y_i) \cdot \sum_{r=1}^m a_r \cdot \langle \mathsf{S}_{i,r}(t), \beta_{i,r}(t) \rangle \cdot \langle \mathsf{S}_{i,r}(t+1) - \mathsf{S}_{i,r}(t), x_i \rangle$$

- *Define*

$$C_5 := \frac{1}{2} \|\mathsf{F}(t) - \mathsf{F}(t+1)\|_2^2$$

*With probability at least $1 - \delta$, we obtain:*

- *Part 1.*

$$L(t + 1) \leq (1 - \eta\lambda/2) \cdot L(t)$$

- *Part 2.*

$$L(t) \leq (1 - \eta\lambda/2)^t \cdot L(0)$$

*Proof.* **Proof of Part 1.** Firstly:

$$L(t + 1) = L(t) + C_1 + C_2 + C_3 + C_4 + C_5$$
$$\leq L(t) - \eta\lambda L(t) + \frac{1}{8}\eta\lambda L(t) + \frac{1}{8}\eta\lambda L(t) + \frac{1}{8}\eta\lambda L(t) + \frac{1}{8}\eta\lambda L(t)$$
$$= (1 - \eta\lambda/2) \cdot L(t)$$

where the first step is based on Lemma G.1, the second step combines Lemma G.2, G.3, G.4, 27 and G.6, the final step results from basic algebraic manipulations.

**Choice of $m$ and $\eta$.** Following Lemma G.2, G.3, G.4, 27 and G.6, we choose:

$$m \geq \Omega\Big(\lambda^{-3}n^7d^2 \operatorname{poly}(\exp(B^2), \exp(D))\Big)$$
$$\eta \leq O\Big(\lambda n^{-4}d^{-1} \cdot \operatorname{poly}(\exp(B^2), \exp(D))^{-1}\Big)$$

**Proof of Part 2.** We have

$$L(t) \leq (1 - \eta\lambda/2)^t \cdot L(0)$$

which can be derived from Part 1. □

**Lemma H.2.** *Suppose we have the following:*

- *Let $i \in [n]$ and $r \in [m]$.*
- *Let $C > 0$ be a sufficiently large constant.*
- *Let $\sigma > 0$ be a small constant.*
- *Define $u \in \mathbb{R}$ as specified in Claim C.2.*
- *Define $\mathcal{P} \in \mathbb{R}^N$ as specified in Claim C.2.*
- *Define $t > 0$ be an integer.*
- *Define $h \in \mathbb{R}^N$ as specified in Definition C.3.*
- *Define $\xi \in \mathbb{R}$ as specified in Definition C.3.*
- *Define training dataset $\mathcal{D} := \{(x_i, y_i)\}_{i=1}^n \subset \mathbb{R}^d \times \mathbb{R}$ as specified in Definition C.3.*
- *Define $L(t) \in \mathbb{R}$ as specified in Definition D.4.*
- *Define $\mathsf{F}_i(t) \in \mathbb{R}$ as specified in Definition E.4.*
- *Define $B \in \mathbb{R}$ as specified in Definition K.1.*
- *Define $D \in \mathbb{R}$ as specified in Definition K.2.*

*With probability at least $1 - \delta$, we obtain:*

- *Part 1.*

$$L(0) \leq O(nB^2)$$

- *Part 2.*

$$L(t) \le \exp\left(O(B^2 D)\right) \cdot O(nR^2) + O(nB^2)$$

*Proof.* **Proof of Part 1.** Firstly, we have

$$
\begin{aligned}
y_i &= u_{i,d+1} + \xi_{i,d+1} \\
&= \langle \mathcal{P}_{i,d+1}, h_{i,1} \rangle + \xi_{i,d+1}
\end{aligned}
\tag{34}
$$

Above, the 1st equation is based on Definition C.3. The 2nd equation is trivially from Claim C.2. And we have $\xi_{i,d+1} \sim \mathcal{N}(0, \sigma^2)$ and $h_{i,1} \sim \mathcal{N}(0, I_N)$ which follows from Definition C.3, and $\|\mathcal{P}_{i,d+1}\|_2 = 1$ which follows from Claim C.2.

Then we can show

$$\langle \mathcal{P}_{i,d+1}, h_{i,1} \rangle \sim \mathcal{N}(0,1)$$

where this step comes from Fact B.4.

Thus, we can get

$$y_i \sim \mathcal{N}(0, 1 + \sigma^2)$$

where this step comes from Eq. (34), Fact B.3 and $\xi_{i,d+1} \sim \mathcal{N}(0, \sigma^2)$.

Consequently, with probability at least $1 - \delta$, we have

$$
\begin{aligned}
|y_i| &\le C\sqrt{1 + \sigma}\sqrt{\log(1/\delta)} \\
&\le O(B)
\end{aligned}
\tag{35}
$$

where the first inequality is derived from Fact B.5, and the second inequality uses Definition K.1.

Finally, we can show

$$
\begin{aligned}
L(0) &= \frac{1}{2} \sum_{i=1}^{n} (\mathsf{F}_i(0) - y_i)^2 \\
&= \frac{1}{2} \sum_{i=1}^{n} (-y_i)^2 \\
&\le O(nB^2)
\end{aligned}
\tag{36}
$$

Above, the first equation is trivially from Definition D.4. The second equation is due to Assumption D.7, and the last step uses Eq. (35) and basic algebraic manipulations.

**Proof of Part 2.** Firstly, we can show that

$$
\begin{aligned}
L(t) &= \frac{1}{2} \|\mathsf{F}(t) - y\|_2^2 \\
&= \frac{1}{2} \sum_{i=1}^{n} \left( (\mathsf{F}_i(t) - \mathsf{F}_i(0)) - (\mathsf{F}_i(0) - y_i) \right)^2 \\
&\le \frac{1}{2} \left( \sum_{i=1}^{n} ((\mathsf{F}_i(t) - \mathsf{F}_i(0))^2 + \sum_{i=1}^{n} ((\mathsf{F}_i(0) - y_i)^2 \right) \\
&\le \frac{1}{2} \|\mathsf{F}(t) - \mathsf{F}(0)\|_2^2 + O(nB^2) \\
&\le \exp\left(O(B^2 D)\right) \cdot O(nR^2) + O(nB^2)
\end{aligned}
$$

Above the first equation is due to Definition D.4 The second equation is a result of basic algebra and the $\ell_2$ norm definition. The third inequality comes from further algebraic manipulation, and the fourth inequality is based on Definition E.4 and Eq. (36). The final step is based on Lemma I.5 Part 2. $\qquad\square$

## H.2 Induction for Gradients

**Lemma H.3.** *Assuming the following conditions are satisfied:*

- *Let $i \in [n]$ and $r \in [m]$.*
- *Let training dataset $\mathcal{D} := \{(x_i, y_i)\}_{i=1}^{n} \subset \mathbb{R}^d$.*
- *Define $a \in \mathbb{R}^m$ as specified in Definition D.2.*
- *Define $L(t) \in \mathbb{R}$ as specified in Definition D.4.*
- *Let $w(0) \in \mathbb{R}^m$ be initialized as Definition D.2 and updated by Definition D.5.*
- *Define $\mathsf{S}_{i,r}(t) \in \mathbb{R}$ as specified in Definition E.3.*
- *Define $\mathsf{F}_i(t) \in \mathbb{R}$ as specified in Definition E.4.*
- *Define $B > 1$ as specified in Definition K.1.*
- *Define $D > 1$ as specified in Definition K.2.*
- *Let $\delta \in (0, 0.01)$.*

*With probability at least $1 - \delta$, we obtain:*

$$|\Delta w_r(t)| \leq \frac{n^{3/2}}{\sqrt{m}} \cdot \exp(O(B^2 D)) \cdot \|\mathsf{F}(t) - y\|_2$$

*Proof.* Firstly, we have that $\forall i \in [n]$

$$|(\mathsf{F}_i(t) - y_i) \cdot x_{i,d} \cdot (\langle \mathsf{S}_{i,r}(t), x_i^{\circ 2}\rangle - \langle \mathsf{S}_{i,r}(t), x_i\rangle^2)| \leq |\mathsf{F}_i(t) - y_i| \cdot O(B) \cdot \exp(O(B^2 D))$$
$$= \exp(O(B^2 D)) \cdot |\mathsf{F}_i(t) - y_i| \qquad (37)$$

where the first step is trivially from Lemma I.4 and Part 1 of Lemma K.3, the second step uses the fact $O(\mathrm{poly}(B)) \leq \exp(O(B))$.

Then, we can proceed to show that

$$\begin{aligned}
|\Delta w_r(t)| &= |\frac{\mathrm{d}}{\mathrm{d}w_r(t)} L(t)| \\
&= |\frac{1}{\sqrt{m}} a_r \sum_{i=1}^{n} (\mathsf{F}_i(t) - y_i) \cdot x_{i,d} \cdot (\langle \mathsf{S}_{i,r}(t), x_i^{\circ 2}\rangle - \langle \mathsf{S}_{i,r}(t), x_i\rangle^2)| \\
&\leq \frac{1}{\sqrt{m}} \cdot n \max_{i \in [n]} |(\mathsf{F}_i(t) - y_i) \cdot x_{i,d} \cdot (\langle \mathsf{S}_{i,r}(t), x_i^{\circ 2}\rangle - \langle \mathsf{S}_{i,r}(t), x_i\rangle^2)| \\
&\leq \frac{1}{\sqrt{m}} \cdot n \cdot O(B) \cdot \max_{i \in [n]} |(\mathsf{F}_i(t) - y_i)| \cdot |\langle \mathsf{S}_{i,r}(t), x_i^{\circ 2}\rangle - \langle \mathsf{S}_{i,r}(t), x_i\rangle^2| \\
&\leq \frac{1}{\sqrt{m}} \cdot n \cdot O(B) \cdot \max_{i \in [n]} |(\mathsf{F}_i(t) - y_i)| \cdot (|\langle \mathsf{S}_{i,r}(t), x_i^{\circ 2}\rangle| + |\langle \mathsf{S}_{i,r}(t), x_i\rangle^2|) \\
&\leq \frac{1}{\sqrt{m}} \cdot n \cdot O(B) \cdot \exp(O(B^2(D+R))) \cdot O(B^2) \cdot \|\mathsf{F}(t) - y\|_1 \\
&\leq \frac{1}{\sqrt{m}} \cdot n \cdot \exp(O(B^2 D)) \cdot \|\mathsf{F}(t) - y\|_1 \\
&\leq \frac{n^{3/2}}{\sqrt{m}} \cdot \exp(O(B^2 D)) \cdot \|\mathsf{F}(t) - y\|_2
\end{aligned}$$

Above, the first equation is derived from Definition D.5. The second equation uses the result of Part 6 of Lemma E.5, and the third inequality is derived from $a_r \sim \mathsf{Uniorm}\{-1, +1\}$ and $|\sum_{i=1}^{n} x_i| \leq n|\max_{i \in [n]} x_i|$. The fourth inequality is a consequence Part 1 of Lemma K.3, and the fifth inequality

is trivially from triangle inequality, the sixth step combines the definition of $\ell_1$ norm, Eq. (5) and Eq. (6), the seventh step applies the fact that $O(\mathrm{poly}(B)) \leq \exp(O(B^2))$, $R \in (0, 0.01)$ and $B \geq 1$ and the last step uses the inequality $\|x\|_1 \leq \sqrt{n}\|x\|_2$. $\qquad\square$

## H.3  Induction for Weights

**Lemma H.4.** *Assuming the following conditions are satisfied:*

- *Let $i \in [n]$ and $r \in [m]$.*

- *Let training dataset $\mathcal{D} := \{(x_i, y_i)\}_{i=1}^n \subset \mathbb{R}^d$.*

- *Define $a \in \mathbb{R}^m$ as specified in Definition D.2.*

- *Define $L(t) \in \mathbb{R}$ as specified in Definition D.4.*

- *Let $w(0) \in \mathbb{R}^m$ be initialized as Definition D.2 and updated by Definition D.5.*

- *Define $\mathsf{S}_{i,r}(t) \in \mathbb{R}$ as specified in Definition E.3.*

- *Define $\mathsf{F}_i(t) \in \mathbb{R}$ as specified in Definition E.4.*

- *Define $B > 1$ as specified in Definition K.1.*

- *Define $D > 1$ as specified in Definition K.2.*

- *Let $\delta \in (0, 0.01)$.*

- *Choose $m \geq \Omega(\lambda^{-2} n^7 \mathrm{poly}(\exp(B^2), \exp(D)))$.*

*Then, with probability no less than $1 - \delta$, we will get*

$$R := \max_{t \geq 0} \max_{r \in [m]} |w_r(t) - w_r(0)| \leq \frac{\lambda}{n \, \mathrm{poly}(\exp(B^2), \exp(D))}$$

*Proof.* We have:

$$R := \max_{t \geq 0} \max_{r \in [m]} |w_r(t) - w_r(0)|$$

$$\leq \eta \lim_{t \to +\infty} \max_{r \in [m]} \sum_{\tau=1}^t |\Delta w_r(\tau)|$$

$$\leq \eta \lim_{t \to +\infty} \sum_{\tau=1}^t \frac{n^{3/2}}{\sqrt{m}} \cdot \exp(O(B^2 D)) \cdot \|\mathsf{F}(\tau) - y\|_2$$

$$\leq \eta \lim_{t \to +\infty} \sum_{\tau=1}^t \frac{n^{3/2}}{\sqrt{m}} \cdot \exp(O(B^2 D)) \cdot (1 - \eta\lambda/2)^\tau L(0)$$

$$\leq \eta \lim_{t \to +\infty} \sum_{\tau=1}^t \frac{n^{3/2}}{\sqrt{m}} \cdot \exp(O(B^2 D)) \cdot (1 - \eta\lambda/2)^\tau O(nB^2)$$

$$\leq O(\frac{n^{5/2}}{\sqrt{m}\lambda}) \cdot \exp(O(B^2 D))$$

$$\leq \frac{\lambda}{n \, \mathrm{poly}(\exp(B^2), \exp(D))}$$

Above the first step is based on the definition of $R$, the second step results from basic algebra, the third step follows from Lemma H.3, the fourth step use basic algebraic manipulations and Part 2 of Lemma H.1, the fifth step is based on Lemma H.2 Part 1, the sixth step is based on Fact B.9 and $B^2 \leq \exp(O(B^2 D))$, last step is a consequence of the choice of $m$. $\qquad\square$

# I  Asymmetric Learning

## I.1  Main Results 1: Attention Convergence with Asymmetric Learning

**Theorem I.1.** *Assuming the following conditions are satisfied:*

- *Denote $v_{\min} := \min\{\frac{1}{d}\sum_{k=1}^{d}(x_{i,k} - \overline{x}_i)^2\}_{i=1}^{n}$.*

- *Choose $m \geq \Omega\Big(\lambda^{-3}n^7 d^2 \operatorname{poly}(\exp(B^2), \exp(D))\Big).$*

- *Choose $\eta \leq O\Big(\lambda n^{-4}d^{-1} \cdot \operatorname{poly}(\exp(B^2), \exp(D))^{-1}\Big).$*

- *Choose $T \geq \Omega\Big(\frac{1}{\eta\lambda}\log(nB^2/\epsilon)\Big)$*

*Consequently, the following holds with probability at least $1 - \delta$:*

$$L(T) \leq \epsilon.$$

**Asymmetric Learning.** *We can also show that for any $t \geq \Omega(\frac{m}{\eta\lambda v_{\min}})$:*

- *Part 1.*

$$\Pr[w_r(t) > 0 | a_r = 1] \geq 1 - \delta$$

- *Part 2.*

$$\Pr[w_r(t) < 0 | a_r = -1] \geq 1 - \delta$$

*Proof.* We have:

$$
\begin{aligned}
L(t) &\leq (1 - \eta\lambda/2)^t L(0) \\
&\leq (1 - \eta\lambda/2)^t \cdot O(nB^2) \\
&\leq \epsilon
\end{aligned}
$$

Above, the first inequality is due to Part 2 of Lemma H.1, and the second inequality is due to Lemma H.2 Part 1. The final step uses Fact B.9 and plugging $t = \Omega(\frac{1}{\eta\lambda}\log(nB^2/\epsilon))$.

**Choice of $m$ and $\eta$.** Combining Lemma H.1 and H.4, we have:

$$
\begin{aligned}
m &\geq \Omega\Big(\lambda^{-3}n^7 d^2 \operatorname{poly}(\exp(B^2), \exp(D))\Big) \\
\eta &\leq O\Big(\lambda n^{-4}d^{-1} \cdot \operatorname{poly}(\exp(B^2), \exp(D))^{-1}\Big)
\end{aligned}
$$

**Proof of Part 1.** When $a_r = 1$, we have:

$$
\begin{aligned}
w_r(t) &= w_r(0) - \eta \sum_{\tau=1}^{t} \Delta w_r(\tau) \\
&\geq -O(B) + t\eta \cdot O(\frac{n\gamma v_{\min}}{\sqrt{m}}) \cdot \exp(-O(B^2 D)) \\
&> 0
\end{aligned}
$$

Above, the 1st equation is based on Definition D.5, and the 2nd inequality is based on Lemma K.3 Part 1 and Lemma I.3 Part 1. The last inequality follows from plugging $t \geq \Omega(\frac{m}{\eta\lambda v_{\min}})$.

**Proof of Part 2.** When $a_r = -1$, we have:

$$w_r(t) = w_r(0) - \eta \sum_{\tau=1}^{t} \Delta w_r(\tau)$$

$$\leq O(B) - t\eta \cdot O(\frac{n\gamma v_{\min}}{\sqrt{m}}) \cdot \exp(-O(B^2 D))$$
$$< 0$$

Above, the 1st equation is based on Definition D.5, and the 2nd inequality is based on Lemma K.3 Part 1 and Lemma I.3 Part 1. The last inequality follows from plugging $t \geq \Omega(\frac{m}{\eta\lambda v_{\min}})$. □

## I.2 Main Results 2: Attention Fails in Learning Residual Feature

**Theorem I.2.** *Let all pre-conditions in Theorem I.1 hold.For any Gaussian vector $x \sim \mathcal{N}(0, \sigma'^2 \cdot I_d)$. For all $r \in [m]$ that satisfies $a_r = -1$, with a probability at least $1 - \delta$, we have:*

$$\mathbb{E}[\mathsf{softmax}_d(x_d \cdot w_r(t) \cdot x)] \leq \mathbb{E}[\mathsf{softmax}_k(x_d \cdot w_r(t) \cdot x)]$$

*Proof.* Define:

$$h_k(x) := \mathsf{softmax}_k(x) - \mathsf{softmax}_d(x)$$

Note that $h_k(x)$ is a convex function for $[x_k, x_d]$ for any $k \in [d-1]$.

Then following Jensen's inequality, we have

$$\mathbb{E}[h_k(x)] \geq h_k(\mathbb{E}[x]) \iff \mathbb{E}[\mathsf{softmax}_k(x) - \mathsf{softmax}_d(x)] \geq \mathsf{softmax}_k(\mathbb{E}[x]) - \mathsf{softmax}_d(\mathbb{E}[x])$$
$$\iff \mathbb{E}[\mathsf{softmax}_k(x)] - \mathbb{E}[\mathsf{softmax}_d(x)] \geq \mathsf{softmax}_k(\mathbb{E}[x]) - \mathsf{softmax}_d(\mathbb{E}[x])$$

Above the 2nd step is based on simple algebras.

Since $x \sim \mathcal{N}(0, \sigma'^2 \cdot I_d)$, then we have:

$$\mathbb{E}[x_d^2 \cdot w_r(t)] = w_r(t)$$
$$\mathbb{E}[x_d x_k \cdot w_r(t)] = 0$$

Besides, following Theorem I.1, when $a_r = -1$, with probability at least $1 - \delta$, we have:

$$w_r(t) < 0$$

Thus we obtain:

$$\mathsf{softmax}_k(\mathbb{E}[x]) - \mathsf{softmax}_d(\mathbb{E}[x]) \geq 0$$

Finally, we have:

$$\mathbb{E}[\mathsf{softmax}_k(x)] - \mathbb{E}[\mathsf{softmax}_d(x)] \geq \mathsf{softmax}_k(\mathbb{E}[x]) - \mathsf{softmax}_d(\mathbb{E}[x])$$
$$\geq 0$$

□

## I.3 Gradient Direction

**Lemma I.3.** *Assuming the following conditions are satisfied:*

- *Denote $v_{\min} := \min\{\frac{1}{d} \sum_{k=1}^{d}(x_{i,k} - \overline{x}_i)^2\}_{i=1}^{n}$*

*With probability at least $1 - \delta$, we have:*

- *Part 1. If $a_r = 1$, we have:*

$$\Delta w_r(t) \leq -O(\frac{n\gamma v_{\min}}{\sqrt{m}}) \cdot \exp(-O(B^2 D))$$

- *Part 2. If $a_r = -1$, we have:*

$$\Delta w_r(t) \geq O(\frac{n\gamma v_{\min}}{\sqrt{m}}) \cdot \exp(-O(B^2 D))$$

*Proof.* For $i \in [n]$, we have:

$$\begin{aligned}
\mathbb{E}[x_{i,d} y_i] &= \mathbb{E}[h_i^\top \mathcal{P}_d \cdot \mathcal{P}_{d+1}^\top h_i + h_i^\top \mathcal{P}_d \cdot \xi_{i,d} + \mathcal{P}_{d+1}^\top h_i \cdot \xi_{i,d+1}] \\
&= \mathbb{E}[h_i^\top \mathcal{P}_d \cdot \mathcal{P}_{d+1}^\top h_i] \\
&= \langle \mathcal{P}_d, \mathcal{P}_{d+1}^\top \rangle \\
&= \gamma
\end{aligned}$$

Above the first equation follows from Claim C.2 and Definition C.3, and the 2nd equation is based on $h_{i,k} \sim \mathcal{N}(0,1)$ and $\xi_{i,k} \sim \mathcal{N}(0, \sigma^2)$ independently. Basic algebras and Claim C.2 can obtain the last step.

Hence, we apply Hoeffding inequality to $\sum_{i=1}^n x_{i,d} y_i$, we have:

$$\begin{aligned}
\left| \sum_{i=1}^n x_{i,d} y_i - \sum_{i=1}^n \mathbb{E}[x_{i,d} y_i] \right| &\leq O(B^2 \sqrt{n \log(n/\delta)}) \\
&\leq O(\sqrt{n} B^3) \quad (38)
\end{aligned}$$

Above the first inequality follows from $x_{i,d} \leq B$ and $y_i \leq B$, and the second inequality follows from $\sqrt{\log(n/\delta)} \leq B$.

We can obtain:

$$\sum_{i=1}^n x_{i,d} y_i \geq n\gamma - O(\sqrt{n} B^3) \quad (39)$$

Above, the inequality can be derived from Eq. (38).

Next, we can show that:

$$\begin{aligned}
\sum_{i=1}^n (\mathsf{F}_i(t) - y_i) \cdot x_{i,d} &= \sum_{i=1}^n \mathsf{F}_i(t) x_{i,d} - \sum_{i=1}^n y_i x_{i,d} \\
&\leq \exp(O(B^2 D)) \cdot O(nRB) - n\gamma + O(\sqrt{n} B^3) \\
&\leq -n\gamma + O(\sqrt{n} B^3) \\
&\leq -O(n\gamma) \quad (40)
\end{aligned}$$

Above, the first equation is trivially obtained by simple algebra, and the second inequality follows from Part 1 of Lemma I.5, Part 1 of Lemma K.3 and Eq. (39). The third inequality follows from plugging $R \leq O(\exp(-O(B^2 D)) \cdot /(n^{0.5} B^4))$, the last step follows from $n \geq O(N/\gamma)$.

**Proof of Part 1.** When $a_r = 1$, following Lemma E.5, we have:

$$\begin{aligned}
\Delta w_r(t) &= \frac{1}{\sqrt{m}} a_r \sum_{i=1}^n (\mathsf{F}_i(t) - y_i) \cdot x_{i,d} \cdot \left( \langle \mathsf{S}_{i,r}(t), x_i^{\circ 2} \rangle - \langle \mathsf{S}_{i,r}(t), x_i \rangle^2 \right) \\
&\leq \frac{\exp(-O(B^2 D)) v_{\min}}{\sqrt{m}} a_r \sum_{i=1}^n (\mathsf{F}_i(t) - y_i) \cdot x_{i,d} \\
&\leq -O(\frac{n\gamma v_{\min}}{\sqrt{m}}) \cdot \exp(-O(B^2 D))
\end{aligned}$$

where the second step follows from Lemma I.4, the third step follows from Eq. (40).

**Proof of Part 2.** This proof is similar to the **Proof of Part 1** of this Lemma above. □

## I.4 Basic Lower Bound

**Lemma I.4.** *Assuming the following conditions are satisfied:*

- *Let $i \in [n]$ and $r \in [m]$.*

- *Let integer $t > 0$.*

- *Let training dataset $\mathcal{D} := \{(x_i, y_i)\}_{i=1}^{n} \subset \mathbb{R}^d$.*

- *Define $\mathsf{S}_{i,r}(t) \in \mathbb{R}$ as specified in Definition E.3.*

- *Define $B > 1$ as specified in Definition K.1.*

- *Define $D > 1$ as specified in Definition K.2.*

- *Let $R \in (0, 0.01)$.*

- *Let $\delta \in (0, 0.1)$.*

- *Denote $v_{\min} := \min\{\frac{1}{d} \sum_{k=1}^{d} (x_{i,k} - \overline{x}_i)^2\}_{i=1}^{n}$ where $\overline{x}_i := \frac{1}{d} \sum_{k=1}^{d} x_{i,k}$.*

*Then, with a probability no less than $1 - \delta$, we have:*

$$\langle \mathsf{S}_{i,r}(t), x_i^{\circ 2} \rangle - \langle \mathsf{S}_{i,r}(t), x_i \rangle^2 \geq \exp(-O(B^2 D)) \cdot v_{\min}$$

*Proof.* Define

$$\overline{x}_{i,r} := \langle \mathsf{S}_{i,r}(t), x_i \rangle$$

We have:

$$
\begin{aligned}
\langle \mathsf{S}_{i,r}(t), x_i^{\circ 2} \rangle - \langle \mathsf{S}_{i,r}(t), x_i \rangle^2 &= \langle \mathsf{S}_{i,r}(t), (x - \mathbf{1}_d \cdot \overline{x}_{i,r})^{\circ 2} \rangle \\
&\geq \min_{k \in [d]} \mathsf{S}_{i,r}(t) \langle \mathbf{1}_d, (x - \mathbf{1}_d \cdot \overline{x}_{i,r})^{\circ 2} \rangle \\
&\geq \min_{k \in [d]} \mathsf{S}_{i,r}(t) \cdot O(d v_x) \\
&\geq \exp(-O(B^2 D)) \cdot v_{\min}
\end{aligned}
$$

where the first two steps can be derived from simple algebras, the second step follows from Fact B.8, and the last step follows from Part 9 of Lemma K.3 and $R \leq B$. $\qquad\square$

## I.5 Model Outputs Concentration during Training

**Lemma I.5.** *Assuming the following conditions are satisfied:*

- *Let $i \in [n]$ and $r \in [m]$.*

- *Let integer $t > 0$.*

- *Define training dataset $\mathcal{D} := \{(x_i, y_i)\}_{i=1}^{n} \subset \mathbb{R}^d \times \mathbb{R}$ as specified in Definition C.3.*

- *Define $a \in \mathbb{R}^m$ as specified in Definition D.2.*

- *Define $\mathsf{S}_{i,r}(t) \in \mathbb{R}^d$ as specified in Definition E.3*

- *Define $\mathsf{F}_i(t) \in \mathbb{R}$ as specified in Definition E.4.*

- *Define $B$ as specified in Definition K.1.*

- *Define $D$ as specified in Definition K.2.*

- *Let $R \in (0, 0.01/B^2)$.*

- *Let $\delta \in (0, 0.1)$.*

*Then, with a probability at least $1 - \delta$, we have*

- *Part 1.*

$$|\mathsf{F}_i(t) - \mathsf{F}_i(0)| \leq \exp\left(O(B^2 D)\right) \cdot O(R)$$

- *Part 2.*

$$\|\mathsf{F}(t) - \mathsf{F}(0)\|_2 \leq \exp\left(O(B^2 D)\right) \cdot O(R\sqrt{n})$$

*Proof.* **Proof of Part 1.** Firstly we have

$$|\mathsf{F}_i(t) - \mathsf{F}_i(0)| = |\frac{1}{\sqrt{m}} \sum_{i=1}^{m} a_r \cdot \langle S_{i,r}(t), x_i \rangle - \frac{1}{\sqrt{m}} \sum_{i=1}^{m} a_r \cdot \langle S_{i,r}(0), x_i \rangle|$$

$$= |\frac{1}{\sqrt{m}} \sum_{i=1}^{m} a_r \cdot \langle S_{i,r}(t) - S_{i,r}(0), x_i \rangle|$$

where the first step is trivially from Definition E.4 and the second step follows from simple algebra.

Then we proceed to show that, $\forall i \in [n]$ and $r \in [m]$,

$$\begin{aligned}
|a_r \cdot \langle S_{i,r}(t) - S_{i,r}(0), x_i \rangle| &= |\langle S_{i,r}(t) - S_{i,r}(0), x_i \rangle| \\
&\leq \|S_{i,r}(t) - S_{i,r}(0)\|_2 \|x_i\|_2 \\
&\leq \sqrt{d} \cdot \exp(O(B^2 D)) \cdot O(RB^2)/d \cdot \sqrt{d} \cdot O(B) \\
&= \exp(O(B^2 D)) \cdot O(RB^3) \quad (41)
\end{aligned}$$

Above, the first equation can be obtained from Definition D.2. The second inequality is a consequence of Cauchy Inequality. The third inequality is from Part 1,13 of Lemma K.3 and the definition of $\ell_2$ norm, and the final equation is trivially from basic algebra.

Now we can use Hoeffding Inequality (Lemma B.1) to random variables $a_r \cdot \langle S_{i,r}(t) - S_{i,r}(0), x_i \rangle$, for $r \in [m]$. Besides, we have

$$\mathbb{E}[\sum_{r=1}^{m} a_r \cdot \langle S_{i,r}(t) - S_{i,r}(0), x_i \rangle] = 0$$

where this step follows from $a_r \sim \mathrm{Uniform}\{-1, +1\}$.

Also, we have:

$$\begin{aligned}
|a_r \cdot \langle S_{i,r}(t) - S_{i,r}(0), x_i \rangle| &\leq \exp(O(B^2 D)) \cdot O(RB^3) \\
&\leq \exp(O(B^2 D)) \cdot O(R) \quad (42)
\end{aligned}$$

Above, the 1st inequality is based on Eq. (41) and the 2nd inequality is based on $O(\mathrm{poly}(B)) \leq \exp(O(B^2))$.

Then, with probability at least $1 - \delta$:

$$\begin{aligned}
|\frac{1}{\sqrt{m}} \sum_{i=1}^{m} a_r \cdot \langle S_{i,r}(t) - S_{i,r}(0), x_i \rangle| &\leq \frac{1}{\sqrt{m}} \exp(O(B^2 D)) \cdot O(R) \cdot \sqrt{m \log(m/\delta)} \\
&\leq \exp(O(B^2 D)) \cdot O(RD) \\
&\leq \exp(O(B^2 D)) \cdot O(R)
\end{aligned}$$

where the first step is a consequence Hoeffding Inequality (Lemma B.1) and Eq. (42), the second step is trivially from simple algebras and Definition K.2 and the last step is derived from the fact $O(\mathrm{poly}(D)) \leq \exp(O(D))$.

**Proof of Part 2.** We have

$$\|\mathsf{F}(t) - \mathsf{F}(0)\|_2 = \sqrt{\sum_{i=1}^{n} (\mathsf{F}_i(t) - \mathsf{F}_i(0))}$$

$$\leq \exp(O(B^2 D)) \cdot O(R\sqrt{n})$$

Above, the first equation is trivially from $\ell_2$ norm, and the second inequality can be obtained by applying the result of Part 1 of this lemma and simple algebra.

Then we finished the proof. □

# J  Generalization

## J.1  Main Results 2: Attention Fails in Generalizing Sign-Inconsistent Next-step-prediction While Residual Linear Does Well

**Proposition J.1.** *Assuming the following conditions are satisfied:*

- *Let all pre-conditions in Theorem I.1 hold.*
- *Define $\mathcal{R}(\cdot)$ as specified in Definition C.4.*
- *Let $d = N$.*

*Then with a probability at least $1 - \delta$, there is not existing $w_r(t) \in \mathbb{R}^m$ satisfies:*

$$\mathcal{R}(f) \leq O(\sigma^2)$$

*Proof.* We have:

$$\sum_{i=1}^{n_{\text{test}}} a_r \cdot \mathsf{softmax}_d(x_{\text{test},i,d} \cdot w_r(t) \cdot x_{\text{test},i}) > 0$$

where this step follows from $w_r(t) < 0$ when $a_r = -1$ in Theorem I.2.

Denote:

$$\mathcal{P}_x := [\mathcal{P}_1 \quad \mathcal{P}_2 \quad \cdots \quad \mathcal{P}_{d-1}] \in \mathbb{R}^{N \times d-1}$$

and

$$\mathcal{P}_y := \mathcal{P}_{d+1} \in \mathbb{R}^N$$

Then there doesn't exist any vector $w_{\text{attn}} \in \mathbb{R}^{d-1}$ that satisfies:

$$\mathcal{P}_x w_{\text{attn}} = \mathcal{P}_y$$

□

## J.2  Residual Linear Network

**Definition J.2.** *Given an input vector $x \in \mathbb{R}^d$. Denote $w_{\text{lin}} \in \mathbb{R}^d$ as the model weight. The residual linear network is defined by:*

$$f_{\text{lin}}(x) := \langle w_{\text{lin}}, x - x_d \cdot \mathbf{1}_d \rangle + x_d$$

**Proposition J.3.** *Assuming the following conditions hold:*

- *Define $\mathcal{R}(\cdot)$ as specified in Definition C.4.*
- *Let $d = N$.*

*Then there exists and exists only one $w_{\text{lin}}^*$ that satisfies:*

$$\sum_{k=1}^{d-1} w_{\text{lin},k} \cdot \mathcal{P}_k = \mathcal{P}_{d+1} - \mathcal{P}_d$$

*Hence, we have:*

$$\mathcal{R}(f_{\text{lin}}) \leq O(\sigma^2)$$

*Proof.* Denote:

$$\mathcal{P}_x := [\mathcal{P}_1 - \mathcal{P}_d \quad \mathcal{P}_2 - \mathcal{P}_d \quad \cdots \quad \mathcal{P}_{d-1} - \mathcal{P}_d \quad \mathcal{P}_d - \mathcal{P}_d] \in \mathbb{R}^{N \times d}$$

and

$$\mathcal{P}_y := \mathcal{P}_{d+1} - \mathcal{P}_d \in \mathbb{R}^N$$

We choose:

$$w_{\text{lin}}^* := (\mathcal{P}_x^\top \mathcal{P}_x)^{-1} \mathcal{P}_x^\top \mathcal{P}_y$$

Since $d = N$, we have:

$$
\begin{aligned}
\mathcal{R}(f_{\text{lin}}) &= \lim_{n_{\text{text}} \to +\infty} \frac{1}{n_{\text{text}}} \sum_{i=1}^{n_{\text{test}}} (f_{\text{lin}}(x_{\text{test},i}) - y_{\text{test},i})^2 \\
&= \lim_{n_{\text{text}} \to +\infty} \frac{1}{n_{\text{text}}} \sum_{i=1}^{n_{\text{test}}} (\xi_{\text{test},i,d} - \xi_{\text{test},i,d+1})^2 \\
&\leq O(\sigma^2)
\end{aligned}
$$

where the last step is based on the variance of $\xi_{\text{test},i,d} - \xi_{\text{test},i,d+1}$. □

# K   Taylor Series

**Definition K.1.** *For $\delta \in (0, 0.1)$, $\sigma \in \mathbb{R}$ and a sufficiently large constant $C > 0$, we define:*

$$B := \max\{\sqrt{(1 + \sigma^2) \log(nN/\delta)}, 1\}$$

**Definition K.2.** *For $\delta \in (0, 0.1)$, $\sigma \in \mathbb{R}$ and a sufficiently large constant $C > 0$, we define:*

$$D := \max\{\sqrt{\log(m/\delta)}, 1\}$$

**Lemma K.3.** *Assuming the following conditions are satisfied:*

- *Define training dataset $\mathcal{D} := \{(x_i, y_i)\}_{i=1}^n \subset \mathbb{R}^d \times \mathbb{R}$ as specified Definition C.3.*
- *Define $B > 1$ as specified in Definition K.1.*
- *Define $D > 1$ as specified in Definition K.2*
- *Define $R := \max_{t \geq 0} \max_{r \in [m]} |w_r(t) - w_r(0)|$.*
- *Let $w(0) \in \mathbb{R}^m$ be initialized as Definition D.2 and updated by Definition D.5*
- *Define $\mathsf{u}_{i,r}(t) \in \mathbb{R}^d$ as specified in Definition E.1.*
- *Define $\alpha_{i,r}(t) \in \mathbb{R}$ as specified in Definition E.2.*
- *Define $\mathsf{S}_{i,r}(t) \in \mathbb{R}$ as specified in Definition E.3.*
- *Let $R \in (0, 0.01/B^2)$.*

- $\forall i \in [n], r \in [m], k \in [d], t \geq 0$.
- Let $\delta \in (0, 0.1)$.

*Consequently, with probability at least $1 - \delta$, we have:*

- *Part 1. $|x_{i,k}| \leq O(B)$.*
- *Part 2. $|w_r(0)| \leq O(D)$.*
- *Part 3. $|w_r(t)| \leq O(D + R)$.*
- *Part 4. $\exp(-O(B^2 D)) \leq \mathsf{u}_{i,r,k}(0) \leq \exp(O(B^2 D))$.*
- *Part 5. $\exp(-O(B^2(D + R))) \leq u_{i,r,k}(t) \leq \exp(O(B^2(D + R)))$.*
- *Part 6. $d \cdot \exp(-O(B^2 D)) \leq \alpha_{i,r}(0) \leq d \cdot \exp(O(B^2 D))$.*
- *Part 7. $d \cdot \exp(-O(B^2(D + R))) \leq \alpha_{i,r}(t) \leq d \cdot \exp(O(B^2(D + R)))$.*
- *Part 8. $\frac{\exp(-O(B^2 D))}{d} \leq \mathsf{S}_{i,r,k}(0) \leq \frac{\exp(O(B^2 D))}{d}$.*
- *Part 9. $\frac{\exp(-O(B^2(D+R)))}{d} \leq \mathsf{S}_{i,r,k}(t) \leq \frac{\exp(O(B^2(D+R)))}{d}$.*
- *Part 10. $|u_{i,r,k}(t) - u_{i,r,k}(0)| \leq \exp(O(B^2 D)) \cdot O(RB^2)$.*
- *Part 11. $|\alpha_{i,r}(t) - \alpha_{i,r}(0)| \leq d \exp(O(B^2 D)) \cdot O(RB^2)$.*
- *Part 12. $|\alpha_{i,r}(t)^{-1} - \alpha_{i,r}(0)^{-1}| \leq \exp(O(B^2 D)) \cdot O(RB^2)/d$.*
- *Part 13. $|\mathsf{S}_{i,r,k}(t) - \mathsf{S}_{i,r,k}(0)| \leq \exp(O(B^2 D)) \cdot O(RB^2)/d$*

*Proof.* **Proof of Part 1.** We have

$$x_{i,k} = u_{i,k} + \xi_{i,k}$$
$$= \langle \mathcal{P}_{i,k}, h_{i,1} \rangle + \xi_{i,k}$$

Above, the first equation is trivially from Definition C.3, and the second equation is also trivially from Claim C.2. And we have $\xi_{i,k} \sim \mathcal{N}(0, \sigma^2)$ and $h_{i,1} \sim \mathcal{N}(0, I_N)$ following from Definition C.3 and $\|\mathcal{P}_{i,k}\|_2 = 1$ from Claim C.2.

Hence, we can have

$$\langle \mathcal{P}_{i,k}, h_{i,1} \rangle \sim \mathcal{N}(0, 1)$$

where the step is a consequence of Fact B.4.

And we have

$$x_{i,k} \sim \mathcal{N}(0, 1 + \sigma^2)$$

Thus, with a probability $1 - \delta$:

$$|x_{i,k}| \leq C\sqrt{(1 + \sigma^2)log(1/\delta)}$$
$$\leq O(B)$$

Above, the first inequality is derived by using Fact B.5, and the second inequality is trivially from the Definition of $B$ (Definition K.1).

**Proof of Part 2.** We have

$$w_r(0) \sim \mathcal{N}(0, 1)$$

Above the step can be trivially from Definition D.2.

Thus, with a probability no less than $1 - \delta$, we have

$$|w_r(0)| \leq C\sqrt{\log(1/\delta)}$$
$$= O(D)$$

Above the first inequality is a consequence of Fact B.5, and the second equation is trivially from the Definition K.2.

**Proof of Part 3.** By following the Lemma statement, we can show that

$$|w_r(t) - w_r(0)| \leq R$$

where the step can be obtained from the definition of $R$.

Then we have

$$|w_r(t)| \leq |w_r(0) + R|$$
$$\leq |w_r(0)| + |R|$$
$$\leq O(D + R)$$

Above, the first inequality is a result of simple algebra, the 2nd inequality applies triangle inequality, and the last step is trivially from Part 2 of this lemma.

**Proof of Part 4.** We have

$$|x_{i,d} \cdot w_r(0) \cdot x_{i,k}| \leq O(B^2 D)$$

The inequality above can be trivially obtained by using Part 1,2 of this lemma.

Hence, we get

$$u_{i,r,k}(0) = \exp(x_{i,d} \cdot w_r(0) \cdot x_{i,k})$$
$$\in [\exp(-O(B^2 D)), \exp(O(B^2 D))]$$

Above, the first equation is trivially from Definition E.1, and the second step is derived by using basic algebra.

**Proof of Part 5.** We have

$$|x_{i,d} \cdot w_r(t) \cdot x_{i,k}| \leq O(B^2 \cdot (D + R))$$

where this step combines Part 1,3 of this Lemma.

Hence, we get

$$u_{i,r,k}(t) = \exp(x_{i,d} \cdot w_r(t) \cdot x_{i,k})$$
$$\in [\exp(-O(B^2(D + R))), \exp(O(B^2(D + R)))]$$

where the 1st step is trivially from Definition E.1, and the 2nd step applies basic algebra.

**Proof of Part 6.** We have

$$\alpha_{i,r}(0) = \langle \mathsf{u}_{i,r}(0), \mathbf{1}_d \rangle$$
$$= \sum_{k=1}^{d} \mathsf{u}_{i,r,k}(0)$$

where the first step is trivially from Definition E.2, and the second step applies simple algebra.

Thus we have

$$d \cdot \exp(-O(B^2 D)) \leq \alpha_{i,r}(0) \leq d \cdot \exp(O(B^2 D))$$

where this step can be trivially derived from Part 4 of this lemma.

**Proof of Part 7.** We have

$$\alpha_{i,r}(t) = \langle \mathsf{u}_{i,r}(t), \mathbf{1}_d \rangle$$

$$= \sum_{k=1}^{d} \mathsf{u}_{i,r,k}(t)$$

where the first step is trivially from Definition E.2, and the second step comes from the definition of the inner product. Thus we have

$$d \cdot \exp(-O(B^2(D+R))) \le \alpha_{i,r}(t) \le d \cdot \exp(O(B^2(D+R)))$$

where this step can be obtained by Part 5 of this lemma.

**Proof of Part 8.** We have

$$\mathsf{S}_{i,r,k}(0) = \alpha_{i,r}(0)^{-1} \cdot \mathsf{u}_{i,r,k}(0)$$

where this step follows from Definition E.3. Then we have

$$\frac{\exp(-O(B^2D))}{d} \le \mathsf{S}_{i,r,k}(0) \le \frac{\exp(O(B^2D))}{d}$$

where this step can be obtained by combining Parts 4,6 of this lemma.

**Proof of Part 9.** We have

$$\mathsf{S}_{i,r,k}(t) = \alpha_{i,r}(t)^{-1} \cdot \mathsf{u}_{i,r,k}(t)$$

where this step follows from Definition E.3. Then we have

$$\frac{\exp(-O(B^2(D+R)))}{d} \le \mathsf{S}_{i,r,k}(t) \le \frac{\exp(O(B^2(D+R)))}{d}$$

where this step can be obtained by combining Part 5,7 of this lemma.

**Proof of Part 10.** We have

$$
\begin{aligned}
&|\mathsf{u}_{i,r,k}(t) - \mathsf{u}_{i,r,k}(0)| \\
=~& |\exp(x_{i,d} \cdot w_r(t) \cdot x_{i,k}) - \exp(x_{i,d} \cdot w_r(0) \cdot x_{i,k})| \\
=~& |\exp(x_{i,d} \cdot w_r(0) \cdot x_{i,k}) \cdot (\exp\left(x_{i,d}x_{i,k} \cdot (w_r(t) - w_r(0))\right) - 1)| \\
=~& \left|\exp(x_{i,d} \cdot w_r(0) \cdot x_{i,k}) \cdot \left(x_{i,d}x_{i,k} \cdot (w_r(t) - w_r(0)) + \Theta(1) \cdot x_{i,d}^2 x_{i,k}^2 \cdot (w_r(t) - w_r(0))^2\right)\right| \\
\le~& |\exp(x_{i,d} \cdot w_r(0) \cdot x_{i,k}) \cdot (RB^2 + \Theta(1) \cdot R^2 B^4)| \\
\le~& |\exp(x_{i,d} \cdot w_r(0) \cdot x_{i,k}) \cdot O(RB^2)| \\
=~& |\mathsf{u}_{i,r,k}(0) \cdot O(RB^2)| \\
\le~& \exp(O(B^2D)) \cdot O(RB^2)
\end{aligned}
$$

Above the first equation is trivially from Definition E.1, the second equation can be obtained by using simple algebra, the third equation is a consequence Fact B.6, the fourth inequality combines the result of Part 1 of this lemma and $|w_r(t) - w_r(0)| \le R$, the fifth inequality applies simple algebra, the sixth step comes from Definition E.1 and the last step is derived from Part 6 of this lemma.

**Proof of Part 11.** We have

$$
\begin{aligned}
&|\alpha_{i,r}(t) - \alpha_{i,r}(0)| \\
=~& |\sum_{k \in [d]} \mathsf{u}_{i,r,k}(t) - \sum_{k \in [d]} \mathsf{u}_{i,r,k}(0)| \\
\le~& \sum_{k \in [d]} |\mathsf{u}_{i,r,k}(t) - \mathsf{u}_{i,r,k}(0)| \\
\le~& d \cdot \exp(O(B^2D)) \cdot O(RB^2)
\end{aligned}
$$

Above the first equation is trivially from Definition E.2, the second step can be obtained by using triangle inequality, and the last step is derived from Part 10 of this lemma.

**Proof of Part 12.** We have

$$
\begin{aligned}
|\alpha_{i,r}(t)^{-1} - \alpha_{i,r}(0)^{-1}| &= \alpha_{i,r}(t)^{-1} \cdot \alpha_{i,r}(0)^{-1} \cdot |\alpha_{i,r}(0) - \alpha_{i,r}(t)| \\
&\leq d \cdot \exp(O(B^2 D)) \cdot O(RB^2) \cdot \frac{\exp(O(B^2(D+R)))}{d} \cdot \frac{\exp(O(B^2 D))}{d} \\
&= \exp(O(B^2 D)) \cdot O(RB^2)/d
\end{aligned}
$$

Above, the first equation is based on simple algebra, the 2nd step is due to Parts 6, 7, and 10 of this lemma, and the last step can be obtained from applying basic algebras and the fact that $R \ll D$.

**Proof of Part 13.** We have

$$
\begin{aligned}
&|S_{i,r,k}(t) - S_{i,r,k}(0)| \\
&= |\alpha_{i,r}(t)^{-1} \cdot \mathsf{u}_{i,r,k}(t) - \alpha_{i,r}(0)^{-1} \cdot \mathsf{u}_{i,r,k}(0)| \\
&= |(\alpha_{i,r}(t)^{-1} \cdot \mathsf{u}_{i,r,k}(t) - \alpha_{i,r}(t)^{-1} \cdot \mathsf{u}_{i,r,k}(0)) + (\alpha_{i,r}(t)^{-1} \cdot \mathsf{u}_{i,r,k}(0) - \alpha_{i,r}(0)^{-1} \cdot \mathsf{u}_{i,r,k}(0))| \\
&\leq |\alpha_{i,r}(t)^{-1}| \cdot |\mathsf{u}_{i,r,k}(t) - \mathsf{u}_{i,r,k}(0)| + |\mathsf{u}_{i,r,k}(0)| \cdot |\alpha_{i,r}(t)^{-1} - \alpha_{i,r}(0)^{-1}| \\
&\leq \exp(O(B^2 D)) \cdot O(RB^2)/d + \exp(O(B^2 D)) \cdot O(RB^2)/d \\
&\leq \frac{\exp(O(B^2 D))}{d} \cdot O(RB^2)
\end{aligned}
$$

Above the first equation is trivially from Definition E.3, the 2nd step is due to simple algebra, the 3rd step can be obtained by applying triangle inequality, the fourth step combines Parts 4, 7, 10, 12 of this lemma, and the final step is based on simple algebra.

$\square$

