# OpenReview forum: "Curse of Attention: A Kernel-Based Perspective for Why Transformers Fail to Generalize on Time Series Forecasting and Beyond"
_CPAL.cc/2025/Proceedings_Track — CPAL 2025 (Proceedings Track) Poster_

### Official Review · Reviewer_nUL5 · 2025-01-09
**A Kernel-Based Perspective for Why Transformers Fail to Generalize on Time Series Forecasting and Beyond**

**Rating:** 7
**Confidence:** 2

**Review:**

The authors show that a simple residual linear model avoids this pitfall and achieves better generalization, even against highly over-parameterized attention networks. They conclude that this failure mode is especially pronounced in sign-inconsistent next-step prediction tasks, underscoring fundamental limitations of standard transformers in certain TSF settings.

Strengths:

(1) Provides a novel theoretical explanation using NTK analysis that clarifies why transformers can fail on time series forecasting.

(2) identifies and proves the asymmetric learning phenomenon, offering a solid mathematical foundation.

Weakness:

(1) Analyses are conducted on a simplified two-layer “attention-like” model, which may not fully capture the complexity of real-world transformer architectures.

(2) limited empirical validation

---

### Official Review · Reviewer_u2Ex · 2025-01-10
**Reviews**

**Rating:** 6
**Confidence:** 2

**Review:**

**Summary**

The paper provides a theoretical framework for explaining the poor performance of transformers in time series forecasting (TSF). It introduces the concept of Asymmetric Learning as a central issue in attention mechanisms.

**Strengths:**
1. Provides a novel theoretical explanation for why transformers underperform in time series forecasting (TSF), introducing the concept of Asymmetric Learning.

2. this paper Offers rigorous mathematical analysis using the neural tangent kernel (NTK) framework.


**Weakness:**

1. Lack of empirical results on TSF datasets.

2. Clarity could be improved with more intuitive explanations and visual aids.

---

### Official Review · Reviewer_X1zr · 2025-01-14

**Rating:** 6
**Confidence:** 3

**Review:**

Strengths:

- Novel theoretical explanation for why transformers underperform linear models on TSF
- Proves results on simplified attention model and superiority of residual linear network
- Experiments validate theoretical findings

Suggestions:
- Strong assumptions and simplified model limit ...
- The scope is purposefully restricted to time series data and sign inconsistency, as reflected in the title. However, it would be interesting to explore if the core insights on asymmetric learning extend more broadly, as hinted by "and Beyond". Future work testing this framework on other data types and tasks could reveal the generality of the findings.
- While well-written overall, some of the mathematical notation and proofs are quite dense. Providing additional intuition or visual aids explaining key concepts like asymmetric learning would improve accessibility for the reader.
- Figure 1 (b) is not clear to me. For example, what does the red line mean, and which is X_pre or X_last? The color seems very confusing to me. ..

---

### Meta-Review · Area_Chair_DEkr · 2025-02-02

**Recommendation:** Accept (Poster)
**Confidence:** 3

**Metareview:**

The paper presents a theoretical explanation for the poor performance of transformers on time series forecasting (TSF) tasks, introducing the concept of Asymmetric Learning. The reviewers appreciate the theoretical contributions and the validation of theoretical findings through experiments. However, concerns were raised regarding the paper’s reliance on a simplified two-layer model (which is typical) and the limited empirical validation, which may not fully capture real-world transformer behavior. Despite these limitations, the reviewers found the work compelling and insightful, leaning toward acceptance.

---

### Decision · Program_Chairs · 2025-02-11

Accept (Poster)